# Multiomic ALS signatures highlight subclusters and sex differences suggesting the MAPK pathway as therapeutic target

Lucas Caldi Gomes [1,14], Sonja Hänzelmann [2,3,4,14], Fabian Hausmann [3,4], Robin Khatri [3,4], Sergio Oller[3,4], Mojan Parvaz[1], Laura Tzeplaeff [1], Laura Pasetto[5], Marie Gebelin[6], Melanie Ebbing[3,4], Constantin Holzapfel[3,4], Stefano Fabrizio Columbro[5], Serena Scozzari[5], Johanna Knöferle[1], Isabell Cordts[1], Antonia F. Demleitner [1], Marcus Deschauer[1], Claudia Dufke [7], Marc Sturm [7], Qihui Zhou[8,9], Pavol Zelina[10], Emma Sudria-Lopez[10], Tobias B. Haack [7,11], Sebastian Streb [12], Magdalena Kuzma-Kozakiewicz[13], Dieter Edbauer [8,9], R. Jeroen Pasterkamp [10], Endre Laczko [12], Hubert Rehrauer [12], Ralph Schlapbach[12], Christine Carapito [6], Valentina Bonetto [5], Stefan Bonn [3,4] ✉ & Paul Lingor [1,8,9] ✉

Amyotrophic lateral sclerosis (ALS) is a debilitating motor neuron disease and lacks effective disease-modifying treatments. This study utilizes a comprehensive multiomic approach to investigate the early and sex-specific molecular mechanisms underlying ALS. By analyzing the prefrontal cortex of 51 patients with sporadic ALS and 50 control subjects, alongside four transgenic mouse models (C9orf72-, SOD1-, TDP-43-, and FUS-ALS), we have uncovered significant molecular alterations associated with the disease. Here, we show that males exhibit more pronounced changes in molecular pathways compared to females. Our integrated analysis of transcriptomes, (phospho)proteomes, and miRNAomes also identified distinct ALS subclusters in humans, characterized by variations in immune response, extracellular matrix composition, mitochondrial function, and RNA processing. The molecular signatures of human subclusters were reflected in specific mouse models. Our study highlighted the mitogen-activated protein kinase (MAPK) pathway as an early disease mechanism. We further demonstrate that trametinib, a MAPK inhibitor, has potential therapeutic benefits in vitro and in vivo, particularly in females, suggesting a direction for developing targeted ALS treatments.

Amyotrophic lateral sclerosis (ALS) is the most frequent motor neuron disease, is more common in males, and leads to paralysis and death within a few years of symptom onset on average[1]. While most cases are sporadic (sALS) with no family history, approximately 10% of cases have a genetic cause (fALS). The most common genetic variants include *C9orf72*, *SOD1*, *TARDBP*, and *FUS*, with around 10% of patients with sALS carrying disease-causing mutations[2]. The etiology of sALS remains unclear, and effective disease-modifying treatments for the disease are currently unavailable[3]. Enhancing our understanding of early disease mechanisms could help identify the diagnostic and prognostic biomarkers and uncover more efficient therapeutic drug targets. Although direct analysis of affected nervous system tissues

remains the gold standard in neuropathology, patient samples are only available postmortem and in limited quantities. These limitations increase the risk of primarily describing end stages of the disease and obscuring mechanisms occurring in earlier phases, which are potentially more suitable drug targets. As in Alzheimer's or Parkinson's disease, ALS is believed to spread over time−from the motor cortex to other cortical brain areas[4−6]. Numerous studies have analyzed the motor cortex in ALS[7−9]. However, as this area is the most severely affected by ALS, it primarily reflects the final stages of the disease[6]. In contrast, the prefrontal cortex (PFC) in Brodmann area 6 typically exhibits only intermediate TDP-43 pathology at the time of death[6], suggesting that analyzing this area could provide insights into earlier disease-mediated alterations.

Previous investigations of ALS brain tissue have mainly focused on individual molecular subsets, such as transcripts[7,9,10], miRNAs[11], and proteins[12], suggesting that ALS is a complex and heterogeneous disease. Recent studies have identified potentially distinct ALS populations, which were stratified into different subclusters based on transcriptomics and gene set enrichment analyzes (GSEA)[9,13].

In this study, we deciphered early ALS disease mechanisms by profiling the transcriptomes, miRNAomes, and (phospho)proteomes in the PFCs of patients with ALS and four ALS mouse models. Our findings highlighted significant sex differences and demonstrated that ALS is not a homogeneous disease but comprises different molecular subtypes that correlate with individual transgenic mouse models of the disease. Multiomic data integration identified several known ALS disease mechanisms, but also less-prioritized pathways, such as the MAPK pathway. Validation studies in vitro and in vivo underscored MEK2 as a potential target for early therapeutic interventions.

## Results

### Cohort and data description

We conducted an in-depth multiomic characterization of the human postmortem PFC, (Brodmann area 6) from 51 patients with neuropathologically confirmed ALS and 50 control (CTR) patients, to describe early molecular changes in sALS (Fig. 1a, Table 1, Supplementary Data 1). On average, 19.641 transcripts, 736 miRNAs (mature miRNAs and hairpin precursors), and 2.344 proteins were detected per sample (Supplementary Fig. 1a−c). A *C9orf72* repeat expansion was detected in one patient with ALS, and another individual carried a pathogenic variant of *NEK1*(c.3107 C > G, p.Ser1036Ter)[14] (Supplementary Fig. 2, Supplementary Data 2). For clarity and accuracy, we refer to all cases simply as "ALS". Four transgenic mouse models were analyzed to identify parallels with human ALS. To allow for comparability with the human tissue, PFCs from presymptomatic/early disease stages were collected from C9orf72-, SOD1-, TDP-43-, or FUS-ALS. An equal distribution of wild-type and transgenic littermates, as well as sexes, was ensured (n = 20 per model, sex, condition, survival and sampling time points depicted in the Methods and in Supplementary Fig. 3). In mice, an average of 17.020 transcripts, 842 miRNAs (mature miRNAs and hairpins), 2.568 proteins, and 6.755 phosphosites were detected (Supplementary Fig. 1a−b). The overall sample quality was consistently high for all omics analyzes (Supplementary Fig. 4).

### Transcriptomic analysis reveals sex-specific alterations in ALS-affected PFC tissue

Principal component analysis was performed to evaluate the impact of disease, sex, and sample origin on the transcriptome, revealing a moderate separation by condition (silhouette score: 0.11 [ALS]; −0.03 [CTR]) but a separation by sex (silhouette score: 0.29 [male], 0.15 [female]) (Fig. 1b). Consequently, we analyzed differentially expressed genes (DEGs) separately for males and females (Supplementary Data 3). The number of DEGs was significantly higher in males (n = 73) than in females (n = 2), which was confirmed by down-sampling analysis using 20 bootstraps (Fig. 1c, Supplementary Data 4). Similar to

transcriptional changes, also differential alternative splicing (DAS) analysis revealed marked sex differences in the abundance of different splicing events. We observed more DAS events in male human samples (Fig. 1d). DAS was observed in males for *CLTB*, *TPRN*, *NRN1*, and *CAMK2N1* and in females for *TPRN* and the gene encoding TMEM170A-CFDP1, a readthrough transmembrane protein (AC009163.5)(Supplementary Data 5). TPRN, a stereocilium-associated protein previously described only in non-syndromic deafness, may play additional roles in the pathogenesis of ALS. Overrepresentation analysis using GO showed enrichment for negative regulation of ERK1 and ERK2 cascade and negative regulation of MAPK cascade (p.adj <0.1) for male samples (Supplementary Data 6).

Enrichment analyzes (KEGG) of transcriptomics results for males showed significant enrichment for several synapse-related pathways (retrograde endocannabinoid signaling, synaptic vesicle cycle, long-term potentiation, glutamatergic synapse), as well as pathways related to immune response, extracellular matrix (ECM), and diverse protein processing and protein metabolism terms (which are important components of the KEGG pathway for neurodegenerative diseases [hsa05022]) (Fig. 1e, Supplementary Fig. 5, Supplementary Data 7). For females, the most significant results were related to ribosomal function and oxidative phosphorylation, as well as mitochondria-related terms. Further significantly enriched categories included neurodegenerative disorders and cell adhesion molecules. Oxidative stress was inferred from the enrichment of oxidative phosphorylation pathways in both males and females, and the MAPK pathway was frequently enriched for weighted gene co-expression network analysis (WGCNA[15]) (details in Supplementary Data 8 and in the Methods section).

### Stratification of human ALS into four molecular subclusters based on transcriptomic data

Based on frequently enriched terms for transcriptomics results (Supplementary Data 7), we conducted hierarchical clustering analyzes for pathways of interest (Fig. 1f). This analysis revealed four distinct clusters labeled as C1−C4. The regulation of immune response served as a dichotomizing factor, distinguishing patients with ALS into C1 and C2 vs. C3 and C4. At the second level of arborization, the ECM played a primary role (C1 vs. C2), along with synaptic function and protein folding (C3 vs. C4, Fig. 1f). These clusters are reminiscent, but not identical to previously proposed subtypes[9], where C1 and C2 align with ALS-Ox (oxidative stress) and showing less resemblance to ALS-TE (elevated transposable element expression), while C3 and C4 correspond to the ALS-Glia (glial dysfunction) subtype (Supplementary Fig. 6).

Finally, to characterize the clusters using similarly regulated RNA networks, we performed WGCNA[15], resulting in 20 modules (Fig. 1g, Supplementary Data 8). The turquoise module was enriched for mitochondrial respiration and positively correlated with C1 and C2, and driven by neuronal alterations, suggesting increased oxidative respiration in PFC neurons. The yellow module, enriched for synaptic function, exhibited a similar regulation in C1 and C2 (Fig. 1h). In contrast, the tan and lightcyan modules, enriched for immune response and RNA splicing, were positively correlated with C3 and C4 (Fig. 1g). These findings indicate that molecular subclusters and sex-specific differences drive heterogeneity in the PFC of patients with ALS.

### Male-predominant deregulation of miRNA and protein expression, and integration of multiomic data

To explore the role of miRNA-mediated regulation in ALS[16], we analyzed small RNAs, confirming previously identified sex differences. Male ALS patients exhibited more significant downregulation in both mature miRNAs and miRNA hairpins compared to females (Fig. 2a, b). Specifically, males showed 17 mature DE miRNAs (15 downregulated, 2 upregulated) while females exhibited 9 DE miRNAs (4 downregulated,

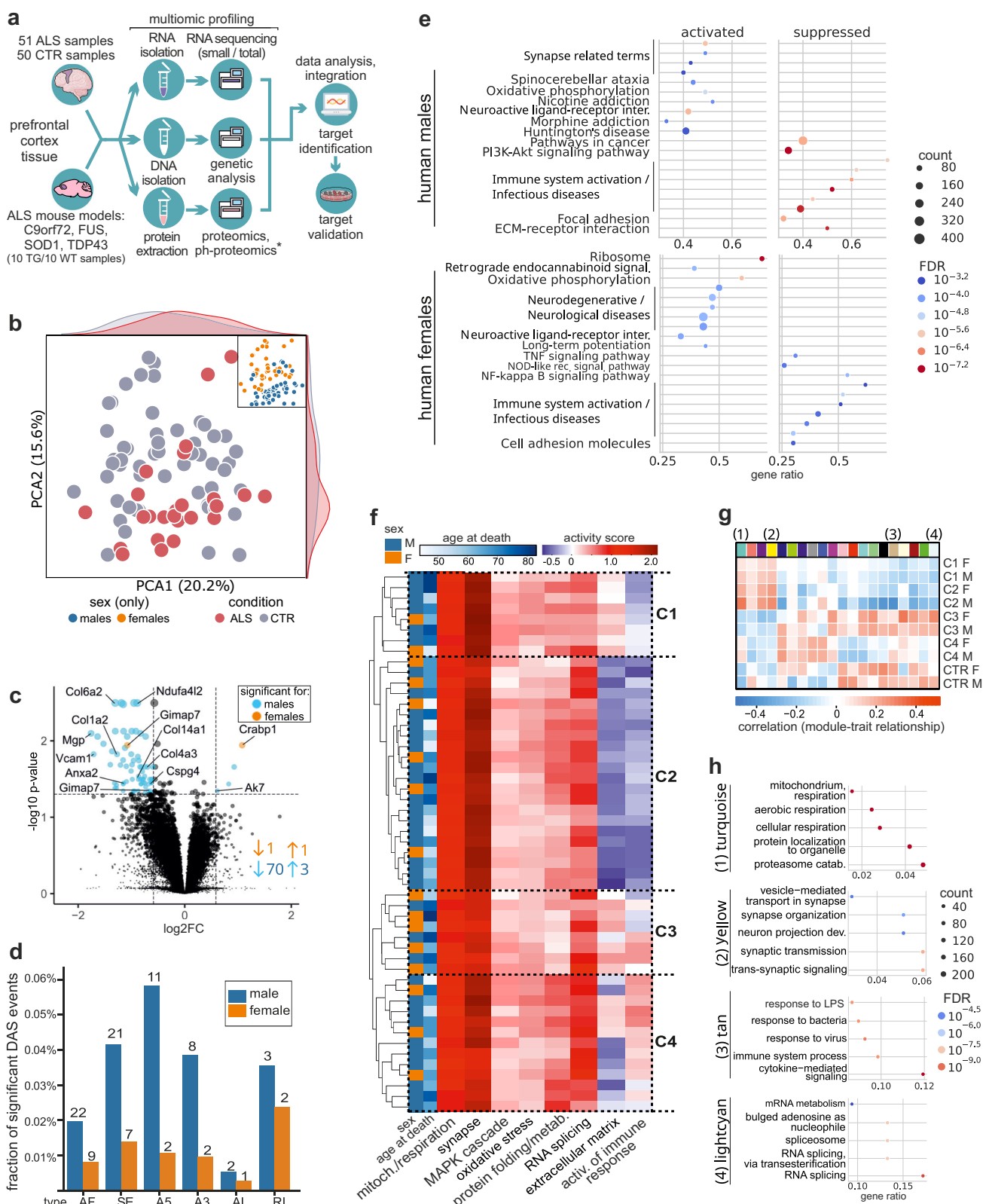

5 upregulated) (p < 0.1, fold change > 1.5, Supplementary Data 9). Further analysis of miRNA hairpins highlighted sex-dependent differences in early miRNA biogenesis. Males demonstrated a more pronounced dysregulation with 82 DE hairpins (71 downregulated, 11 upregulated), while females displayed 13 DE hairpins (11 downregulated, 1 upregulated). This suggests potential defects in miRNA biogenesis as an early disease mechanism.

To assess the significance of miRNA expression changes in our human disease clusters, we conducted a cluster-specific DE analysis (Fig. 2c, Supplementary Data 10). Only miRNAs with a minimum fold change of 0.1 and significance in at least two clusters were considered, ensuring the identification of robust expression alterations. Clusters 1, 2, and 4 exhibited distinctive miRNA profiles, while such changes were not observed in cluster 3. In more detail, miRNA miR-4472, modulating

**Fig. 1 | Identification of transcriptomic subclusters and sex-specific differences in human ALS patients. a** Overview of the sample processing workflow. Prefrontal cortex samples were prepared for multiomics experiments from the human cohort (51 ALS/50 CTR samples), as well as from four selected ALS mouse models (C9orf72, FUS, SOD1, and TDP-43; 10 Tg/10 CTR animals per group). Panel **a** was created with BioRender.com released under a Creative Commons Attribution-NonCommercial-NoDerivs 4.0 International license. **b** Principal component analysis (PCA) on the 500 most variable genes of the human samples. Blue and orange indicate the sex, and the condition is indicated in pink (ALS) or gray (CTR). **c** Volcano plot of deregulated proteins in humans. *x*-axis: log2 fold change; *y*-axis: -log10 p-value for each protein (DE analysis done with the limma package in R, two-sided test; Benjamini-Hochberg multiple test correction). Blue and orange circles indicate significant differential changes: left side, decrease (low ALS); right side, increase (high ALS). **d** Differential alternative splicing (DAS) analysis. The plot displays the results for human male and female samples for various splice events, i.e., alternative exon (AE), skipped exon (SE), alternative 5′-splice site (A5), alternative 3′-splice site

(A3), alternative last exon (AL), and retained intron (RI) events. Each event is represented by a separate bar, the height of which represents the fraction of significant events in ALS vs. CTR. Blue: male results; orange: female results. **e** Sex-specific enrichment analysis reveals crucial pathways in neurodegeneration and ALS pathology. **f** Hierarchical clustering analysis of enriched pathways and literature insights reveals four distinct clusters (C1–C4) in ALS patients. Immune response regulation dichotomizes patients into C1/C2 vs. C3/C4, while ECM, synaptic function, and protein folding further differentiate C1 vs. C2 and C3 vs. C4. **g** Heatmap showing modules from weighted gene co-expression network analysis (WGCNA) associated with the clusters through similarly regulated RNA networks. **h** Pathway enrichment per WGCNA cluster (top hits). The turquoise module, upregulated in C1 and C2, especially in males, is enriched for mitochondrial respiration, suggesting increased oxidative activity in PFC neurons. The yellow module, associated with synaptic function, exhibits a similar regulation in C1 and C2, while the tan and lightcyan modules, enriched for immune response and RNA splicing, respectively, are upregulated in C3 and C4.

E cadherin and vimentin via *RGMA*, presented the overall highest significant fold change in cluster 2. miR-181c-3p, a miRNA proposed as a circulating biomarker for ALS[17], was downregulated in both clusters 1 and 2. In cluster 4 the miR-340-5p, which is neuronal injury-related, presented the highest fold change[18]. Overall, miRNA dysregulation appeared particularly important for clusters 2 and 4, while less pronounced effects were seen in clusters 1 and 3.

We also assessed the proteomic signatures of early-stage ALS using mass spectrometry in the same samples. Here, we detected 379 and 251 differentially expressed proteins (DEPs) in males and females, respectively, with annexin A2 (ANXA2) being the only protein downregulated in both sexes (p < 0.1). Interestingly, we identified several neurodegeneration-related proteins, such as MATR3, SPART, and

SCNA (involved in the genetic forms of ALS, spastic paraplegia, and Parkinson's disease, respectively)[19–21] (Fig. 2d, Supplementary Data 11). The projection of transcriptomic clusters onto proteomic data did not reproduce the subclustering, likely because of the much smaller number of mapped entities (Supplementary Fig. 7). Functional enrichment and unsupervised clustering identified relevant pathways in both sexes, such as synaptic function, immune response, and ECM/cytoskeleton (Fig. 2e, f). In contrast, transmembrane transport, lipid metabolism, development, catalytic activity and ERK1/2 signaling were enriched in females, whereas cell metabolism and tyrosine kinase-related pathways were enriched in males (Fig. 2e, f, Supplementary Data 12).

After using various omics modalities to identify the molecular pathways associated with early ALS, we employed a biologically motivated approach that focused on identifying valid interaction triplets involving miRNAs, transcripts, and proteins in male and female samples. The triplet miR-769-3p−*ANXA2*−ANXA2 was particularly intriguing (Supplementary Fig. 8, Supplementary Data 13), since ANXA2 is involved in angiogenesis and autophagy[22], which are mechanisms known to be altered in ALS[23,24].

Next, we integrated transcriptomic, small RNA, and proteomic data using Multi-Omics Factor Analysis (MOFA)[25]. Because sex was an important differentiating factor (Fig. 2g), MOFA was performed by sex (Fig. 2h, Supplementary Fig. 9, Supplementary Data 14). In males, factor 1, mainly driven by hairpin miRNAs, explained 23.7% of the variance. Downregulation of hsa-miR-7851, −1285-1, −5096, and a cluster of −1273 isoforms strongly contributed to its weight (Supplementary Fig. 10). Transcriptome-based factor 3 correlated best with disease condition and was driven by genes responsible for vesicular function (*RAB3C, NSF*), cell survival (*BCL2, BHLHB9*), and RNA metabolism (*SNORA73B, RN7SL2*) (Fig. 2h). Proteome-dominated factor 4 contains ZO2 and CD44, which are involved in myelination and blood−central nervous system barrier (BCNSB) formation, respectively. Finally, factor 7, which also showed a strong correlation with the disease condition, was dominated by neurofilament heavy, medium, and light polypeptides (NfH, NfM, and NfL, respectively) as well as proteins involved in $Ca^{2+}$-binding (HPCL4) and ECM formation (PGCA) (Supplementary Data 14).

In females, factors 1–3 explained 42.6% of the variance, but factors 10 and 12 correlated best with the disease condition (Fig. 2h). Synaptic genes, such as *RAB3C, NAPB*, and *SNAP25*, contribute to factor 1 and are upregulated in ALS. The hsa-miR-1285-1, miR-5096, and miR-1273 clusters also contributed to factor 2. Interestingly, MAPK1 plays a central role in the miR-1273 target network (Supplementary Fig. 11). Factor 3 (similar to male factor 4) included the oligodendrocyte and myelin markers CD44 and ZO2. Factor 10 contained the antiproteases SERPINA1 and SERPINA3[26] as well as chitinases CHI3L1 and CHI3L2, which are known biomarkers for ALS[27]. Factor 12 showed a negative

## Table 1 | Summary of the demographics of the human cohort

| | Control | ALS | *p*-value |
|---|---|---|---|
| **Donors (number)** | 50 | 51 | |
| **Age at death (years)** | 75 (43–94) | 67 (44–83) | **0.0032** |
| **Sex female** | 56.0% | 31.4% | **0.0163** |
| **Postmortem interval (hours)** | 22 (5–95) | 23 (2.5–98) | 0.4324 |
| Unknown | 8.0% | 5.9% | |
| **Onset** | | | |
| Bulbar | - | 17.6% | |
| Spinal | - | 37.3% | |
| Upper limb | - | 31.6% | |
| Lower limb | - | 63.1% | |
| Limb unknown | - | 5.3% | |
| Thoracic | - | 2.0% | |
| Unknown | - | 43.1% | |
| **Disease duration (years)** | - | 3 (1–28) | |
| Unknown | - | 39.2% | |
| **Brain bank origin** | | | **<0.0001** |
| NBB | 18.0% | 17.6% | |
| Oxford BB | 20.0% | 27.5% | |
| ICL MS & PD TB | 38.0% | 0.0% | |
| London NDBB | 24.0% | 54.9% | |

*ALS* amyotrophic lateral sclerosis. Continuous variables are presented as medians and ranges. Categorical variables were compared using a two-sided Fisher's exact test (for single groups) or a two-tailed chi-square test (for multiple groups); continuous variables were compared using the Mann–Whitney test. The p-values shown are uncorrected for multiple testing; p-values less than 0.05 are indicated in bold. NBB: The Netherlands Brain Bank; OBB: Oxford Brain Bank; ICL MS&PD TB: Imperial College London—Multiple Sclerosis and Parkinson's Tissue Bank; London NDBB: London Neurodegenerative Diseases Brain Bank.

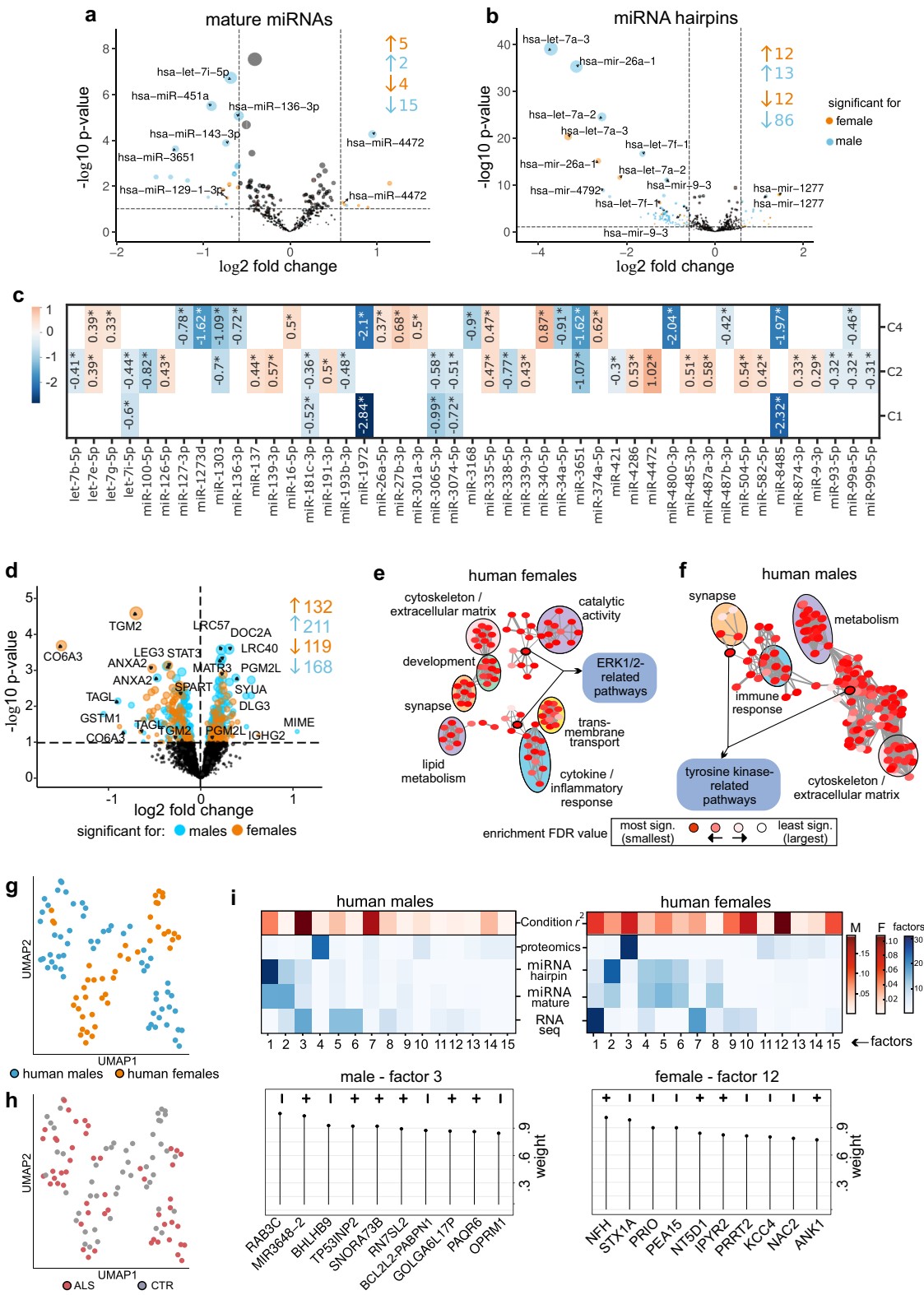

correlation with MAPK–ERK1/2 regulator PEA15 and a positive correlation with NFH (Supplementary Fig. 12). MOFA analysis underlined the mechanisms identified in the individual omics analyzes, such as the downregulation of miRNA clusters (particularly in males, Supplementary Fig. 10), ECM components, and oligodendrocyte and myelin markers. Overall, the integration of multiomic data revealed important sex-specific molecular networks of ALS. This unbiased data integration strategy highlighted known ALS biomarkers (neurofilaments and chitinases) and the MAPK pathway, especially in females, as important molecular hubs (Supplementary Figs. 10–12, Supplementary Data 14).

## Murine models of ALS partially recapitulate human ALS

We used four mouse models representing the most frequent disease-causing mutations, C9orf72, SOD1, TDP-43, and FUS and performed sex-specific analyzes for male and female animals. Strikingly, the C9orf72 model exhibited the most pronounced transcriptomic

**Fig. 2 | Validation of sex-specific dysregulation in ALS patients in micro-RNAomic, proteomic and multiomic analyzes. a, b** Volcano plot of microRNA analyzes of human samples, separated by sex for mature (**a**) and hairpin (**b**) microRNAs. We used DESEq2 for DE analysis and Benjamini-Hochberg for multiple test correction. The *x*-axis shows the log2 fold change in ALS vs CTR, whereas the *y*-axis shows the -log10 p-value. Orange and blue dots represent DEGs in females and males, respectively. miRNA-mediated regulation in ALS reveals a notable sex-dependent pattern. Male ALS patients exhibit a more pronounced downregulation of mature and hairpin miRNAs compared to females. **c** Heatmap showing mature miRNA expression changes for the identified ALS clusters. miRNA candidates significant in at least one condition were considered (p < 0.05). The scale (left-hand side) shows the range of log2 fold change values. **d** Volcano plot of differentially expressed proteins (DEP) in human samples (calculated with limma, two-sided test; multiple test correction with Benjamini-Hochberg). The *x*-axis shows the log2 fold change ALS and CTR, whereas the *y*-axis shows the -log10 p-values. Orange and blue dots represent DEGs in females and males, respectively. 379 DEPs in males and 251

in females. ANXA2 emerges as the sole protein downregulated in both sexes (p < 0.1). **e, f** Functional enrichment and unsupervised clustering using REVIGO, a tool used for summarizing Gene Ontology (GO) terms. Important pathway clusters are revealed in both sexes, including synaptic function, immune response, and ECM/cytoskeleton. Females (**e**) exhibited enrichment in transmembrane transport, lipid metabolism, development, catalytic activity, and ERK1/2 signaling, while males (**f**) showed enrichment in cell metabolism and tyrosine kinase-related pathways. The size of the circles represents the number of genes in the GO-BP terms, and the color of each node represents the enrichment FDR values. **g, h** Uniform manifold approximation and projection (UMAP) of MOFA factor analysis for sex (**g**) and condition (ALS vs. CTR) (**h**). Sex (**g**) emerges as a robust differentiating factor when integrating transcriptomic, small RNA, and proteomic data. **i** MOFA correlation analysis displays which factors are dominated by which omic layer. Components of the representative factors 3 (males) and 12 (females) are displayed in feature-weight plots (bottom part). These factors are the ones that better correlate with disease conditions for each sex.

changes, mirroring the rapid disease progression observed in this model (Fig. 3a, Supplementary Fig. 3). In each mouse model, we found differentially regulated pathways that were also identified in the human cohort: The C9orf72 model prominently featured alterations in immune and inflammatory response pathways, the SOD1 model in the ERK1/2 cascade, development, and response to oxidative stress, whereas transcription and endopeptidase activity were altered in the TDP-43 model. In contrast, the FUS model exhibited the least transcriptional deregulation without enrichment of pathways (Fig. 3b, Supplementary Data 15).

Further, transcriptome-based analyzes were performed to explore DAS in our mouse models (Supplementary Fig. 13, Supplementary Data 6). It is well documented that TARDBP/TDP-43 and FUS regulate alternative splicing and transcript usage in hundreds of genes[28,29]. TDP-43 also represses cryptic exon-splicing events in *STMN2*[30] and *UNC13A*[31]. We were not able to detect any significant cryptic splicing events for these two genes in our data. However, it is important to note that such cryptic exon inclusion events are specific to neurons with TDP-43 pathology[32]. They are also not conserved in mice[31] and are rarely detectable in bulk RNA-Seq at our sequencing depth[33]. Despite the absence of notable cryptic splicing events, our analysis revealed differential splicing in important genes such as *FLNB, CPLANE1* (involved in ciliogenesis and migration), and *ATP1B1* (a membrane-bound Na⁺/K⁺-ATPase) in multiple mouse models (Supplementary Data 6). DAS analyses provided additional insights into sex-specific variations for the mouse cohorts (Supplementary Fig. 13) showing enrichment in the terms mitochondria and myelin sheath (C9orf72), GTPase activity and myelin sheath (SOD1), DNA binding and heat shock protein binding (TDP-43), and protein binding and ribosome (FUS) (Supplementary Data 5).

To assess if the observed alterations in gene expression and protein abundance reflect changes in the cellular composition of the PFC, we estimated the cell-type fractions for the subclusters of human and mouse ALS models using deep learning-based deconvolution[34] from the gene expression data (Fig. 3c, Supplementary Fig. 14). Interestingly, the human subclusters showed changes in the cell fractions that were partially reflected in the individual mouse models. The human C1 and C2 clusters were well correlated and showed a relative decrease in glial and endothelial cells, and a relative increase in excitatory neurons (Fig. 3c, d). SOD1 animals also displayed a slight reduction in glial cells, while a relative increase in endothelial cells was observed for all mouse models. A modest correlation was observed for the SOD1, TDP43 and FUS mouse models with the cluster C4 (Fig. 3c). The C9orf72 model and human C3 were characterized by a strong increase in glial and endothelial cells and a decrease in excitatory

neurons, respectively, suggesting strong neuroinflammation and neuronal loss. The FUS and TDP-43 models showed intermediate levels of glial and neuronal cells, also assessed by cell type enrichment in WGCNA modules (Fig. 3d, Supplementary Figs. 14 and 15). Thus, the observed transcriptional changes are partially driven by cell composition. Interestingly, our data suggest that the neurovascular unit in ALS[24] may be affected differently in different subclusters of patients with ALS. We then correlated the deconvolved fractions of humans and mice with each other. Overall, our transcriptome analyzes revealed correlations between human clusters and mouse models: C1 and C2 showed the best correlation with the SOD1 model (C1: 0.11 and C2: 0.42), whereas C3 correlated best with the C9orf72 model (0.31) and, to a lesser extent, TDP-43 (0.23) and FUS (0.14) models. Finally, C4 showed a weak correlation with the FUS model (0.14).

To further correlate molecular alterations in mouse models with our human data, we investigated the proteome in PFC samples, where again sex-specific differences were reproduced (Supplementary Data 11, Supplementary Fig. 16). Similar to transcriptomic data, the strongest changes were observed in the C9orf72 model. Sequestosome 1/p62, a product of the ALS-causing gene *SQSTM1*[35], showed the strongest upregulation in C9orf72 males, indicating reduced autophagic flux in a dipeptide accumulation model[36] (Supplementary Fig. 16). SOD1 mice showed one upregulated DEP, exportin-1 (XPO1), a major regulator of nuclear RNA export. XPO1 was also among the significantly regulated proteins in the C9orf72 model (Supplementary Data 11). Unsupervised clustering for enrichment analysis of proteomic mouse data using REVIGO revealed multiple pathway clusters that were also identified in the human data. For example, most models presented clusters or individual terms related to synaptic function. Other enriched pathways include cytoskeleton and morphogenesis (for C9orf72 females, SOD1 and FUS males); mitochondria/cell respiration/cell metabolism (present for C9orf72 females, TDP43 females, FUS males and SOD1 for both sexes). A cluster for the MAPK cascade was captured for SOD1 animals (males) (adjusted p-value = 0.04), which also exhibited enrichment for several terms related to endocytosis and regulation of cellular transport (adjusted p-values < 0.04)(Fig. 3e).

To further compare the representation of human ALS-related proteomic changes in murine models, we performed a comparative clustering analysis of Gene Ontology (GO) term in semantic space[37]. Human proteomics results showed prominent clustering for differentiation and development (females: groups 3, 4, 7, 11 and 13; males: group 8), synapse (females: groups 16; males: group 2, 10, 15), and immune/defense response (females: group 12; males: groups 4, 6 and 11) (Supplementary Fig. 17). In contrast, the C9orf72 model showed clustering for RNA processing, ribosome, translation, ATP synthesis,

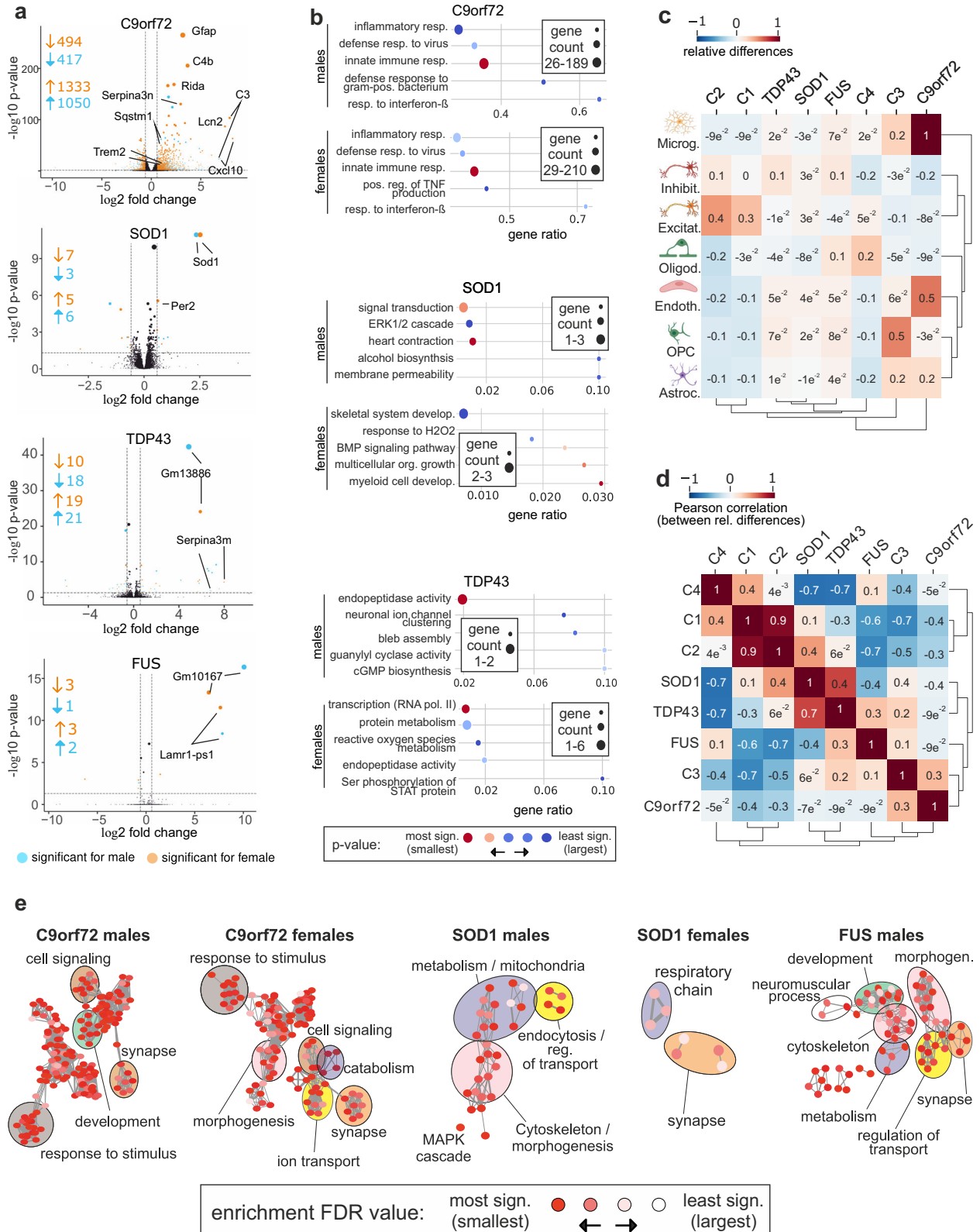

**a** ... **b** ... **c** ... **d** ... **e**

C9orf72 males — cell signaling, synapse, development, response to stimulus

C9orf72 females — response to stimulus, cell signaling, catabolism, synapse, ion transport, morphogenesis

SOD1 males — metabolism / mitochondria, endocytosis / reg. of transport, Cytoskeleton / morphogenesis, MAPK cascade

SOD1 females — respiratory chain, synapse

FUS males — development, neuromuscular process, morphogen., cytoskeleton, metabolism, synapse, regulation of transport

enrichment FDR value: most sign. (smallest) — least sign. (largest)

development, cell adhesion, transport, and synapse. The SOD1 model showed strong clustering for ATP synthesis, mitochondrial respiration, translation, and vesicle-mediated transport. Thus, enrichment results underscored the pathways previously identified in the RNA sequencing data.

In addition, multiple mouse omics datasets, including phospho-proteomics, were also integrated to visualize valid interacting partners based on their expression levels in so-called quadruplets. This analysis identified a coherent regulation of GFAP, SQSTM1, ATXN10 (Supplementary Fig. 18, Supplementary Data 13) and XPO1 (Supplementary Data 13). Several different phosphosites of SQSTM1 were also found significantly upregulated in the phosphoproteomic analysis of C9orf72 mice (Supplementary Fig. 18).

**Fig. 3 | ALS mouse models partially resemble human ALS subclusters. a** Volcano plots of DE genes for male and female mice across ALS models (C9orf72, SOD1, TDP-43, and FUS). The *x*-axis illustrates the log2 fold change between ALS and control (CTR), while the *y*-axis displays the -log10 adjusted p-values. Orange and blue dots denote differentially expressed genes in females and males, respectively. DESEq2 was used for DE analysis and Benjamini-Hochberg for multiple test correction. **b** Analysis of enriched Gene Ontology (GO) terms using topGO in male and female samples. The dot plot displays significantly enriched pathways in males and females, represented by circles colored by their corresponding adjusted p-values (-log10 transformed) on the x-axis and gene count on the y-axis. The size of each circle corresponds to the number of genes annotated in the GO gene set. **c** Heatmap showing the differences (ALS vs. ctrl) in the estimated cell-type fractions (microglia, inhibitory neurons, excitatory neurons, oligodendrocytes, endothelial cells, oligodendrocyte precursor cells [OPC], astrocytes) for human subclusters and

mouse ALS models. Relative differences in cell type abundances are depicted. **d** Pearson Correlation analysis of deconvolved cell-type fraction changes (as shown in c) between human and mouse ALS models. Correlation between relative differences are shown. Subgroups C1 and C2 demonstrated the strongest correlation with the SOD1 model, while C3 exhibited the best correlation with the C9orf72 model, with additional correlations with the TDP-43 and FUS models. A weak correlation was noted between C4 and the FUS model. **e** REVIGO-based summary of proteomics gene set enrichment results for the mouse models. The plots summarize the functional similarity for each model by reducing redundant GO-BP (biological process) terms and clustering the remaining non-redundant terms. Each cluster represents a network of similar GO-BP terms. Networks with 5 or more nodes for each model were selected for display. The size of the circles represents the number of genes in the GO-BP terms, and the color of each node represents the enrichment FDR values.

## Alterations of the MAPK/ERK pathway are identified across multiple omics layers in mouse models and human patients

Finally, we wanted to use the multiomic data generated from humans and mice to identify signaling pathways that could represent potential therapeutic targets for ALS. Although multiple molecular pathways identified in our analysis merit therapeutic validation, we decided to focus on the MAPK/ERK pathway, which was altered consistently across several data types, integration methods, and different subclusters. For example, ERK1/2 signaling was enriched in human females, whereas cell metabolism and tyrosine kinase pathways were enriched in males (Fig. 2e–f, Supplementary Data 12). Integrative analyzes performed with MOFA also identified different interactors of the MAPK/ERK pathway: PEA15 and isoforms of miR-1273 were important contributors to the weight of different MOFA factors for both males and females and directly regulate MAPK signaling (Supplementary Figs. 10 and 12, Supplementary Data 9 and 14). Enrichment analyzes with transcriptomics data showed disruptions in the ERK1/2 cascade for the SOD1 mouse model and DAS results for human males showed enrichment for terms related to the ERK1/ERK2 cascade and the MAPK cascade (Supplementary Data 15). To explore alterations of the MAPK pathway in more depth, we analyzed mouse and human transcriptomics with WGCNA, showing the contribution of MAPK signaling in multiple modules. While deregulations in the MAPK cascade are observed for all mouse models and the human cohort (both for significant levels and absolute counts), the C9orf72 model presented the most striking changes (Fig. 4a). Next, based on the DAS analyzes for human and mouse cohorts, we calculated enrichment analyzes across all species and sexes and grouped the top resulting pathways in five major groups (Fig. 4b). Among them, MAPK clearly shows a sex-specific expression in SOD1 males—corroborating the enrichment results from transcriptomics analyzes for this model, while human males showed enrichment for more general terms related to kinase-activity. We also carried out a miRNA expression analysis, which identified miRNA-451a consistently deregulated across mouse models and human samples with ALS (Fig. 4c, Supplementary Fig. 19). Target prediction analyzes revealed that miRNA-451a interacts with multiple members of the MAPK pathway (Fig. 4d), including *MAPK1, AKT1, and BCL2*, which are known for their roles in the regulation of cell growth, survival, and apoptosis[38]. In addition, centrality measurements for targets of this miRNA showed *MAPK1* as the top hit, followed by *AKT1, IKBKB, BCL2* and *MYC*. This indicates an important role of miR-451a in the context of MAPK signaling (Fig. 4e).

We next investigated the role of the MAPK pathway in human subclusters C1-C4. Interestingly, the classical MAPK pathway was activated in C1 and C2 and downregulated in C3. Meanwhile, C1 and C2 showed downregulation of the JNK and p38 MAPK pathways, which were activated in C3 (Supplementary Fig. 20). In addition to individual omics, also MOFA results from the ALS-associated factor 12 pointed towards a prominent role of the MAPK pathway, where multiple of its members contributed to the PPI network (Fig. 4f). Overall, many of the

analyzes pointed towards a significant role of the MAPK pathway in ALS, specifically ERK1/2, across organisms and data types.

## Validation of MEK2 inhibition by trametinib in vitro and in vivo

Currently, licensed drugs target glutamatergic synapse function (riluzole), oxidative stress (edaravone), mitochondrial function (tauroursodeoxycholic acid/phenylbutyrate), or SOD1 itself (tofersen). Other mechanisms identified in this analysis, such as immune response, ECM/BCNSB function, or the MAPK/ERK pathway are not yet targeted by licensed drugs. We concentrated on mitogen-activated protein kinase kinase 2 (MAP2K2 or MEK2), as it appeared to be upregulated in the human PFC and multiple mouse models. Furthermore, MEK2 can be modulated by the clinically approved inhibitor, trametinib[39]. A scheme for its mechanism of action and the expected effects in the context of the MAPK pathway is depicted in Fig. 4g.

First, we used primary cortical neuronal cultures from P0–P1 C57BL/6 J mice that were treated with glutamate as an in vitro model of excitotoxicity in ALS[40]. Treatment with 5 mM glutamate (6 hours) increased cell death (caspase-positive neurons) and reduced the average neurite length, both of which were counteracted by trametinib (Fig. 5a–d). Glutamate treatment did not affect total MEK2 or ERK1/2 protein expression, but significantly increased pErk1/2 levels, whereas trametinib attenuates glutamate-induced phosphorylation of Erk1/2 (Fig. 5e–f, Supplementary Fig. 21). We observed a significant increase of pMEK2 activation in response to trametinib treatment. This could be explained by inhibition of the feedback circuit: MEK2 inhibition induces a decrease of pERK1/2, which also counteracts the feedback loop, resulting in the activation of the pathway and the accumulation of activated pMEK2[41,42]. (Figs. 4g, 5e, f, Supplementary Figs. 21 and 22). Our data suggest that trametinib attenuates MEK2 activity and reduces Erk1/2 phosphorylation resulting in decreased cell death and increased neurite outgrowth under excitotoxic stress.

To validate the importance of the MAPK pathway in vivo, we selected the SOD1 mouse model, which showed the strongest similarity with the largest human ALS subcluster (C1 and C2) (Fig. 3c–d). In females, we observed pMEK2 levels substantially increased with disease progression, whereas in males, pMEK2 levels returned to control levels after week 14 (Fig. 6a). Furthermore, in an initial study, we treated SOD1 mice with trametinib for seven weeks, starting from week 9 (presymptomatic stage), and observed a reduction in ERK1/2 phosphorylation compared with that in vehicle-treated female and male mice (Fig. 6b). Trametinib significantly reduced the autophagy receptor p62 expression in the spinal cord of female but not male mice, correlating with its previously described neuroprotective role of increasing autophagy via transcription factor EB activation[43] (Fig. 6c). p62 also co-localizes with ubiquitin and mutant SOD1 in protein aggregates[44]. Accordingly, we detected a significant reduction in detergent-insoluble SOD1 and ubiquitin in trametinib-treated females but not in males (Fig. 6d, e). Finally, we investigated the effect of trametinib on neurodegeneration and muscle denervation. NfL plasma

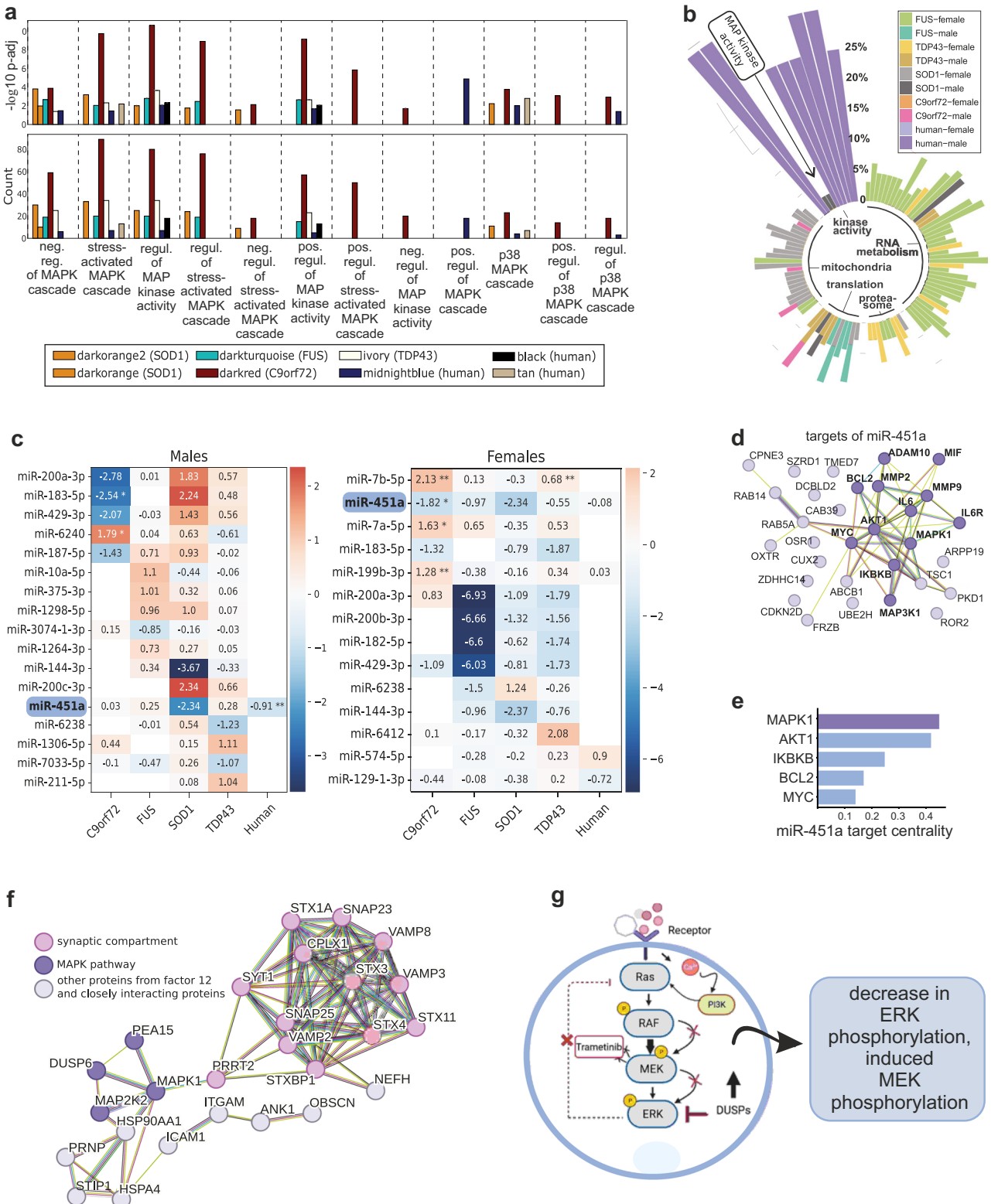

concentration[45] and Acetylcholine Receptor γ subunit (AChRγ) expression[46] were significantly reduced after trametinib treatment in female SOD1 mice (Fig. 6f, Supplementary Fig. 24). We also observed strong MEK2 phosphorylation in the motor neurons of the spinal cord, the main tissue involved in the pathology of this animal model[47] (Fig. 6g–h). Phospho-proteomic analyzes supported the regulation of the MAPK pathway by trametinib in females, demonstrating in addition

a reduced phosphorylation of its upstream regulator CAMK2A (Supplementary Fig. 23).

We thus focused on female mice to assess the effect of trametinib on the onset and progression of the disease. In a follow-up study, a second group of female SOD1 mice was treated with the same schedule up to survival. Body weight monitoring and motor tests (grip strength, extension reflex) were performed in comparison with vehicle-treated

**Fig. 4 | Consistent dysregulation of the MAPK pathway in human ALS patients and mouse models. a** The occurrence and importance of MAPK pathways and other related kinase pathways are shown across all mouse models and the human samples, highlighting distinct activities within the co-expression modules identified by WGCNA. For that, we selected gene modules from individual WGCNA analyzes for mice and humans by filtering for terms with MAPK/MAP kinase. The upper panel shows the significance (-log10 p-value; right-tailed fisher's exact test with Benjamini-Hochberg correction) and the lower panel shows absolute counts. The legend below the bars depicts the origin of the hits. **b** GO-enrichment results for DAS genes. Bar heights represent the fraction of Gene Ratio of differentially alternatively spliced genes in the pathways. All pathways have an adjusted p-value < 0.1. **c** Differential expression analyzes for mature miRNAs for humans and ALS mouse models. Mice exhibited pronounced differential expression (DE) of miRNAs, with the C9orf72 model showing the most significant changes. **d** Targets of miR-451a, such as MAPK1, AKT1, BCL2, Il6R, IKBKB, MIF, MMP2, and MMP9, are involved in cell growth, survival, apoptosis, inflammatory signaling, and ECM

remodeling. **e** Target centrality measurement for miR-451a reveals MAPK1 as the top hit for target centrality measures based on the network shown in panel (**d**). **f** Protein-protein interaction (PPI) network of the genes in MOFA factor 12 in females. MAPK1 is an important molecular hub and interacts with PEA15, PRRT2, MEK2, DUSP6, HSPA4 and HSP90AA1. Further proteins of interest are highlighted (bold/dark purple). **g** Scheme for the mechanism of action of trametinib in the context of the MAPK pathway. MEK/Erk are activated through the Ras/RAF/PI3K cascade, showing important roles for cell proliferation/survival. ERK activation is subject to negative feedback regulation both downstream and upstream of MEK. This involves the expression of DUSPs, as well as the direct phosphorylation and inhibition of proteins such as RAF. Trametinib binds to the activation loop of MEK, disrupting the Raf-dependent phosphorylation of the target. This results in reduced expression of p-MEK and p-Erk. The pharmacological inhibition of MEK by trametinib swiftly eliminates this feedback mechanism, inducing the phosphorylation of MEK. Panel **g** was created with BioRender.com released under a Creative Commons Attribution-NonCommercial-NoDerivs 4.0 International license.

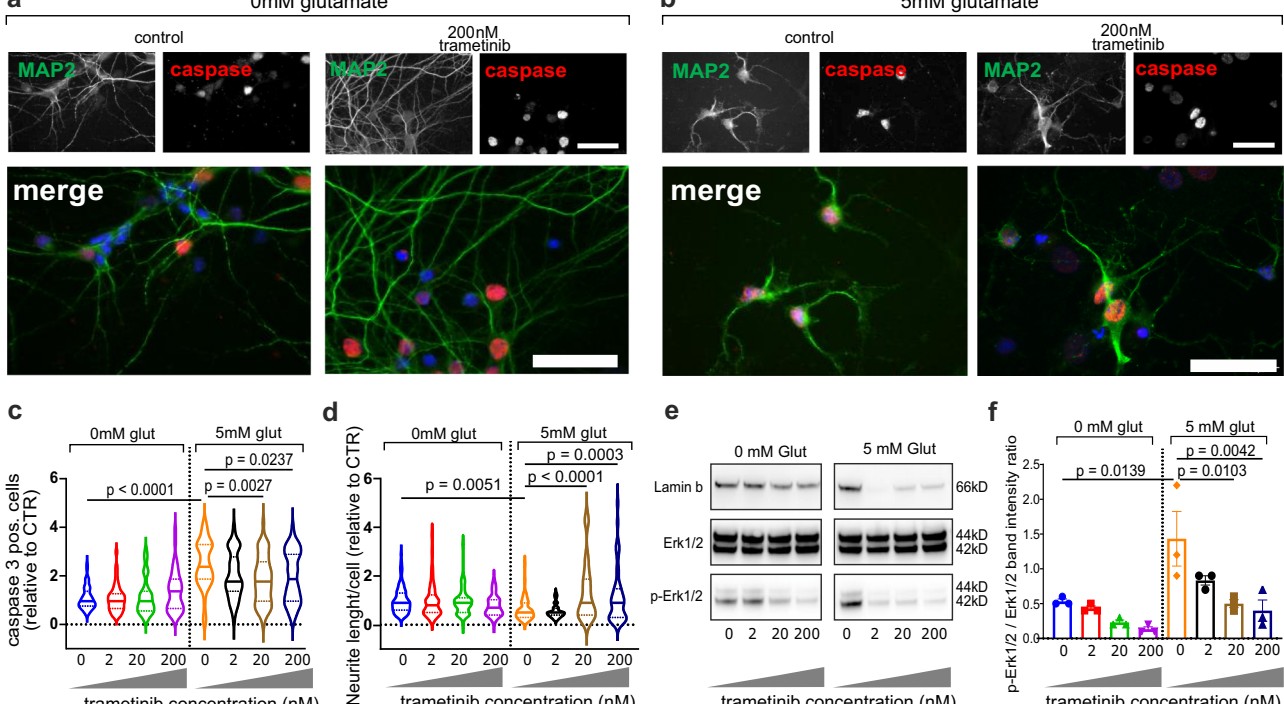

**Fig. 5 | Effects of the MEK2-inhibitor trametinib on the MAPK pathway in vitro. a**, **b** Effects of different concentrations of trametinib on apoptosis in glutamate (5 mM)- and non-glutamate-treated cells analyzed by immunostaining. Representative photomicrographs for two of the analyzed conditions (control [vehicle] and 200 nM trametinib). Scale bar: 40 μm. **c**, **d** Quantification plots showing the effects of treatment with trametinib on cell survival (caspase 3 staining) (**c**) and neurite outgrowth (**d**) for glutamate- or vehicle-treated (control) cells, for all analyzed conditions (control [vehicle]; 2 nM trametinib; 20 nM trametinib; 200 nM trametinib). Data represent the mean ± standard error of the mean (SEM) (n = 5 independent neuronal cultures), and were tested using pairwise Tukey's test after one-way analysis of variance. In (**c**), p < 0.0001 for 0 mM glutamate − 0 nM trametinib vs. 5 mM glutamate − 0 nM trametinib conditions; p = 0.0027 for 5 mM glutamate − 0 nM trametinib vs. 5 mM glutamate − 20 nM trametinib conditions; p = 0.0237 for 5 mM glutamate − 0 nM trametinib vs. 5 mM glutamate − 200 nM

trametinib conditions. In (**d**), p = 0.0051 for 0 mM glutamate − 0 nM trametinib vs. 5 mM glutamate − 0 nM trametinib conditions; p < 0.0001 for 5 mM glutamate − 0 nM trametinib vs. 5 mM-glutamate − 20 nM trametinib conditions; p = 0.0003 for 5 mM glutamate − 0 nM trametinib vs. 5 mM glutamate − 200 nM trametinib conditions. Non-significant comparisons not depicted in the panels. **e**, **f** Western blot analysis and quantification of trametinib effects on pErk1/2 with and without glutamate treatment. Data represent the mean ± SEM (n = 3 different cultures) and were tested using pairwise Tukey's test after one-way analysis of variance. In (**f**), p = 0.0139 for 0 mM glutamate − 0 nM trametinib vs. 5 mM glutamate − 0 nM trametinib conditions; p = 0.0103 for 5 mM glutamate − 0 nM trametinib vs. 5 mM glutamate − 20 nM trametinib conditions; p = 0.0042 for 5 mM glutamate − 0 nM trametinib vs. 5 mM glutamate − 200 nM trametinib conditions. Non-significant comparisons not depicted in the panels.

mice. Although no effect was observed in body weight loss and behavioral analysis (available as source data files), there was a significant delay in the age of paralysis (Fig. 6i), the onset of the disease, and significantly improved survival (Fig. 6j). We also observed that a significant effect on NfL plasma concentration was maintained along the progression of the disease (Fig. 6k). Thus, trametinib markedly affects the clearance of protein aggregates, leading to neuroprotection

in female SOD1 mice, suggesting that MEK2 is a promising therapeutic target for ALS, particularly in females.

## Discussion
In this study, we conducted individual and combined analyzes of multiple omics datasets to comprehensively understand the molecular architecture of ALS in the PFC, an area affected in the later

 

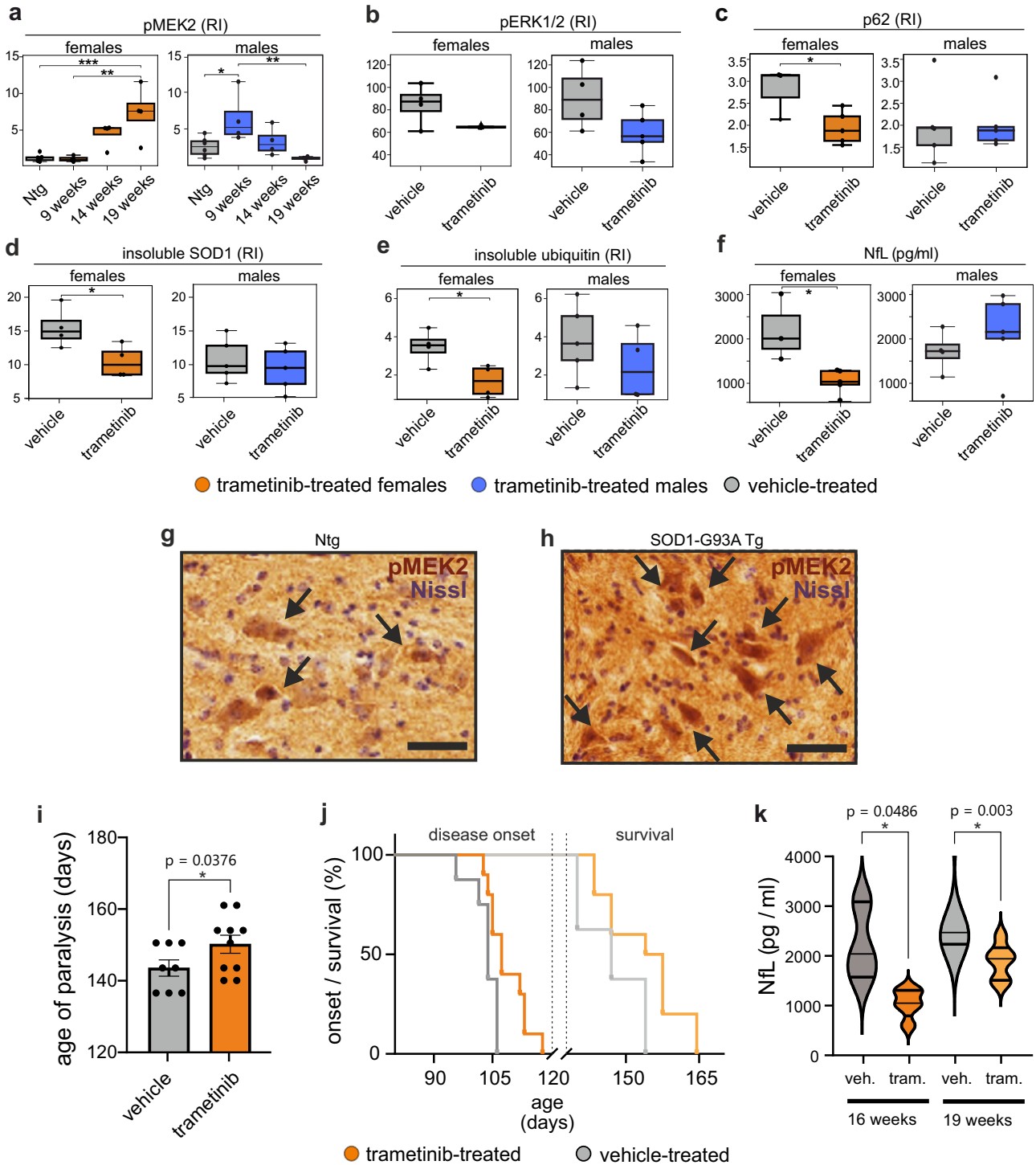

stages by TDP-43 pathology and that has the potential to reveal early disease mechanisms[6]. We observed that significant sex-specific differences are often more pronounced in males, which is aligned with the higher prevalence of ALS in males[1]. Previous studies have also identified sex-specific differences in the blood of patients with ALS[48,49] as well as therapeutic responses in ALS mouse models[50,51]. However, current therapeutic options, clinical trials, and Food and Drug Administration guidelines for clinical trial design[41] do not consider patient sex in differential therapy. Our data, especially the sex-specific differences in MAPK signaling and therapeutic tractability, suggest that sex should be considered as a covariate in future ALS clinical trials.

Phenotypic heterogeneity is clearly recognized in ALS[52], as reflected by stratification in onset or disease progression rate, and previous transcriptomic analyzes have suggested molecular subtypes[13]. We identified four clusters in our human ALS cohort based on the transcriptome, partially mirroring previously identified ALS-Ox, ALS-Glia, and ALS-Transposable elements subtypes[9]. While the ALS-Glia and ALS-Ox clusters correlated with clusters C1 and C2 and C3 and C4, respectively, the ALS-TE cluster was only marginally represented in our data. As we analyzed the PFC, an area wherein TDP-43 aggregation is observed later than that in the motor cortex, our data reinforce that ALS-TE is driven by TDP-43 dysfunction[9]. Because alterations in the PFC may represent earlier changes in ALS, clustering differences

**Fig. 6 | Modulation of the MAPK pathway member MEK2 by trametinib attenuates ALS pathology in vivo. a** Western blot analysis of pMEK2 in lumbar spinal cords of SOD1 transgenic (tg) and non-transgenic (Ntg) mice at 9, 14, and 19 weeks. Data are shown as box plots (Ntg, n = 6; SOD1, n = 4 per group) with median center lines, 0.25-0.75 interquartile range boxes, and 1.5x IQR whiskers. Results are expressed as relative immunoreactivity (RI). *p < 0.05, pairwise Tukey's test after one-way ANOVA. **b–f** Results of the study in female and male SOD1 mice un/treated with trametinib (9 to 16 weeks). Data are presented as box plots with center line on median, box bounds indicating the 0.25-0.75 IQR and whiskers at 1.5 times the IQR below the first and above third quartile. Dot blot analysis of pERK1/2 (female, n = 4 per group; male, n = 4 vehicle-treated mice; n = 5 trametinib-treated mice) (**b**), p62 (female, n = 3 vehicle-treated mice; n = 5 trametinib-treated mice; male, n = 5 per group) (**c**), insoluble SOD1 (female, n = 4 per group; male, n = 5 vehicle-treated mice; n = 4 trametinib-treated mice) (**d**), and ubiquitin (female, n = 4 per group; male, n = 5 per group) (**e**) in the spinal cord of SOD1 female and male mice treated with trametinib or vehicle. Data are mean ± SEM and are expressed as relative immunoreactivity (RI). *p < 0.05, Student's t-test. Plasma neurofilament light chain levels were analyzed in female and male (**f**) SOD1 mice treated or not with trametinib. Data are mean ± SEM (female, n = 3 vehicle-treated mice and n = 5 trametinib-treated mice; male, n = 4 vehicle-treated mice and n = 5 trametinib-

treated mice). *p < 0.05, two-tailed Student's t-test. **g, h** Diffuse pMEK2 immunostaining in the lumbar spinal cord of non-transgenic (**g**) and SOD1 mice (**h**) at 19 weeks. In the ventral horns, pMEK2 staining was mainly observed in motor neurons. Scale bar: 50 μm. Experiments included three animals per group; three independent slices analyzed per animal. **i–k** In a preclinical study, female SOD1 mice were treated with trametinib from 9 weeks until end of life. Paralysis was assessed by age at loss of reflex, with hind limbs and paws retracted. The mean age differed significantly between trametinib-treated mice (150.2 ± 8 days) and vehicle-treated mice (143.5 ± 6.5 days). Data are mean ± SEM (vehicle n = 8; trametinib n = 10). *p = 0.0376, one-tailed Student's t-test. **j** Kaplan–Meier curve for disease onset and survival of SOD1 female mice treated with vehicle (n = 8) or trametinib (n = 10). Log–rank Mantel–Cox test for disease onset, *p = 0.0177 (mean ± SD = vehicle 103.4 ± 3.5 days and trametinib 108.7 ± 5 days). Log–rank Mantel–Cox test for survival, *p = 0.0405 (mean ± SD = vehicle 147 ± 6.5 days and trametinib 153.7 ± 8 days). **k** Plasma NfL levels were analyzed in SOD1 female mice, with and without trametinib treatment, at 16 and 19 weeks. Data are mean ± SEM (16 weeks: vehicle n = 3, trametinib n = 5; 19 weeks: vehicle n = 5, trametinib n = 6). At 16 weeks: p = 0.0486 for vehicle-treated animals vs. trametinib-treated animals; at 19 weeks: p = 0.003 for vehicle-treated animals vs. trametinib-treated animals (Benjamini-Hochberg corrected p-values after one-way ANOVA).

---

between the two studies may also reflect the evolution of clusters in different stages of the disease. Molecular subtypes of ALS may evolve with time and could be subject to changes during the disease course. Exploring liquid biomarkers could further aid in patient stratification and personalized clinical trials.

Furthermore, we suggest that the molecular phenotype in the four analyzed mouse models can be partially approximated to features displayed in the ALS cohort and the different subclusters, providing evidence for the validity of these models in recapitulating parts of the disease in humans. For example, the oldest and most frequently used model, the SOD1.G93A mouse, displayed several common features present in the largest clusters C1 and C2. For example, we observed a relative decrease in different types of glial cells, as well as in molecular pathways (e.g. synaptic- and mitochondria-related terms captured in clustering analysis for humans and REVIGO analysis for the mouse model, for instance). Overall, although not representative of all human ALS cases, our findings indicate that this model represents an important subgroup of the disease based on the identified dysregulated pathways.

Integrated analysis of multiomic datasets has identified several dysregulated pathways relevant for ALS, including mitochondrial respiration/oxidative stress, transcriptional regulation/splicing, and protein misfolding/degradation[3]. Additionally, previously less prioritized pathways—including the dysregulation of the ECM and BCNSB or the MAPK pathway—were identified. The deregulation of pathways that are only distantly related suggests the need for combinatorial drug therapies to address ALS mechanistically. It is important to highlight that we found evidence of female-predominant deregulation of the MAPK pathway in integrative analyzes, but it did not contribute to the separation of clusters C1–C4. Therefore, the MAPK pathway may be an interesting therapeutic target for all human ALS subgroups. MAPKs are fundamental signal transducers that are involved in cell proliferation, differentiation, survival, and death, and as such not specific for ALS[53]. Extracellular and intracellular signals are integrated by MAPK, and the overactivation of MAPK signaling, for example, via the abnormal phosphorylation of ERK1/2, has been reported in human and ALS mouse models[54]. A Ser/Thr kinase belonging to the MAPK superfamily—the MAPK/MAK/MRK [MOK] kinase—has also been linked to impaired microglial function and inflammatory insults associated with ALS pathophysiology[55]. Increased phosphorylation of ERK1/2 was also observed in our glutamate toxicity model, and pMEK2 expression was increased in the SOD1 mouse model, reproducing the aberrant activation of this pathway that was

restored by the MEK2-inhibitor trametinib. However, this effect was most pronounced in females, suggesting a sex-specific efficacy. Our experimental approach did not target neurons exclusively. Nevertheless, our primary cultures mainly contained neurons (~95%), and all staining procedures for both in vivo and in vitro experiments focused on neurons. While other cell types may be involved, the evidence presented here supports the significance of the MEK2–MAPK pathway in neurons.

Indeed, trametinib significantly affected the onset of the disease, the age of paralysis and the survival of SOD1 mice, but not the rate of disease progression, in which non-neuronal supporting cells are thought to have a role[56]. Although the MAPK pathway may represent a less specific answer of the cell in response to stressors, its involvement in ALS pathomechanisms suggests it could be a valuable target for potential combination therapies. Currently, one phase I/II clinical trial is examining the safety, tolerability, and efficacy of trametinib in patients with ALS (clinicaltrials.gov identifier:NCT04326283). Our data support the independent evaluation of male and female patients and suggest that the ECM, immune response, and RNA processing machinery are potential targets for therapeutic intervention that need further exploration.

Despite the limitations of our study (e.g., the retrospective view permitted by postmortem tissue analysis and restricted balancing of sex and disease stage due to the limited number of well-characterized postmortem brain samples), we aimed to provide valuable insights into the molecular architecture of ALS. While analyzing over 100 brain samples helped identify molecular subclusters, a larger sample size may provide a more detailed view of the subgroups. Nevertheless, analyzing the PFC tissue allowed us to study the earlier stages of the disease, offering a unique perspective compared with those of previous studies. Our robust multiomic, computational, and integrative approaches identified subclusters that are reminiscent of previously identified subtypes, yet show clear differences. Additionally, we emphasize the role of splicing and transcript usage in early disease regulation, involving several ALS-related genes. Overall, our study contributes towards unraveling the complexity of ALS and lays the groundwork for further research in the field.

Our study also compared human brain tissue with samples from transgenic mouse models of ALS. Although clear correlations were found between mouse models and molecular subtypes of sALS, the four analyzed models represent specific scenarios. Other models for each of the four studied genes could potentially yield different results[57]. Furthermore, we did not analyze DNA methylation status or other post-translational modifications (such as glycation, methylation,

or acetylation) that could provide further insights into ALS-specific dysregulation, as much as a single cell-based analysis could yield an additional layer of detail.

In this thorough integration of human tissue- and animal model-derived multiomic data, we emphasized sex-specific differences in ALS pathology, identified molecular clusters, and highlighted their importance for clinical trials and the development of therapeutic strategies for this disease. The validation of omics results relies on multiple systems, as each mouse model system only reproduces parts of the human pathology. Our data suggest the need for the validation of additional molecules and pathways and justify the further exploration of the MAPK pathway as an ALS therapeutic target.

## Methods

The results presented here comply with all relevant ethical regulations. Ethical approval for the use of human tissues was obtained from the Ethics Committees of the University Medical Center Göttingen (2/8/18 AN) and the Technical University Munich (145/19 S-SR).

All animal experiments complied with international and local animal welfare standards and were approved by the respective regulatory organs of the involved research centers. The experimental protocols for the collection of PFC tissue from transgenic SOD1 and FUS mice were prospectively reviewed and approved by the Mario Negri Institutional Animal Care and Use Committee and the Italian Ministry of Health (Prot. No. 9F5F5.143, ministry authorization no. 4/2020-PR; *Direzione Generale della Sanità Animale e dei Farmaci Veterinari, Ufficio 6*). The collection of PFC tissue from C9orf72 transgenic mice followed the regulations from the German Animal Welfare Act (*Tierschutzgesetz/Tierschutz-Versuchstierverordnung, Regierungsbezirke Oberbayern*, Prot. No. TV 55.2-2532.Vet_02-17-106). The collection of PFC tissue from TDP43 transgenic animals was approved by the (CCD) *Centrale Commissie Dierproeven* of Utrecht University (CCD license: AVD 1150020171565) and was in accordance with Dutch animal welfare laws (*Wet op de Dierproeven 2014*) and European regulations (guideline 2010/63/EU). The validation experiments in SOD1 transgenic mice, including the treatment with trametinib, were reviewed and approved by the Mario Negri Institutional Animal Care and Use Committee and the Italian Ministry of Health (Prot. No. 9F5F5.250, ministry authorization no. 496/2023; *Direzione Generale della Sanità Animale e dei Farmaci Veterinari, Ufficio 6*).

### Human postmortem PFC samples

Human PFC samples were provided by four different brain banks: The Netherlands Brain Bank (NBB), London Neurodegenerative Diseases Brain Bank (London NDBB), Imperial College London−Multiple Sclerosis and Parkinson's Tissue Bank (ICL MS&PD TB), and the Oxford Brain Bank (OBB). Race, ethnicity or social information were not considered for cohort composition. In total, 51 ALS and 50 CTR samples (without any signs of neurodegenerative diseases) were included. Frozen tissues were shipped on dry ice to the rechts der Isar Hospital Department of Neurology at the Technical University of Munich and stored at −80 °C (Supplementary Data 1). For sampling, PFC samples were transferred to a cryostat chamber at −20 °C and punched with a 20-G Quincke Spinal Needle (Becton Dickinson, Franklin Lakes, NJ, USA). Subsequently, ~20 mg tissue was collected in RNASe-/DNAse-free tubes and stored at −80 °C until further use.

### ALS animal models

Four transgenic mouse models representing the most frequent ALS-causing genes were used for the multiomic studies. Mice were housed in standard cages in pathogen-free facilities, under a 12-h light/dark cycle. The animals were provided with unrestricted access to food and water. Prior to euthanasia for tissue collection, the animals were fully anesthetized in a CO2 chamber. After spinal reflexes were tested (by pinching the tail and upper and lower limbs with a laboratory tweezer),

the animals were subsequently euthanized by cervical dislocation. B6;129S6-Gt(ROSA)26Sortm1(TARDBP*M337V/Ypet)Tlbt/J mice[58] (here simply referred to as TDP-43 mice) were provided by the Department of Translational Neuroscience of the University Medical Center Utrecht. This model was generated by inserting an 80 kb genomic fragment carrying the human TDP-43 locus (including a patient-derived M337V mutation). TDP-43 transgenic and control wild-type mice were euthanized at the age of 26 weeks (presymptomatic stage) for biomaterial collection. B6SJL-Tg(SOD1*G93A)1Gur/J mice[47] (here referred to as SOD1 mice) were provided by the Laboratory of Translational Biomarkers, Mario Negri Institute for Pharmacological Research, Milan. High-copy number B6 congenic Tg(SOD1*G93A)1Gur/J SOD1*G93A male mice from The Jackson Laboratory (Bar Harbor, ME, USA) were bred with C57BL/6 female mice to obtain non-transgenic and mutant transgenic G93A*SOD1-expressing mice. SOD1 transgenic and control animals were sacrificed 14 weeks after birth (presymptomatic stage). (Poly)GA-NES/C9orf72(R26(CAG-Isl-175GA)−29×Nes-Cre) mice ref. [59] (here referred to as C9orf72 mice) were provided by the German Center for Neurodegenerative Diseases in Munich. To generate transgenic animals, plasmids were electroporated to enable the conditional expression of dipeptide-repeat proteins (DPRs). DPRs were created by inserting 175 GFP-(GA) genes encoded by non-repeating alternate codons downstream of a floxed stop cassette in the pEX CAG stop-bpA vector. Electroporation was performed on murine recombination-mediated cassette exchange embryonic stem cells at the Rosa26 safe harbor locus. Mouse lines with germline transmission, known as GAstop, were obtained and subsequently backcrossed to a C57BL/6 J background until a confirmed purity of over 98% was achieved (using SNP genotyping). C9orf72 transgenic and control animals were sacrificed 4.5 weeks after birth (early symptomatic stage). Tg (Prnp-FUS)WT3Cshw/J mice[60] (hereafter referred to as FUS mice) were provided by the Laboratory of Translational Biomarkers, Mario Negri Institute for Pharmacological Research, and were sacrificed four weeks after birth. For each model, a total of 20 transgenic and non-transgenic mice were selected and balanced for condition and sex (TDP-43 model: 5 females vs. 5 males for transgenic and control cohorts; SOD1 model: 5 females vs. 6 males for the control cohort and 5 females vs. 4 males for the transgenic cohort; C9orf72 model: 6 females vs. 4 males for the control cohort and 4 females vs. 6 males for the transgenic cohort; FUS model: 5 females vs. 5 males for transgenic and control cohorts).

### Preparation of PFC from ALS mouse models

The mice were perfused with 50 mL of ice-cold phosphate-buffered saline (PBS) prior to microdissection. The head was removed by cutting the base of the skull followed by skin removal. The skull was removed using small incisions followed by microdissection of the PFC in both hemispheres. The olfactory bulb and cerebellum were removed by cutting the cerebellar peduncle starting from the olfactory bulb and continuing along the interhemispheric fissure using tweezers with fine tips. The cortex was removed from the rest of the brain. Incisions were made in the middle of the cortex to excise the PFC (Supplementary Fig. 25). Freshly prepared PFCs were collected in nuclease-free tubes and stored at −80 °C until RNA and protein isolation experiments were performed.

### RNA and DNA isolation from human and mouse tissue samples

Total RNA was isolated from human and animal PFC samples using TRIzol Reagent (Sigma-Aldrich, Taufkirchen, Germany). To ensure proper handling of the RNA, all related experiments were conducted in an RNA workstation fume hood. In brief, 500 µL TRIzol was added to each sample, and the tissues were homogenized using a plastic homogenizer. Subsequently, 50 µL of 1-bromo-3-chloro-propane (Sigma-Aldrich) was added. The reaction tubes were inverted for 10−15 s and incubated at room temperature (22−25 °C) for three

minutes. The resulting lysates were centrifuged at 12 000 × g for 15 min at 4 °C, leading to phase separation. The RNA-containing aqueous phase was carefully collected and transferred to a fresh nuclease-free tube. To precipitate the RNA, 250 μL of 2-propanol (AppliChem, Darmstadt, Germany) and 2 μL of GlycoBlue Coprecipitant (15 mg/mL) (Thermo Fisher Scientific, Waltham, MA, USA) were added to each sample. The mixture was thoroughly vortexed and incubated overnight at −20 °C. Samples were centrifuged at 12 000 × g for 30 min at 4 °C the following day. The supernatant was discarded, and the RNA pellets were washed three times with 75% ice-cold ethanol (AppliChem). After air-drying the pellets for a few minutes under a fume hood, they were reconstituted with 15–20 μL nuclease-free water (Sigma-Aldrich). The reconstituted RNA was completely dissolved by incubation at 55 °C for two minutes in a thermoshaker. Following RNA isolation, DNase treatment was performed to eliminate DNA contamination. For this purpose, 5 μL 10× DNAse I Incubation Buffer (Life Technologies, Carlsbad, CA, United States), 5 μL DNase I (2 U/μL), and 0.5 μL RNaseOUT (40 U/μL) were added to each sample. Nuclease-free water was added to a final volume of 50 μL, and the samples were then incubated at 37 °C for 20 min. Finally, the RNA samples were purified and concentrated using an RNA Clean & Concentrator-5 Kit (Zymo Research, Irvine, CA, USA) following the manufacturer's instructions. A QIAamp DNA Mini Kit (Qiagen, Hilden, Germany) was used for DNA isolation from human midbrain samples according to the manufacturer's instructions. After RNA and DNA isolation, the concentration and purity of nucleic acids were assessed using a NanoDrop One spectrophotometer (Thermo Fisher Scientific). RNA integrity was evaluated using an Agilent 6000 NanoKit on a 2100 Bioanalyzer (Agilent, Santa Clara, CA, USA).

### DNA sequencing experiments and C9orf72 repeat expansion analysis

Prior to the DNA sequencing experiments, the quality of the isolated DNA was determined using a 4200 TapeStation System (Agilent). The AmpliSeq protocol (Illumina, San Diego, CA, USA) was used in subsequent steps. We used a target panel of 566 amplicons covering 30 ALS-related genes: *TARDBP, DCTN1, ALS2, ERBB4, TUBA4A, CHMP2B, NEK1, MATR3, SQSTM1, FIG4, C9orf72, SIGMAR1, VCP, GLE1, SETX, OPTN, HNRNPA1, KIF5A, TBK1, ANG, SPG11, CCNF, FUS, PFN1, MAPT, VAPB, SOD1, CHCHD10, NEFH*, and *UBQLN2*. Using this panel, DNA samples (50–100 ng) were amplified over 14 PCR cycles. The amplicons were digested, and AmpliSeq combinatorial dual indices were ligated for multiplexing. The quality and quantity of the enriched libraries were validated using a 4200 TapeStation System (Agilent). The average fragment size of the amplified product was 480 bp. The libraries were standardized to a concentration of 9 nM in Tris−Cl (10 mM, pH 8.5) supplemented with 0.1% Tween 20. Cluster generation and sequencing were performed using the MiSeq platform (Illumina) following the standard protocol. The loading concentration was 9 pM, and 15% phiX was added. The sequencing configuration was a 250-bp paired-end sequencing. C9orf72 repeat expansion analysis was performed using the AmplideX PCR/CE C9orf72 Kit (Asuragen, Austin, TX, USA). Briefly, DNA samples (n = 51, 40 ng each) were amplified using the three-primer GGGGCC-repeat-primed configuration (combining flanking primers and a GGGGCC-repeat-specific primer). This configuration allows the sizing of GGGGCC alleles of up to 145 repeats and simultaneous detection of expanded GGGGCC alleles with > 145 repeats. The PCR conditions consisted of initial denaturation (five minutes, 98 °C), 37 cycles (35 s, 97 °C; 35 s, 62 °C; and 3 min, 72 °C), and final elongation (10 min, 72 °C). Capillary electrophoresis was carried out on an ABI 3730 DNA Analyzer (Applied Biosystems, Waltham, MA. USA) using a ROX 1000 Size Ladder (Asuragen), followed by analysis with Gene-Mapper 4.0 software (Applied Biosystems) and conversion of peak size to GGGGCC repeat length via the calibration curve method according to the manufacturer's instructions.

### Preparation of RNA libraries for sequencing experiments

The mRNA and small RNA sequencing experiments were conducted at the Functional Genomics Center, Zürich. For mRNA sequencing, two different library preparation kits were used: the TruSeq Stranded mRNA Kit (Illumina; short read sequencing) and the SMARTer Stranded Total RNA-Seq Kit v2 Pico Input Mammalian (Takara Bio USA, San Jose, CA, USA; short read sequencing). In the TruSeq protocol, total RNA samples (100–1000 ng) were subjected to poly-A enrichment and reverse transcription to generate double-stranded cDNA. The cDNA was fragmented, end-repaired, adenylated, and ligated to TruSeq adapters containing unique dual indices for multiplexing. Fragments with adapters at both ends were selectively enriched using PCR amplification, resulting in a smear with an average fragment size of ~260 bp. The libraries were subsequently normalized to 10 nM in Tris−Cl (10 mM, pH 8.5) containing 0.1% Tween 20. For the SMARTer Stranded Total RNA-Seq Kit protocol, total RNA samples (0.25–10 ng) were reverse-transcribed using random priming into double-stranded cDNA in the presence of a template switch oligo (TSO). The generated cDNA fragments contained sequences derived from random priming oligos and the TSO. PCR amplification with primers binding to these sequences incorporated full-length Illumina adapters, including a multiplexing index. The ribosomal cDNA was cleaved using ZapR in the presence of mammalian R-Probes. The remaining fragments were enriched through a second round of PCR amplification using primers designed to match the Illumina adapters, resulting in a smear with an average fragment size of ~360 bp. The libraries were normalized to 5 nM in Tris−Cl (10 mM, pH 8.5) containing 0.1% Tween 20. To validate the quality and quantity of the isolated RNA and enriched libraries, an Agilent Fragment Analyzer was used for the TruSeq kit, while the 4200 TapeStation System (Agilent) was used for the SMARTer Kit.

### RNA sequencing

Sequencing was performed using the Illumina platforms NovaSeq 6000 (for transcriptomics) and HiSeq 2500 (for small RNA sequencing) according to standard protocols. Small RNA sequencing was performed using RealSeq-AC miRNA (SomaGenics, Santa Cruz, CA, USA) (short read sequencing). All samples were quantified and quality was controlled using an Agilent Fragment Analyzer. Briefly, RNA samples (1 ng–1 ug) were adaptor-ligated, circularized, and reverse-transcribed into cDNA. The cDNA samples were amplified using PCR, which also incorporated sample barcodes. The library product, a peak with a fragment size of ~149 bp, was normalized to 10 nM in Tris−Cl (2 mM, pH 8.5) containing 0.1% Tween 20. The quality and quantity of the enriched libraries were validated using an Agilent Fragment Analyzer. Transcriptomics data were processed using the Nextflow Core RNASeq pipeline version 3.0[61]. The data were demultiplexed with bcl2fastq, and the fastq files underwent several quality checks, including FastQC[62]. Salmon[63] was used for pseudo alignment and quantitation with a Salmon index built using GRCm39 with annotations from GENCODE vM26 for mouse data and GRCh38 with annotations from GENCODE v37 for human data. Count matrices from Salmon were used for the downstream analyzes. The count matrices were filtered, retaining genes with at least ten counts in 50% of the samples for any condition or sex. We used the clusterProfiler R package[64] and GO biological processes and molecular functions for GSEA, filtering terms by size between 10 and 500 genes and correcting for multiple testing (Benjamini−Hochberg correction). For the DE analysis we used a cut-off of adjusted p-value 0.05 and threshold of log2 fold change (±1.5).

### Global proteomics of mouse and human PFC tissue samples

Tissues from both human PFCs and four different mouse models were prepared. The tissues were ground with a biomasher using 350 μL MeOH:H₂O (4:1). Protein pellets were resuspended in 200 μL Laemmli buffer (10% SDS, 1 M Tris pH 6.8, and glycerol) and then centrifuged at 16.600 x g at 4 °C for five minutes. The protein

concentration was determined using a DC assay (Bio-Rad, Hercules, CA, USA) following the manufacturer's instructions. Each sample, containing 100 μg protein lysate, was heated at 95 °C for five minutes and loaded onto a 5% acrylamide SDS–PAGE stacking gel (prepared in-house). Prior to overnight digestion at 37 °C using modified porcine trypsin (Mass Spectrometry Grade, Promega, Madison, WI, USA) at an enzyme–protein ratio of 1:80, the gel bands were reduced and alkylated. The resulting peptides were extracted using 60% acetonitrile (ACN) followed by 100% ACN. The peptides were resuspended in 30 μL $H_2O$, 2% ACN, and 0.1% fluoroacetic acid. iRT peptides (Biognosys, Schlieren, Switzerland) were added to each sample as an internal quality control (QC). NanoLC–MS/MS analyzes were conducted using a nanoAcquity UltraPerformance LC (UPLC) system (Waters, Milford, MA, USA) coupled to a Q-Exactive Plus Mass Spectrometer (Thermo Fisher Scientific). The solvent system consisted of 0.1% formic acid (FA) in water (solvent A) and 0.1% FA in ACN (solvent B). Samples equivalent to 800 ng protein were loaded onto a Symmetry C18 precolumn (20 mm × 180 μm with 5 μm diameter particles, Waters) over three minutes at a flow rate of 5 μL/min with 99% solvent A and 1% solvent B. Peptides were separated on an ACQUITY UPLC BEH130 C18 column (250 mm × 75 μm with 1.7 μm diameter particles) at a flow rate of 400 nL/min. The following gradient of solvent B was used: 1–8% over 2 min, 8–35% over 77 min, 35–90% over 1 min, 90% for 5 min, and 90–1% over 2 min. Samples from each cohort were injected in a random order. The system was operated in data-dependent acquisition (DDA) mode, automatically switching between MS (mass range 300–1800 $m/z$ with R = 70 000, automatic gain control [AGC] fixed at 3.10^6 ions, and the maximum injection time set to 50 ms) and MS/MS (mass range 200–2000 $m/z$ with R = 17 500, AGC fixed at 1.10^5, and the maximum injection time set to 100 ms) modes. The ten most abundant precursor ions were selected on each MS spectrum for further isolation and higher energy collisional dissociation, excluding monocharged and unassigned ions. The dynamic exclusion time was set to 60 s. A sample pool was injected as an external QC every six samples for the human cohort and every five samples for the mouse cohort. MaxQuant[65] version 1.6.14 was used to process raw data. The Andromeda[66] search engine was used to assign peaks with trypsin/P specificity against a protein sequence database generated in-house containing all human (20 421 entries as of August 24, 2020) or mouse entries (17 134 entries for SOD1 and TDP-43 models as of March 27, 2020, and 17 061 entries for C9orf72 and FUS models as of September 29, 2020) extracted from UniProtKB/Swiss-Prot. Methionine oxidation/acetylation of the protein N-termini was used as a variable modification, whereas cysteine carbamidomethylation was used as a fixed modification. The "match between runs" option was enabled to facilitate protein quantification. To control false discoveries, a maximum false discovery rate of 1% was implemented (at both peptide and protein levels), employing a decoy strategy. Intensities were extracted from the Proteingroup.txt file for statistical analyzes. The MaxQuant protein vs. sample table was used for downstream analyzes, including label-free quantitation intensities. Only Swiss-Prot proteins were retained, whereas TrEMBL proteins were removed for higher reliability. After filtering out low abundant proteins, i.e., proteins that were detected in less than 50% of the samples in any combination of condition and sex, and imputing missing values using the missForest[67] algorithm, the intensities were log2-transformed and used for principal component exploration, constructing heat maps, and differential abundance analysis. The limma package[68] was used for linear modeling, and $p$-values were using the Benjamini–Hochberg correction. The protein names were mapped to the corresponding genes and used to search for enriched biological processes and molecular functions using the criteria described

in the transcriptomics data processing, with a $p < 0.01$ threshold for functional annotation analyzes.

## Phosphoproteomics on mouse PFC brain tissue samples

Starting with the protein extracts from the global proteomics experiments, protease inhibitors (Sigma-Aldrich, P8340) and phosphatase inhibitors (final concentration in $Na_3VO_4 = 1$ mM) were added to samples. Protein concentration was determined using the RC DC Protein Assay (Bio-Rad) according to the manufacturer's instructions. Proteins for each sample (250 μg) were reduced and alkylated prior to an in-house optimized single-pot, solid-phase-enhanced sample preparation protocol (adapted from Hughes et al., Nat Protoc, 2019). Briefly, beads A (Sera-Mag SpeedBeads, Fisher Scientific, Schwerte, Germany, 45152105050250) and B (Sera-Mag SpeedBeads, Fisher Scientific, 65152105050250) were combined (1:1) and, after three washing steps with $H_2O$, were added to the samples (bead–protein ratio of 10:1 for each type of bead, meaning a 20:1 ratio for the combination of beads). After inducing protein binding to the beads with 100% ACN for 18 min, the bead and protein mixtures were washed twice with 80% ethyl alcohol and once with 100% ACN before being resuspended in 95 μL $NH_4HCO_3$ prior to overnight on-bead digestion (enzyme–protein ratio of 1:20) at 1 000 rpm at 37 °C using modified porcine trypsin/Lys-C Mix (Mass Spectrometry Grade, Promega). The digestion was stopped using trifluoroacetic acid (TFA) (final pH <2). The recovered peptides were resuspended in 170 μL 80% ACN and 0.1% TFA, and MS PhosphoMix I Light (Sigma-Aldrich) was added to each sample (peptide [μg]–mix [fmol] ratio of 1:6). Phosphopeptide enrichment was performed on 5 μL phase Fe(III)–NTA cartridges on an AssayMAP Bravo platform (Agilent) following the immobilized metal affinity chromatography protocol. Briefly, cartridges were washed and primed with 50% ACN, 0.1% TFA and then equilibrated with 80% ACN, 0.1% TFA. Subsequently, 100 μL samples were loaded onto the phase at a rate of 2 μL/min and then washed with 80% ACN, 0.1% TFA before being eluted in 20 μL 1% $NH_4OH$ at 5 μL/min. After enrichment, FA and MS PhosphoMix I Heavy (Sigma-Aldrich) (peptide [μg]–mix [fmol] ratio of 1:6) were added to each sample. Dried phosphopeptides were resuspended in 40 μL $H_2O$, 2% ACN, 0.1% FA. Sample preparation steps for the C9orf72 and FUS mouse models were identical to those previously described for SOD1 and TDP-43, except that proteins were extracted from new tissue samples immediately before the phosphoproteomics experiment.

NanoLC–MS/MS analyzes were conducted using a nanoAcquity UPLC system (Waters) coupled with a Q-Exactive HF-X Mass Spectrometer (Thermo Scientific) equipped with a Nanospray Flex ion source. The solvent system consisted of 0.1% FA in water (solvent A) and 0.1% FA in ACN (solvent B). Samples were loaded onto an ACQUITY UPLC Peptide BEH C18 Column (250 mm×75 μm, 1.7 μm diameter particles) over three minutes at a flow rate of 5 μL/min with 99% solvent A and 1% solvent B. Phosphopeptides were separated on an ACQUITY UPLC M-Class Symmetry C18 Trap Column (20 mm×180 μm, 5 μm diameter particles; Waters) at a flow rate of 400 nL/min using a gradient of solvent B as follows: 1–2% over 2 min, 2–35% over 77 min, and finally 35–90% over 1 min. The samples in each group were injected in random order. The instrument was operated in DDA mode with automatic switching between the MS and MS/MS modes. MS scans were performed in the mass range of 375–1500 $m/z$ with R = 120 000, an AGC fixed at 3.106 ions, and a maximum injection time of 60 ms. MS/MS scans were conducted in the mass range of 200–2000 $m/z$ with R = 15 000, an AGC fixed at 1.105 ions, and a maximum injection time of 60 ms. In each MS spectrum, the ten most abundant ions were selected for further isolation and higher-energy collisional dissociation, excluding monocharged and unassigned ions. A dynamic exclusion time of 40 s was used to prevent the reanalysis of previously selected ions. The obtained raw phosphoproteomics data were processed using

MaxQuant version 1.6.14. Peaks were assigned using the Andromeda search engine with trypsin/P specificity against an in-house-generated protein sequence database containing mouse entries from UniProtKB/Swiss-Prot (17 061 entries as of September 29, 2020). The peptides were required to have a minimum length of seven amino acids, and a maximum of one missed cleavage was allowed. These modifications include methionine oxidation, acetylation of protein N-termini, and phosphorylation of serine, threonine, and tyrosine. Cysteine carbamidomethylation was used as a fixed modification. Protein quantification utilized the "match between runs" option. A maximum false discovery rate of 1% (at both peptide and protein levels) was implemented using a decoy strategy. The intensities were extracted from the Phospho (STY).txt file and processed using the Perseus platform[69] version 2.0.7.0, during which contaminants and reversed proteins, as well as proteins with negative scores, were removed. Using the "expand sites table" option, the intensities of the different phosphopeptides involved in one phosphosite were summed, and phosphosites with a localization probability below 75% were removed. The Perseus output table was used for statistical analyzes.

Only proteins that were detected in more than 50% of the mouse samples of any combination of sex and condition were retained for each dataset. The intensities were log2-transformed, quantile normalization was applied, and missing values were imputed to the lowest quartile. The remaining data processing for phosphoproteomics followed the procedure described above for proteomics data processing. For the DE analysis we used a cut-off of p-value 0.1.

## Differential expression analysis

To compare murine and human samples, transcript counts were normalized using DESeq2 size factor estimation. Subtype-specific differential expression of transcripts was determined using a 1.5-fold change cut-off and an adjusted p-value < 0.05 (unless stated otherwise). Two different models were designed to analyze sex-specific differences: The cohort was divided into male and female cohorts. For each cohort we carried out differential expression between case and control samples using the RNA-seq batch as covariate for the human samples.

Transcriptomics data has been processed using the NextFlow Core RNASeq pipeline, version 3.0 described[61]. The data has been demultiplexed with bcl2fastq, and the fastq files have undergone several quality checks including FastQC[62]. FastQC: A Quality Control Tool for High Throughput Sequence Data. Available online at: http://www.bioinformatics.babraham.ac.uk/projects/fastqc/) and fastq screen[70]. Salmon[63] was used for pseudo alignment and quantitation, with a salmon index built using GRCm39 with annotations from GENCODE vM26 for the mice data and GRCh38 with annotations from GENCODE v37 for the human data. Count matrices from Salmon were used in downstream analyzes.

The Principal Component Analysis and heatmaps used the count matrices from Salmon, after filtering and normalization using a variance stabilizing transformation blind to the experimental design. The count matrices were filtered, keeping genes with at least ten counts in 50% of the samples of any condition and sex.

Following differential expression analysis by DESeq2[71], we searched for relevant biological processes and molecular functions using gene set enrichment analysis on their Gene Ontology terms, using the clusterProfiler R package[64], filtering terms by size between 10 and 500 genes, and adjusting p-values for multiple testing with the Benjamini-Hochberg correction. To assess the heterogeneity of the samples, silhouette scores were calculated on the first two principal components and averaged across conditions.

## Small RNA sequencing data-processing and miRNA target prediction

Small RNA data were processed using the Nextflow Core smRNASeq pipeline version 1.0[61]. Reads were trimmed and aligned against miRBase version 22.1 using Bowtie1[72], both for mature miRNAs and hairpins. miRNAs with at least ten counts in 50% of the samples of any condition and sex were retained, and the rest were filtered out. Unnormalized count matrices were used for subsequent DESeq2[71] differential expression analysis after stratifying by sex and for p-values using the Benjamini−Hochberg correction. Mature miRNAs were mapped to their corresponding genes using miRDB version 6.0[73], excluding matches with scores <60 or >800 targets, as recommended. For each miRNA present in the miRNA expression matrices, we obtained experimentally validated targets from miRTarBase[74] 8.0 and predicted targets from miRDB miRBase version 22[73]. miRTarBase[74] provides the most extensive curated database of validated miRNA−target interactions (MTI) collected from literature using natural language processing to select functional miRNA studies. Additionally, the miRDB database includes MTIs predicted by MirTarget, which uses a support vector machine (SVM) to analyze thousands of high-throughput sequencing experiments[75]. Each experiment is assigned a probability score, which serves as the SVM's output. A higher probability score indicates a higher likelihood of accurate target prediction. Therefore, we set a threshold of 0.6 on the output probabilities to select only highly likely MTIs. Finally, we joined the miRNA−target pairs from both sources for further analysis. For the DE analysis we used a cut-off of p-value 0.1 and threshold of log2 fold change (±1.5). Target network centrality was calculated using networkx (v2.8.8) using the eigenvector_centrality function on the STRING interaction network the miRNA targets.

## Cell-type deconvolution analyzes

Single-cell reference-based cell-type deconvolution of RNA-Seq in mouse models was performed using Scaden[34]. We used healthy adult scRNA-Seq datasets for mouse[76] and human[77]. Scaden uses a fully connected deep neural network ensemble trained on pseudo-bulks simulated from reference scRNA-Seq data. Before deconvolution, we filtered the scRNA-seq data using Scanpy[78] to maintain at least 200 genes expressed per cell and at least five cells expressing one gene. For Scaden, counts per million (CPM) of simulated pseudo-bulks and transcripts per million (TPMs) of the data to be deconvolved were used. In this study, CPM was used for scRNA-Seq instead of TPMs because scRNA-Seq consists of unique molecular identifier counts and does not include gene-length bias. We used a variance cutoff of 0.01, and the mean squared error was calculated for each batch as loss as implemented in the Scaden code repository (https://github.com/KevinMenden/scaden). For comparing the cell type abundances between human ALS subclusters and mouse models, the relative median differences from control samples was computed using the following equation:

Relative difference
$$= (\text{median composition in ALS} - \text{median composition in controls})/ \quad (1)$$
$$(\text{median in composition in control samples}).$$

## Differential alternative splicing analysis

The splicing tool SUPPA2[79] version 2.3 was used to analyze DAS for seven alternative splicing events: exon skipping, mutually exclusive exons, intron retention, alternative 3'-splice site, alternative 5'-splice site, alternative first exon, and alternative last exon. SUPPA2 was used with multipleFieldSelection() to select the TPM values of the transcripts, followed by GenerateEvents with the parameters -f ioe -e SE SS MX RI FL and annotation files GENCODE v37 for human data and GENCODE vM26 for mouse data. The inclusion values (PSI) were calculated using psiPerEvent, and the differences in PSI (ΔPSI) between the mutant and control conditions were determined using diffSplice with parameters -m empirical -l 0.05 -gc −save tpm_events to detect

anomalies in the splicing landscape. For the DE analysis, we used a cut-off of p-value 0.1.

## Multi-omics factor analysis

We used MOFA2[25] version 1.4.0 to integrate data from multiple omics levels for the human cohort, including transcriptomics, miRNA, and proteomics. The MOFA model was trained on the data, and downstream analyzes were performed. Each omics type was preprocessed in its own manner, and the default training parameters were used. The MOFA models were initialized with 15 initial factors, and convergence was reached when the evidence lower bound (ELBO) value did not change with more than a deltaELBO value of 1e-4%.

## Enrichment analyzes

**Gene ontology, pathway enrichment analyzes, and protein interaction networks.** GSEA[80] was performed using gseGO and gseKEGG from the clusterProfiler R package, with biological processes and molecular functions chosen as background databases for GO enrichment. The p-value cutoff was set at 0.05. Differential expression was presented using volcano plots generated with the Enhanced Volcano plot package in R, and over-representation analysis was performed on genes that showed at least one significant DAS event using the clusterProfiler function enrichGO, with a p-value cutoff of 0.1 and Benjamini–Hochberg correction for multiple hypothesis testing. Protein–protein interaction networks were created using the STRING protein interaction network database version 11.0 using standard settings. We summarized the enriched GO terms by clustering pathways based on their descriptions, which is called clustering in semantic space using GO-Figure[37]. In addition, for proteomics results, we performed additional clustering analyzes using REVIGO[81]. This tool allows GO term clustering by hierarchy, considering the following such as enrichment p-adjusted values, semantic similarity and term proximity.

**Weighted Gene Correlation Network Analysis (WGCNA).** To conduct a weighted gene co-expression analysis, we employed the WGCNA package[15]. Pairwise Pearson correlations were calculated to establish signed regulatory networks within the WGCNA. By constructing an adjacency matrix, we applied a soft-thresholding technique to approximate a scale-free topological network. Eigengenes or eigenproteins were calculated as the first principal components of each module. This resulted in the development of several modules. We merged similar modules based on hierarchical clustering (SOD1: 0.4, C9orf72: 0.4, FUS: 0.5, TDP-43: 0.4 for RNA-Seq and 0.25 for proteomics). We calculated the relationships between WGCNA modules and traits. Sex was also included in the traits resulting in four traits: male ALS, female ALS, male CTR, and female CTR. First, we filtered modules based on the significance of the module–gene relationship ($p < 0.05$) and then selected modules that were highly correlated with either male or female ALS. The correlation cutoffs differed between the mouse models (SOD1: 0.5, C9orf72: 0.3, FUS: 0.3, TDP-43: 0.5 and human: 0 for RNA-Seq). The minimal module size was set to 30 with a merge height of 0.4–0.5 and a correlation threshold of 0.3–0.5 (Supplementary Data 16). Over-representation analysis was performed using the enrichGO function of clusterProfiler[64].

**Pathway selection for clustering analyzes with transcriptomics datasets.** The selection of the pathways used for clustering analyzes of human samples was based on the enrichment analyzes performed with the transcriptomics dataset. We selected the top hits from gseGO and gseKEGG analyzes [from the clusterProfiler R package[64]], as well as a co-expression network analysis (weighted gene co-expression analysis - (WGCNA) analysis, analyzing the top 30 most highly enriched terms for each dataset, and then grouped these terms by more general umbrella terms, to select ten pathways that summarized the enriched

terms best: mitochondria/respiration; synapse; MAPK cascade; oxidative stress; nucleocytoplasmic transport; protein folding/metabolism; lipid metabolism; RNA splicing; extracellular matrix and activation of immune response. In more detail, for overlapping KEGG pathway and GO results, a predominant enrichment for extracellular matrix (ECM) and immune response pathways and synapse-related terms was observed for both males and females. The convergence of these themes across sexes suggests common biological themes relevant for the disease and justified their selection for the clustering analyzes. In addition to neurodegenerative disease pathways driven by DE genes involved in protein metabolism and oxidative stress mechanisms (Supplementary Fig. 5), female-specific results showed enrichment for aerobic respiration (GO) and ribosome (KEGG), while protein metabolism was captured for male results (KEGG). Oxidative stress was also inferred from the enrichment of oxidative phosphorylation pathways in both males and females (KEGG). WGCNA results revealed key associations with all of the selected themes, especially for terms related to mitochondria/respiration and protein metabolism, synapse, nucleocytoplasmic transport, lipid metabolism; immune response and RNA splicing. WGCNA results were also very frequently enriched for MAPK-related terms (Supplementary Data 8), justifying its selection for the clustering analyzes.

**Identification of transcriptome-based subclusters in human ALS patients.** We used decoupleR (v1.1.0)[82] to aggregate scores from gene sets derived from GO terms of interest. Gene lists with the components of GO terms of interest were extracted using the AmiGO2 database (https://amigo.geneontology.org/). For the unsupervised per sample enrichment analyzes, we used decoupleR with the consensus from the mlm, ulm, wsum functions[82] and calculated activity scores. These are numerical values associated with genes and represent gene expression levels.

## Mouse primary cortical cultures

Primary cortical neuronal cell cultures were generated from neonatal mouse C57BL/6 J pups aged postnatal day 0–1 (P0–P1) in accordance with ethical guidelines for animal experimentation at local and international levels. Animal care strictly followed official governmental protocols, with utmost effort to minimize the number of animals utilized and to mitigate any potential suffering or distress. Pups were decapitated, and the brains were collected in dissection media containing 10× Hanks balanced salt solution and $NaHCO_3$. The cortex was dissected, the meninges were removed, and small pieces of the cortex were collected in a Falcon tube. Tissues were trypsinized at 37 °C in a water bath for 12 min and treated with 200 µL DNAse I (10 mg/mL). The tissue strips were gently triturated (until the tissue fragmented) in fetal bovine serum using a fire-polished Pasteur pipette. The mixture was subsequently centrifuged at $800 \times g$ for four minutes, and the resulting cell pellet was resuspended and cultured in a neurobasal medium supplemented with B27 and antibiotics (0.06 µg/mL penicillin and 0.1 µg/mL streptomycin). Cells were seeded at a density of $3 \times 10^5$ cells per well in 24-well plates. Prior to cell seeding, the coverslips were acid-washed, rinsed several times with water, sterilized with ethanol and UV light, and placed in a well plate. The plates were then coated with poly-L-ornithine (0.05 mg/mL) overnight and laminin (10 µg/mL) for 2 h in an incubator before use. The cells were maintained at 37 °C in a humidified incubator under 5% $CO_2$ for seven days prior to the experiments. The cell culture medium was replaced every three days during this period. To induce glutamate excitotoxicity, L-glutamic acid (Tocris, UK) was dissolved in 50 mM NaOH, and a stock solution of 50 mM was prepared prior to use. An appropriate concentration of glutamate was prepared in the maintenance medium (neurobasal medium supplemented with B27 and antibiotics). The cells were exposed to 5 mM glutamate by exchanging 1:3 medium at seven days in vitro. After six hours of incubation, glutamate was thoroughly

removed by washing, and the cells were fixed for immunocytochemistry or lysed for protein extraction.

## Immunocytochemistry and microscopy for in vitro experiments

Cells were cultured on coverslips following previously described methods[83] and immunostained at seven days in vitro according to standard techniques. Cells were fixed in 4% paraformaldehyde in PBS at room temperature for 10 min. To quench the free aldehyde groups, the cells were treated with 50 mM $NH_4Cl$ for 15 min and washed with PBS. For the permeabilization of the cell membrane, PBS with 0.25% Triton X-100 was added for 10 min at room temperature. Non-specific binding sites were blocked by applying 10% goat serum in PBS for at least 20 min. Dilutions of primary antibodies were prepared in blocking solution to a final volume of 180 µL per 18-mm coverslip, and cells were incubated for 90 min at 37 °C with shaking. The following primary antibodies were used: mouse anti-MAP2 (Invitrogen, Waltham, MA, USA; #MA5-12826, RRID:AB_10976831) (1:500) and rabbit anti-cleaved caspase 3 (Cell Signaling Technology, Danvers, MA, USA; #9661, RRID: AB_2341188) (1:250). The cells were washed three times for five minutes each with PBS before applying the secondary antibodies. The cells were then incubated with secondary antibodies for 30 min and washed, respectively Alexa Fluor 488 (Invitrogen, Waltham, MA, USA #A-11034, RRID: AB_2576217) (1:250) and goat Anti-rabbit Cy3 (Jackson ImmunoResearch, Ely, United Kingdom, #111-165-144, RRID: AB_2338006) (1:250). For double staining, a second primary antibody was added, and the same steps were repeated. Coverslips were mounted on slides using a mounting medium supplemented with DAPI. Images were captured using an inverted fluorescence microscope (Zeiss, Jena, Germany) with a 63× oil objective and analyzed using ImageJ software. Fifteen random images from each coverslip were analyzed for cell death by counting the number of cleaved caspase-3 positive cells. Neurite lengths were measured using a simple neurite tracing plugin in ImageJ. Statistical analyzes were conducted using GraphPad Prism version 9.4.1 (GraphPad, San Diego, CA, USA). Outliers were identified and removed using the Grubbs test (α = 0.1). Comparisons were done using one-way analysis of variance (ANOVA), and data were plotted as the mean ± standard error of the mean (SEM) of at least five independent experiments. Differences were considered statistically significant at p < 0.05.

## Protein extraction and western blotting

For protein analysis, cells were washed once with 1× PBS, and after adding RIPA lysis buffer, protease inhibitor cocktail (1:25), and phosphatase inhibitor (1:20), cells were incubated on ice for five minutes. The cells were scraped off with a cell scraper, transferred into 1-mL reaction tubes, and homogenized by passage through a U-100 insulin syringe several times. Protein concentration was determined using a Pierce BCA Protein Assay Kit (Thermo Fisher Scientific) following the manufacturer's instructions. One microliter of protein sample was used for the assay. Colorimetric reactions were analyzed using an Infinite M200 PRO ELISA plate reader (Tecan, Männedorf, Switzerland). Twenty grams of each sample were loaded onto a gel (NuPAGE 4 to 12%, Bis-Tris) (Invitrogen, Waltham, MA, USA). NuPAGE LDS sample buffer (1:4) and sample-reducing buffer (1:10) were added to lysed protein before loading onto the gels, incubated with shaking at 75 °C for 13 min, and centrifuged at 12 000 × g at 4 °C. The proteins were subjected to gel electrophoresis at 200 V. Subsequently, the proteins were transferred onto nitrocellulose membranes using an iBlot2 gel transfer device (Thermo Fisher Scientific, Waltham, MA, USA). The membranes were then blocked with 5% nonfat milk in PBST at room temperature for 30 min. For antibody incubation, primary antibodies (diluted in blocking buffer) were added to the membranes and allowed to incubate overnight at 4 °C with rotation. Rabbit anti Lamin B1 (Proteintech, Planegg-Martinsried, Germany, #12987-1-AP, RRID: AB_2136290) (1:2000), rabbit anti p44/42 mitogen-activated protein kinase (anti-Erk1/2) (Cell Signaling Technology, Danvers, MA, USA, #9102, RRID: AB_330744) (1:1000), rabbit anti MEK2 (Cell Signaling Technology, Danvers, MA, USA, #9125, RRID: AB_2140644) (1:1000), rabbit anti Phospho-p44/42 MAPK(ERK1/2) (Cell Signaling Technology, Danvers, MA, USA, #9101, RRID: AB_331646) (1:1000) and rabbit anti Phospho-MEK1/2 (Ser217/221) (Cell Signaling Technology, Danvers, MA, USA, #9121, RRID: AB_331648) (1:1000). After four washes with PBST (five minutes each), the membranes were incubated with peroxidase-conjugated goat anti-rabbit secondary antibodies (Vector Laboratories, Newark, CA, USA, #PI-1000, RRID: AB_2336198) (diluted 1:10 000 in blocking buffer) at room temperature for one hour. Finally, the membranes were washed thoroughly with PBST to remove the unbound antibodies. Blots were incubated with enhanced chemiluminescence reagent and were imaged using a Molecular Imager ChemiDoc (Bio-Rad, Hercules, CA, USA) imaging system. Band signal intensities were quantified using ImageJ software and normalized to housekeeping proteins and controls. Statistical analyzes were conducted using GraphPad Prism version 9.4.1. Outliers were identified and removed using the Grubbs test (α = 0.1). Comparisons were done using one-way ANOVA, and data were plotted as the mean ± SEM of at least three independent experiments. Differences were considered statistically significant at p < 0.05.

## Tissue protein extraction for immunoblot analysis

Spinal cords were homogenized in five volumes (w/v) of 1% boiling SDS[84]. Protein homogenates were sonicated, boiled for 10 min, and centrifuged at 13 500 × g for five minutes. Supernatants were analyzed by dot blotting. For detergent-insoluble protein extraction, mouse tissues were homogenized in 10 volumes (w/v) of buffer (15 mM Tris–HCl pH 7.6, 1 mM dithiothreitol, 0.25 M sucrose, 1 mM $MgCl_2$, 2.5 mM EDTA, 1 mM EGTA, 0.25 M $Na_3VO_4$, 2 mM $Na_4P_2O_7$, 25 mM NaF, 5 µM MG132) and a protease inhibitor cocktail (Roche, Basel, Switzerland)[85]. Briefly, the samples were centrifuged at 10 000 × g, and the pellet was suspended in an ice-cold homogenization buffer containing 2% Triton X-100 and 150 mM KCl. The samples were then centrifuged at 10,000 × g to obtain the Triton-insoluble (insoluble) fraction.

## Immunohistochemistry

Mice were anesthetized and underwent transcardial perfusion using 50 mL PBS followed by 100 mL of 4% paraformaldehyde (in PBS). The spinal cord was promptly dissected, postfixed for three hours, and subsequently transferred to a solution of 20% sucrose in PBS overnight. Once sunk, the spinal cord was transferred to a 30% sucrose solution until it reached the desired consistency. Next, the spinal cord was frozen in n-pentane at 45 °C and stored at ~−80 °C. Prior to freezing, the spinal cord was divided into cervical, thoracic, and lumbar segments and embedded in Tissue-Tek O.C.T. compound (Sakura Finetek, Torrance, CA, USA). Coronal sections (30 µm thick) were sliced from the lumbar spinal cord and subjected to immunohistochemistry. Rabbit monoclonal anti-pMEK (Ser221) (pMEK2) antibody (1:50, Cell Signaling, Danvers, MA, USA, RRID: AB_490903) was used. In brief, slices were incubated with blocking solutions (0.2% Triton X-100 and 2% normal goat serum) at room temperature for one hour. Subsequently, the slices were incubated overnight at 4 °C with primary antibodies. The sections were then incubated with biotinylated secondary antibodies (1:200) for one hour at room temperature, followed by immunostaining using an avidin–biotin kit and diaminobenzidine. To facilitate visualization, sections were counterstained with 0.5% cresyl violet. Sagittal sections (20 µm thick) were sliced from tibialis anterior and subjected to immunofluorescence. Mouse anti-synaptic vesicle glycoprotein 2 A (SV2, 1:50, DSHB, RRID: AB_2315387), mouse anti-neurofilament medium polypeptide 2H3 (1:12, DSHB, RRID: AB_2314897) and α-bungarotoxin (α-BTX) coupled to Alexa Fluor™ 594 (1:500; Invitrogen) was used. In brief, slices were fixed in acetone for

10 minutes and incubated for 1 h at room temperature with α-BTX. Subsequently, the slices were incubated with blocking solutions (0.5% Triton X-100, 5% bovine serum albumin and 5% normal goat serum) at room temperature for one hour. Subsequently, the slices were incubated overnight at 4 °C with primary antibodies. The sections were then incubated with anti-mouse 647 (1:500, Invitrogen) for one hour at room temperature.

Stained sections were captured at 20× and 40× magnification using an Olympus BX-61 Virtual Stage microscope (Olympus Life Sciences, Tokyo, Japan), ensuring complete stitching of the entire section with a pixel size of 0.346 μm. Images were acquired using 6-μm-thick stacks (step size of 2 μm). The different focal planes were merged into a single stack using mean intensity projection to maintain consistent focus across the sample. Finally, the acquired signals were analyzed for each slice using ImageJ and OlyVIA software (Olympus Life Sciences, Tokyo, Japan).

### Immunoblotting for in vivo experiments
Protein levels were quantified using a BCA Protein Assay Kit (Pierce Biotechnology, Waltham, MA, USA) and analyzed by western blotting and dot blotting[86]. Membranes were blocked with a solution of 3% (w/v) BSA (Sigma-Aldrich, St. Louis, MO, USA) and 0.1% (v/v) Tween 20 in Tris-buffered saline (pH 7.5). Subsequently, the membranes were incubated with primary antibodies followed by peroxidase-conjugated secondary antibodies (GE HealthCare, Chicago, IL, USA). The antibodies utilized were rabbit monoclonal anti-pMEK2 (1:2000, Cell Signaling Technology; Danvers, MA, USA, RRID: AB_490903), rabbit monoclonal anti-phospho-p44/42 MAPK (Erk1/2) (Thr202/Tyr204) (1:2000, Cell Signaling Technology, Danvers, MA, USA,; RRID: AB_2315112), rabbit polyclonal anti-human SOD1 (1:1000, StressMarq Biosciences, Victoria, Canada; RRID: AB_2704217), rabbit polyclonal anti-ubiquitin (1:1000, Abcam, Cambridge, UK; RRID: AB_306069), and mouse monoclonal anti-SQSTM1/p62 (p62) (1:500, Abcam, Cambridge, United Kingdom, RRID: AB_945626). Peroxidase-conjugated secondary antibodies (goat anti-mouse and anti-rabbit) (GE HealthCare, Chicago, IL, USA) were used at dilutions of 1:20 000 and 1:10 000, respectively. The blots were developed using the Luminata Forte Western Chemiluminescent HRP Substrate (MilliporeSigma, Burlington, MA, USA) and visualized on a ChemiDoc Imaging System (Bio-Rad, Hercules, CA, USA). Densitometry analysis was performed using Image Lab software version 6.0 (Bio-Rad, Hercules, CA, USA). Protein immunoreactivity was normalized to Ponceau Red staining (Honeywell Fluka, Charlotte, NC, USA) for accurate quantification.

### Preclinical study in SOD1^G93A mice
SOD1^G93A female and male mice received 3 mg/kg dose of trametinib or vehicle (PBS) through intranasal delivery, twice per week. The treatment of SOD1^G93A female and male mice started from 9 weeks of age until 16 weeks of age, when they were sacrificed for biochemical analysis. The effect of trametinib on the progression of the disease in SOD1^G93A female mice was assessed twice a week from 9 weeks of age by measuring body weight, hind limb extension reflex and grip strength. The extension reflex was quantified using the following 3-point score system: 3, hind limbs extending to form an angle of 120 degrees; 2.5, hind limbs extending to <90 degrees with decreased reflex in a hind limb; 2.0, as 2.5 with decreased reflex in both hind limbs; 1.5, loss of reflex with marked flexion in a hind limb; 1, as 1.5 with marked flexion in both hind limbs; 0.5, loss of reflex with hind limbs and paws held close to the body, but still able to walk; 0, as 0.5 but unable to walk[85]. The grip strength test was performed by placing mice on a horizontal metallic grid which was then gently inverted. The latency to fall of each mouse was recorded. The test ended after 90 s. In the case of failure, the measurement was repeated three times and the best performance of the session was considered for the statistical analysis[87]. Age of paralysis was defined as the age at which the mice lost reflex with hind limbs and paws held close to the body (0.5 score at extension reflex test). Disease onset was retrospectively determined as the average age at which the mouse exhibits the first failure, at two consecutive time points, from both maximum weight and maximum performance in extension reflex score and grip strength test. The animals were euthanized when they showed rigid paralysis in the hind limb and a 0.5 score in hind limb extension reflex. Data of behavioral tests and body weight loss were evaluated by two-way ANOVA for repeated measures. Disease onset and survival length were evaluated by log-rank Mantel-Cox test. This study complied with international and local animal welfare standards.

### Plasma neurofilament light chain quantification
Mouse plasma samples were collected in K2-EDTA BD Microtainer blood collection tubes and centrifuged at 5 000 × g for five minutes to obtain the plasma. The plasma NfL was quantified using a Simoa NF-light Advantage (SR-X) Kit (#103400) on a Quanterix SR-X platform (Quanterix, Boston, MA, USA). All reagents used were from a single lot, and measurements were performed according to the manufacturer's protocol.

### Quantitative real-time polymerase chain reaction
Total RNA from Gastrocnemius muscle was extracted using Trizol (Invitrogen, Waltham, MA, USA) and purified with PureLink RNA columns (Life Technologies, Carlsbad, CA, USA). RNA samples were treated with DNase I and reverse transcribed with the High-Capacity cDNA Reverse Transcription Kit (Life Technologies, Carlsbad, CA, USA). For Quantitative real-time polymerase chain reactions (qRT-PCR), we used the Taq Man Gene expression assay (Applied Biosystems, Waltham, MA. USA), on cDNA specimens in triplicate, using 1X Universal PCR master mix (Life Technologies, Carlsbad, CA, USA) and 1X mix containing specific receptor probes for mouse nicotinic acetylcholinergic receptor, gamma subunit (AChRγ) (Mm00437419_m1; Life Technologies, Carlsbad, CA, USA). Relative quantification was calculated from the ratio of the cycle number (Ct) at which the signal crossed a threshold set within the logarithmic phase of the given gene to that of the reference mouse β-actin gene (Mm02619580_g1; Life Technologies, Carlsbad, CA, USA). The means of the triplicate results for each sample were used as individual data for $2^{-\Delta\Delta Ct}$ statistical analysis.

### Statistical analyzes of in vivo experiments
Statistical analyzes of in vivo experimental data were performed using GraphPad Prism 7.0 and statsmodels Python package (v0.13.0). Student's t-tests or one-way ANOVAs, followed by pairwise TukeyHSD post hoc tests, were used to analyze the differences between the experimental groups for each variable. Statistical significance was set at $p < 0.05$.

### Statistics and reproducibility
The experimental designs, statistical analyzes, and reproducibility parameters of this study are described below, encompassing both human and animal models. The human postmortem prefrontal cortex (PFC) samples included 51 ALS and 50 control specimens, sourced from multiple brain banks. The ALS animal models used included four genetically modified mouse strains, with cohorts designed with ten transgenic and ten non-transgenic mice per model, balanced for sex. Experiments were conducted with strict adherence to reproducibility guidelines. All procedures were standardized, and where possible, performed by the same investigator to minimize variation. No statistical method was used to predetermine sample size. The numbers were chosen based on historical data, which suggested that these sample sizes were sufficient to detect meaningful differences in the

parameters measured, with acceptable power and alpha levels. The experiments were not randomized. The investigators were not blinded to allocation during experiments and outcome assessment.

### Reporting summary

Further information on research design is available in the Nature Portfolio Reporting Summary linked to this article.

## Data availability

Raw RNA-Seq and processed gene expression data from animal models generated in this study were deposited to the National Center for Biotechnology Information Gene Expression Omnibus database (GSE234246). Encrypted raw RNA-Seq data for the human cohort generated in this study were deposited to the European Genome Phenome Archive (dataset: EGAD50000000467 and EGAD50000000468 for mRNA and small RNA sequencing data). The RNA-Seq data are available under restricted access, to guarantee the privacy of the subjects of the study and their blood relatives, since it contains sensitive phenotypic/transcriptomics information of the studies subjects. The access can be granted over the EGA database, with a formal request to the respective Data Access Committe (EGAC00001003287). Proteomics and phosphoproteomics datasets have been deposited in the ProteomeXchange Consortium database with the identifiers PXD043300, PXD043297 and PXD051889. Source data are provided with this paper.

## Code availability

Given the size and complexity of the datasets, data processing, and data analysis methods, we integrated all analyzes and raw data into a Data Version Control pipeline[88]. The code for the computational analyzes is available at: https://github.com/imsb-uke/MAXOMOD_Pipeline.

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

## Acknowledgements

We thank all the members of the Lingor, Bonn, Carapito, Schlapbach, Pasterkamp, and Bonetto laboratories for their feedback on the manuscript. This work was performed by the research consortium "Multi-omic analysis of axono-synaptic degeneration in motoneuron disease (MAX-OMOD)" funded in the scope of the E-Rare Joint Transnational Call for Proposals 2018 "Transnational research projects on hypothesis-driven use of multi-omic integrated approaches for discovery of disease causes and/or functional validation in the context of rare diseases." Consortium members: S.B., J.P., V.B., C.C., R.S., and M.K., coordinated by P.L.; L.C.G., M.P., and P.L. were supported by the Bundesministerium für Bildung und Forschung (01GM1917A). P.L., Q.Z., and D.E. were further supported by the Munich Cluster for Systems Neurology (SyNergy, EXC 2145 - ID 390857198). S.H. received funding from BMBF, grant 01GM2202A; STOP-FSGS and DFG, CRC1192. S.O. by SFB1286 SP02 and KFO296 P8; R.K. by FOR5068 P9; F. H. by the M3I excellence initiative and a UKE postdoctoral stipend; C.H. by KFO306 P11; and S.B. by SFB1286 SP02, SFB1192 PB8, and PC3. J.P. was supported by Stichting ALS Nederland (TOTALS). D. E. received funding from the European Union's Horizon Europe research and innovation program under grant agreement No 101057649 and the Association for Frontotemporal Degeneration (AFTD). We thank Tobias B. Huber for his support. The funders had no role in the study design, data collection and analysis, decision to publish, or manuscript preparation.

## Author contributions

P.L. and L.C.G. conceived the project, and S.H., F.H., R.K., S.O., and S.B. conceptualized the computational analysis. P.L., L.C.G., J.K., and M.K.K. designed the sample collection methodology, reviewed the sample and data quality, and coordinated the acquisition of human tissue samples from the brain banks as well as pathological, genetic, and clinical information. A.F.D. reviewed clinical information and provided demographical analyzes. L.C.G., M.P., L.P., M.G., E.S.L., and Q.Z. processed mouse and human brain samples for multiomics experiments with the infrastructure provided by V.B., C.C., D.E., and P.L. M.G., H.R., S.St., and E.L. performed multiomic experiments with the infrastructure provided by C.C. and R.S. C.D., M.S., T.B.H., I.C., and M.D. analyzed the genetic data. S.H., F.H., R.K., S.O., M.E., and C.H. were responsible for the computational data analysis, and conceived and planned the statistical analyzes. L.C.G., M.P., L.P., M.G., S.F.C., S.Sc., and P.Z. conducted the validation experiments with conceptual inputs from P.L., C.C., R.J.P., and V.B. L.C.G., S.H., F.H., R.K., L.T., M.P., L.P., and P.L. contributed to figure generation and assembly. L.C.G., S.H., S.B., and P.L. wrote the manuscript with input from all co-authors. L.C.G. and S.H. contributed equally. S.B. and P.L. contributed equally.

## Funding

## Competing interests

The authors declare no competing interests.

## Additional information

¹Technical University of Munich, School of Medicine, rechts der Isar Hospital, Clinical Department of Neurology, Munich, Germany. ²III. Department of Medicine, University Medical Center Hamburg-Eppendorf, Hamburg, Germany. ³Center for Biomedical AI, University Medical Center Hamburg-Eppendorf, Hamburg, Germany. ⁴Institute of Medical Systems Biology, University Medical Center Hamburg-Eppendorf, Hamburg, Germany. ⁵Research Center for ALS, Istituto di Ricerche Farmacologiche Mario Negri IRCCS, Milan, Italy. ⁶Laboratoire de Spectrométrie de Masse Bio-Organique, Université de Strasbourg,

Infrastructure Nationale de Protéomique, Strasbourg, France. [7]Institute of Medical Genetics and Applied Genomics, University of Tübingen, Tübingen, Germany. [8]German Center for Neurodegenerative Diseases (DZNE), München, Germany. [9]Munich Cluster for Systems Neurology (SyNergy), Munich, Germany. [10]Department of Translational Neuroscience, University Medical Center Utrecht, Utrecht University, Utrecht, The Netherlands. [11]Center for Rare Diseases, University of Tübingen, Tübingen, Germany. [12]Functional Genomics Center Zürich, ETH Zürich and University of Zürich, Zürich, Switzerland. [13]Department of Neurology, Medical University of Warsaw, Warsaw, Poland. [14]These authors contributed equally: Lucas Caldi Gomes, Sonja Hänzelmann. ✉e-mail: stefan.bonn@zmnh.uni-hamburg.de; paul.lingor@tum.de

