## [Peer Review File · Nature Communications]

Multiomic ALS signatures highlight subclusters and sex differences suggesting the MAPK pathway as therapeutic targetEditorial Note: Parts of this Peer Review File have been redacted as indicated to remove third-party material where no permission to publish could be obtained.

REVIEWER COMMENTS

Reviewer #1 (Remarks to the Author):

Manuscript background information: Amyotrophic lateral sclerosis (ALS) is a degenerative motor neuron disease that is more prevalent in males than females and has no effective treatment for sporadic cases (sALS). This study focuses on identifying and characterizing the sex-distinctive multiomic changes induced by early disease progression in ALS by comparing postmortem prefrontal cortices. Additionally, changes in the multiomic landscape of four transgenic mouse models of c9orf72- SOD1-, TDP-43-, and FUS-ALS were analyzed individually and in context of the human samples. From the transcriptomic dataset, four subclusters of ALS were identified. The mitogen-activated protein kinase (MAPK) pathway was also identified as a potential pathway key to early disease progress, and its modulation by trametinib showed promising therapeutic effect for female SOD1 transgenic mice.

Remarks to the Author: The integrated multiomic dataset encompassing both human and transgenic mice models that is the basis of this paper is interesting, though it is possible that one or more of the omics from the human samples have previously been analyzed since they are derived from banked samples. While it is of note that molecules implicated in the MAPK pathway regularly appear in the multiomics data analyses, this is not a novel pathway to be identified for ALS (1), and the sex-selective differences in the trametinib results may not justify the novelty requirements for this journal. Additionally, the provenance of the foundational data for the four subclusters of ALS is not fully explained (see below). I also suggest that because of the rich nature of the methods and results for this paper, it would be better placed in a journal where they could be fully explored.

Major comments:

(1) The study design is questionable. The main cells affected in ALS are the Betz cells located in the primary motor cortex. These cells are unique to higher mammals and are absent from rodents. A single-nuclei approach would have permitted to study the difference between males and females in humans (even if most are depleted, the remaining cells are sufficient to make such analysis). Deconvolution does not allow such resolution and accuracy. There are other groups of cells in the motor cortex affected in ALS and also present in mice. Comparing these would have been more relevant and interesting for the field.

(2) The clustering of human ALS samples was performed on the basis of activities scores across seven pathways. It is not clear to me how these pathways were selected and the means of acquiring the activities scores is relegated to identifying the R package (no version, no parameters, no input data). Because this clustering is foundational to the paper, these methods are critical and should be provided in full detail.

(3) The supplementary files are not of the same quality as the figures in the paper. The order of the files is inconsistent with the figure legends file (making it hard to review), there are subplots mentioned in

the figure legend that are not present (eg SI Fig 4), or potentially missing (SI Fig 9, though hard to tell because ordering was off). Below is an additional sampling of what I found (though by no means comprehensive):

a) SI Fig 11 e) ANOVA is likely more appropriate than multiple t-tests here

b) SI Fig 2 Euclidean distance should be 0 between sample and itself – scale and legend appear to be incorrect

c) SI Figure 26 labeled as male in title when should be female?

(4) The computational methods are incomplete. While a “Reproducible data pipeline” is mentioned in the methods, it does not appear to have been made available. Additionally, there are multiple instances where critical parameters to run the code have not been provided and the package versions have not been provided. There is also no code provided, just links to some of the packages/tools utilized. This lack of detail also extends to the statistics used to analyze the in vivo mouse data, where the post hoc test for the ANOVAs run is not mentioned.

(5) The enrichment methods and their description also could be improved. Throughout the paper, GSEA, ORA, and some graph-based methods (to a lesser extent) are used to identify pathways and processes that are of interest. It is not always clear which of these methods is being used in what context This becomes an issue, because in multiple analyses, there appears to be only one differential molecule supporting the identification of a pathway/term (particularly for the female samples).

Reference:

Sahana TG, Zhang K. Mitogen-Activated Protein Kinase Pathway in Amyotrophic Lateral Sclerosis. *Biomedicines*. 2021 Aug 6;9(8):969. doi: 10.3390/biomedicines9080969. PMID: 34440173; PMCID: PMC8394856.

Reviewer #2 (Remarks to the Author):

The paper by Gomes et al titled “Multiomic ALS signatures highlight sex differences and molecular subclusters and identify the MAPK pathway as a therapeutic target” performs a multi-omic study of transcriptomics, miRNAomic and proteomic (including phosphoproteomics) levels of 51 prefrontal cortex patients (split 70/30 males/females) and 50 controls (split 44/56 males/females) and compared these to four transgenic mouse models (C9orf72, SOD1, TDP-43 and Fus-ALS) to determine if there were changes in the early disease mechanisms and whether sex had an influence. In human patients, transcriptomic level changes were observed in genes clustered to ECM, mitochondrial dynamics and RNA metabolism which was recapitulated in the mouse models and determined that the MAPK pathway was affected in disease and in models with a potential therapeutic target being MAPKK2. The study itself is a very elegant study and provides excellent coverage across multiple omic platforms from a human source

which is crucial in ALS research given that much of the neurons that are affected by ALS are already gone by the time post-mortem tissue is obtained revealing only those that are still alive. The mouse models provide detail into the early disease mechanisms that may lead up to neurodegeneration and the fact that MAPK pathway was identified as the main pathway involved early disease. My comments are below:

- MAPK pathway was identified as the early onset pathway in disease, how disease-specific is this pathway i.e. is it due to ALS or due to a general stress response? Given that MAPK is highly involved in cell stress, apoptosis, inflammation, oncology, other neurodegenerative diseases (PMID: 20079433) it may be premature to claim that MAPK2 is a therapeutic target and more needs to be teased out to determine if the MAPK pathway is the cause or effect of disease pathogenesis (next point below). Could the authors compare the phosphoproteomic and transcriptomic profiles of the ALS mouse models to a simple stress model (e.g. LPS) as a control to determine if these changes are ALS-specific or stress-specific. This is relevant to the trametinib experiments where Nfs are reduced in concentration and is this the result of treating the stress pathway rather than the underlying disease mechanisms?

- It was intriguing that pathways involved with RNA metabolism, protein folding, and protein degradation were not observed in human samples as potentially impaired pathways as the literature has suggested that these are affected in ALS (PMID: 28512398). Could the authors comment on potentially why they think these were not observed as major pathway changes (instead of individual proteins) in either the human samples and/or mouse models?

- This is more a structural comment but there is great data presented in this paper but the written results refer to the Figures in an odd order which I think the Figures should either be re-arranged or the written sections moved around. E.g. Fig 1 is referred in "Cohort composition" section with Fig 2 referred to as well, and then jumps back to Fig 1 in the next section with Fig 3 mentioned here with no mention of Fig 2. In "Transcriptomes of murine ALS.." refers back to Fig 2 and 3. This made it tricky reviewing the paper and should be fixed prior to publication to improve clarity and readability for a reader.

- The finding that there were changes to miRNA hairpins particularly in males is intriguing, was there any convergence with regards to the TFs that are responsible for these? The regulated genes from the miR screen show components of the MAPK pathway (MAPK, AKT1, BCL2) which suggests that the effect on this pathway is downstream from the mechanism of disease. Was there a common TF that is responsible for the pre-processing of the miRs hairpins?

- The authors carried out proteomic screening of the different mouse models and found that the TDP-43 M337V model most closely resembles the human post-mortem proteomic data, what was the genetic status of the human patients obtained? Were they familial or sporadic patients or both? Were the ALS patients pure ALS or did they have co-morbidities with FTD or other diseases? This may bias the interpretation as SOD1 and C9ORF mice tend to develop symptoms later on in disease (PMID: 25977373)

- This might be beyond the scope of the multi-omic study but was there any behavioural or motor testing of the ALS mouse models treated with trametinib? Or has another analog of trametinib such as dabrafenib (or combination therapy) been tested? Have the authors carried out a phosphoproteomic screen to assess the off-target effects with trametinib treatment of primary neurons since if MEK is blocked I would assume the other converging MAPK pathways may compensate for this.

- Minor – AGC targets spelling mistake should be changed to 3×10^6 ions, currently reading as 3106 for MS1 and 1105 for MS2 in the paper

- Minor – Have the authors considered using iPathway or IPA (realising that these are commercial products) to re-analyse their omics data to make more accurate predictions as an alternative to GSEA and WGCNA which are mostly ways to summarise data.

I thought this was a great study incorporating different multi-omic technologies and the authors acknowledge the limitations of the current study and other avenues that are out of scope for this paper. They provide a comprehensive characterisation from the RNA to proteomic level of 100 tissue samples and 4 mouse models and I feel that the results will contribute to both the ALS and proteomics fields.

Reviewer #3 (Remarks to the Author):

Amyotrophic Lateral Sclerosis (ALS) is a progressive fatal adult-onset neurodegenerative disorder characterized by the selective loss of lower and upper motor neurons as well as muscle degeneration. Most forms of ALS are sporadic with only 10% of the cases being inherited in a dominant manner (familial ALS). The four most common causative genes are C9orf72, SOD1, FUS and TDP-43. ALS is a devastating and highly heterogenous disease, for which there is currently no cure. It is thus critical to elucidate whether and if so which pathways are commonly dysregulated during pathogenesis to advance progress toward effective therapeutic development.

To discover potential novel, critical targets that are affected during the disease, Gomes, Hanzelmann and colleagues performed a multiomic analysis of a large cohort of postmortem human brain samples (in particular the prefrontal cortex, a region proposed to be less affected at end-stage, of n=51 ALS patients and 50 control non-neurological controls), as well as of four transgenic mouse models for C9orf72 (polypeptide expression)-, SOD1 (G93A)-, TDP-43 (M337V)- and FUS (WT)-ALS. In particular they analyzed the human and mouse samples using bulk transcriptomics, proteomics and miRNAomics and ultimately integrated all the datasets. The authors describe multiple dysregulated pathways, that are proposed to be gender specific. They further focus on the mitogen-activated protein kinase (MAPK) pathway, and more specifically one target, mitogen-activated protein kinase kinase 2 (MAP2K2 or MEK2) that they propose as an early disease-relevant mechanism. Treatment of ALS mice (or cultured neurons) expressing mutant SOD1 with trametinib, a drug inhibitor targeting MEK2 is reported to attenuate mutant-SOD1 associated pathological hallmarks including SOD1 insolubility, in females further leading the authors to propose MEK2 as a potential therapeutic target to treat ALS.

This study provides an extensive analysis of the molecular changes occurring in ALS in a large cohort of human patients and mouse models. The use and integration of multiple omics approaches of such a large number of samples is certainly impressive and has the potential of being of interest for the scientific community when freely available. Perplexingly, after such a large effort, the authors focus on the MAPK pathway, for which there are already several studies proposing its potential relevance in ALS (Sahana and Zhang 2021, Biomedicines; Kim and Choi 2010, Biochimica et Biophysica Acta; Gibbs et al. 2018, Nature Cell Death and Disease; Pérez-Cabello et al. 2023, PNAS), further limiting the novelty of the findings. Hence, trametinib, a MEK2 inhibitor, is currently being tested in a clinical trial conducted in South Korea.

While the study has the potential to provide valuable insight for disease mechanisms, in the present

format there are several critical shortcomings including the execution of the validation of the targets which is weak, the analysis/interpretation of the datasets (the strong bias in numbers of male samples (n=35) in the ALS cohort that were analyzed compared to females (n=16) which should be taken into consideration when interpreting the data and concluding that there are more pronounced changes in males) and the highly confusing assembling/presentation of the data/figures.

Overall, the sum of the current effort is confusion as to which specific molecular events are critical for disease onset and progression. Without substantial modifications, the manuscript is not appropriate for publication.

Main Concerns:

- The flow is very difficult to follow as the figures are currently organized by techniques and not the overall scientific outcomes (findings). This makes the understanding/reading very confusing as the description of the data is not done in the order in which the figures are currently assembled. Effort should be made to reorganize the content of the main figures so that it is easier to follow the experimental flow and findings. One single paragraph should have the corresponding main figure described based on the scientific outcome (and not the technique- for example currently Figure 1 is composed of transcriptomics data, Figure 2, proteomics..etc..). On this note, the main important missing figure of the paper is the one recapitulating all the findings pointing to MEK2 as the main target to which the authors directed their attention on (corresponding to the paragraph at line 294). This important point is missed in the text, since data is embedded within different main figures with fragmented and weak information throughout the paper.
- The strong justification mentioned in the main text for the initial separation of the available transcriptomic datasets based on sex is weak. The difference which leads the transcriptomic separation of samples in males and females is only 15%. Indeed, in the PCA plot in Suppl. Fig. 5 (which was actually saved under Sup. Fig 4), some samples derived from males and females are juxtaposed. It would be valuable to perform unsupervised clustering of transcriptomes (instead of enriched pathways) of all the samples to justify the separation between males and females. This would be more convincing. Moreover, the WGCNA analysis performed in Figure 1e shows consistency between male and female samples when divided by clusters, and not differences, which does not support the conclusion the authors put forward about the sex-difference. Similarly, in Figure 4g, there is no clear clustering of male samples separated from female samples on the UMAP analysis, which again does not support the claim made. On the contrary, male samples seem to be clustered in two sub-clusters. Moreover, in Figure 4h, factors cannot be compared between male and female samples because the analysis was done separately resulting in different genes representing the specific factors and different correlation scales. The data representation in the current format is misleading. Altogether, the evidence presented does not demonstrate that there is a sex-difference as the authors conclude.
- More concern is directed to the data in Suppl. Fig. 5 where the clear separation between samples derived from ALS patients and control subjects is lacking. For this reason, in Figure 1b, the differences between ALS and control samples, especially in female underrepresented samples, are very subtle. Interestingly, in one of the studies mentioned by the authors (Aronica et al. 2015, Neurobiology of Disease) where less samples are included, the number of differentially expressed genes was very much

higher compared to the one in the present transcriptomic dataset. The authors should discuss this observation. For the separate analysis of males and females, the authors should generate two separate PCA plots with ALS/control female samples and ALS/control male samples.

- The fact that human sample clusters correlate partially with mouse model 'omics datasets do not imply that one mouse strain is modelling one specific human cluster in this study. The correlation is very minor and should be discussed as such. Additionally, in Figure 2d, C3 samples resemble more the mouse models respect to the other human clusters, making this analysis rather weak. In this figure, the authors should report the Pearson correlation coefficient per comparison in the heatmap.
- Supplementary Figure 12 reports transcriptomics changes of genes related to MAPK pathway which show great variability and overall stable expression across groups (white color). This represents the first evidence in support (according to the authors) to the involvement of MAPK signaling in the disease pathogenesis of mouse models and human patients. This conclusion is weak and appears in the text without a clear rationale (line 186-189). Moreover, in main Figure 2e, it is suddenly reported the log₂ FC of MAPK pathway genes in ALS cluster 1 and 3. The authors should report also the log₂ FC of these genes for cluster 2 and 4. Moreover, the size effect is subtle, the log₂ FC scale goes from 1 to -1, which further makes this referee wonder why the authors focused on this target (given that it was not novel).
- The rationale for the prioritization of miR-451a by miRNA expression analysis to support MAPK pathway involvement in ALS is very weak. The authors propose in the main text that all the mouse models and human patients showed deregulation of miR-451a, but from the data presented, only human males, SOD1 male/female mice and C9orf72 female mice suggest such effect. The effect is reported to be significant only in human males and C9orf72 mouse females.
- Finally, to assess the preclinical relevance of the chosen treatment trametinib, behavioral tests and survival curves to assess disease phenotype/course should be performed.

Additional points:

- C3 and C4 clusters in Figure 1c show their heterogeneity, and RNA splicing seems not to be a strong factor of their segregation.
- Supplementary Figure 4 f-g-h missing. RIN analysis should be reported also for human samples in order to exclude defective samples from the analysis.
- In Supplementary Figure 7, it is not clear which genes are showed with the corresponding color. Colors seem to refer to more than the 50 selected ones. Moreover, from this visualization it is clear that some samples do not belong to the assigned clusters defined by Tam et al. (2019).
- Xpo1 shows upregulation also in C9orf72 and SOD1 females. It is not reported if the difference in fold change between males and females is significant/relevant. Importantly, the authors highlight the fact that Xpo1 is a target in an ALS clinical trial (line 251-253), but in human patients the effect is not present.
- In figure 5, the authors should use the antibody for phospho-MEK2 in primary neurons to assess activation of MEK2 in addition to phospho-Erk1/2 readout. Total MEK2 protein levels should be used as normalization factor.
- In phosphorylation analysis through western blot, the normalization signal is always the total protein level of the corresponding protein, not the loading control (in this case lamin b should be reported but

not used as normalization factor, instead the authors should use a total Erk1/2 antibody).

- The performed western blots should be reported in the main figure with the relative quantifications in Figure 5. The uncropped versions of the images should be reported in the supplementary material.

Style/formatting concerns:

- Table titles/legends are missing. It was extremely difficult to understand which supplementary figure was what given that none have a title, and on top of that some were incorrectly saved (only the pdf files were labeled but there were inversions). For the eps files there were no labels so this referee had to guess based on the summary figure legends. This referee strongly recommends writing on each Supplementary figure, their number.

- The authors should include titles and legends in all the heatmaps and graphs to help the reader understand the architecture of the data. Where possible, resize them to make them more readable (often it is too small to read all the text).

Reviewer #4 (Remarks to the Author):

In this study, Gomes and colleagues generated a variety of omics data on postmortem prefrontal cortex tissue from 51 ALS patients and 50 control donors, as well as on 4 well-known mouse models of ALS genetic mutations. They observed clear sex differences in both mice and humans, and were able to cluster the human patients into 4 subtypes, reminiscent of previous studies on an independent post-mortem cohort. By integrating their data together, they prioritized the MAPK signaling pathway and demonstrated that an inhibitor of MAP2K2 rescued the effects of excitotoxicity on primary cortical neurons, again with a sex-specific effect. Overall, they did a thorough and careful analysis and their findings will be useful for the field. However, I found the paper and methods hard to follow and missing some important technical details.

Given that ALS is associated both genetically and clinically with frontotemporal dementia, is it possible that the transcriptomic differences observed in the frontal cortex that separate ALS donors are caused by concomitant FTD in a subset of those donors? Did the authors check whether their donor patients had cognitive decline or FTD symptoms before death?

The sequencing methods list two different library preparation kits used for mRNA-seq - one polyA enriched and one for total RNA with ribodepletion. Library preparation usually has a large effect on gene quantification. Was this library batching balanced across disease groups? Was this adjusted for in differential expression analyses?

How exactly was differential expression performed? Which package was used? What covariates did they include? Please include this.

Given that the clusters overlap significantly between males and females, did the authors not attempt a

ALS vs Control differential expression using all the samples while adjusting for sex? This should boost their discovery power.

The authors should clarify in the main text what thresholds were used for selecting genes for pathway enrichments.

My biggest source of confusion was understanding how exactly clusters C1-C4 were defined. The text describes them as arising from hierarchical clustering of enriched GO/KEGG pathways, but how is that converted into an “activity score” for each ALS patient? Fig 1c mentions “Activity scores calculated by Decoupler”, but there is no explanation of this or citation in the methods.

When presenting WGCNA modules, the convention is to just use the colour names - Mturquoise should be turquoise, etc.

Splicing results are out of order in the results. Should be discussed after Figure 2, could be joined with the miRNA work. Fig 3d in results is linked to TPRN, which is not actually present in that figure - instead a set of pathway enrichments which does not appear to be discussed in the results.

In the splicing analysis they state that cryptic splicing events “are rarely detectable in bulk RNA-Seq at our sequencing depth”. However, several papers have shown that cryptic splicing in STMN2 and UNC13A can be detected in bulk tissue RNA-seq (Prudencio, JCI, 2020; Brown, Nature, 2022, Ma, Nature, 2022). Why did the authors not attempt to at least quantify STMN2 and UNC13A in their samples?

Deconvolution results are better presented in the main text as box-plots rather than the heatmap used in Figure 2. But even the boxplots in the Supplemental figure are confusing as they omit the control humans and non-transgenic mice as a comparison. They also don’t appear to correspond exactly - endothelial cell proportions are clearly increased in C3 samples, but that is not reflected in the heatmap. Is this why?

Fig 2e - how many of the MAPK genes are significant after multiple testing?

Fig 3 - Why was the miRNA analysis not linked back to the C1-4 patient clusters?

Supplemental Figures are lacking keys for colour gradients. Additionally, supplementary figures are mismatching the numbers on the reviewer portal. Would the authors please upload all supplemental figures in a single document, with the corresponding legends paired with each figure?

REVIEWER COMMENTS

Reviewer #1 (Remarks to the Author):

Manuscript background information: Amyotrophic lateral sclerosis (ALS) is a degenerative motor neuron disease that is more prevalent in males than females and has no effective treatment for sporadic cases (sALS). This study focuses on identifying and characterizing the sex-distinctive multiomic changes induced by early disease progression in ALS by comparing postmortem prefrontal cortices. Additionally, changes in the multiomic landscape of four transgenic mouse models of c9orf72- SOD1-, TDP-43-, and FUS-ALS were analyzed individually and in context of the human samples. From the transcriptomic dataset, four subclusters of ALS were identified. The mitogen-activated protein kinase (MAPK) pathway was also identified as a potential pathway key to early disease progress, and its modulation by trametinib showed promising therapeutic effect for female SOD1 transgenic mice.

Remarks to the Author: The integrated multiomic dataset encompassing both human and transgenic mice models that is the basis of this paper is interesting, though it is possible that one or more of the omics from the human samples have previously been analyzed since they are derived from banked samples. While it is of note that molecules implicated in the MAPK pathway regularly appear in the multiomics data analyses, this is not a novel pathway to be identified for ALS (1), and the sex-selective differences in the trametinib results may not justify the novelty requirements for this journal. Additionally, the provenance of the foundational data for the four subclusters of ALS is not fully explained (see below). I also suggest that because of the rich nature of the methods and results for this paper, it would be better placed in a journal where they could be fully explored.

Reference:

Sahana TG, Zhang K. Mitogen-Activated Protein Kinase Pathway in Amyotrophic Lateral Sclerosis. *Biomedicines*. 2021 Aug 6;9(8):969. doi: 10.3390/biomedicines9080969. PMID: 34440173; PMCID: PMC8394856.

Major comments:

Q1 - The study design is questionable. The main cells affected in ALS are the Betz cells located in the primary motor cortex. These cells are unique to higher mammals and are absent from rodents. A single-nuclei approach would have permitted to study the difference between males and females in humans (even if most are depleted, the remaining cells are sufficient to make such analysis). Deconvolution does not allow such resolution and accuracy. There are other groups of cells in the motor cortex affected in ALS and also present in mice. Comparing these would have been more relevant and interesting for the field.

A: We appreciate the comments on our study design and we agree with this reviewer on the importance of Betz cells in ALS. However, our decision to study a cell population distinct from Betz cells was taken very explicitly and based on well-supported assumptions. As this reviewer points out, Betz cells are affected early in the progression of disease. Since we only have the possibility to study human brain tissue in post-mortem samples, studying the few remaining Betz cells would imply studying the most affected region at the time of death. We were interested in identifying disease-relevant mechanisms and studying the most affected region would likely provide us with alterations that represent disease end-stage - such as gliotic changes, degenerated neurons and a strong immune system activation. Similar analyses have been previously performed (e.g., Dols-Icardo et al. *Neurol Neuroimmunol Neuroinflamm* 2020). Other regions that are more distant from the motor cortex, such as the prefrontal cortex (PFC), were also shown to be affected, but later in disease (Brettschneider et al. 2013 *Ann Neurol*, Braak et al. 2013 *Nat Rev Neurol*). To use single nucleus sequencing in our study is a great yet very costly suggestion, which we contemplated several times. In this study, therefore, we opted to first investigate large numbers of patients and mouse models with bulk transcriptomics and deconvolution. Specific findings can now be substantiated and further analyzed using single cell technologies, involving spatial transcriptomics and

proteomics, in the future. We therefore believe that studying the PFC in ALS is both novel and useful, adding new and multi-layered information to data that has been previously derived from the motor cortex.

Q2 - The clustering of human ALS samples was performed on the basis of activities scores across seven pathways. It is not clear to me how these pathways were selected and the means of acquiring the activities scores is relegated to identifying the R package (no version, no parameters, no input data). Because this clustering is foundational to the paper, these methods are critical and should be provided in full detail.

A: We thank the reviewer for this important comment and apologize for the lack of detail. It is indeed crucial to explain the selection of the ALS-related pathways used for the clustering analyses (Fig.1f). The selection of the pathways used here was based on the enrichment analyses performed with the transcriptomics dataset from the human samples, which was the basis for our clustering analyses. As detailed in our methods, enrichment analyses for the RNA sequencing dataset for humans included gene ontology (GO) - *gseGO* and *gseKEGG* analyses [from the *clusterProfiler* R package - Supplementary table 7; Fig 1e], as well as a co-expression network analysis (weighted gene co-expression analysis - (WGCNA) analysis (Supplementary Table 8). We went through the lists of enriched pathways systematically, analyzing the top 30 most highly enriched terms for each dataset, and then grouped these terms by more general umbrella terms, to select ten pathways that summarized the enriched terms best: **mitochondria/respiration; synapse; MAPK cascade; oxidative stress; nucleocytoplasmic transport; protein folding/metabolism; lipid metabolism; RNA splicing; extracellular matrix and activation of immune response.**

In detail, for overlapping KEGG pathway and GO results, we observed a predominant enrichment for **extracellular matrix (ECM)** and **immune response** pathways and **synapse-related** terms for both males and females. The convergence of these themes across sexes suggests common biological themes relevant for the disease and justified their selection for the clustering analyses.

In addition to neurodegenerative disease pathways driven by DE genes involved in **protein metabolism** and **oxidative stress** mechanisms (Supplementary Fig. 5), female specific results showed enrichment for **aerobic respiration** (GO) and **ribosome** (KEGG)(Fig. 1e), while **protein metabolism** was captured for male results (KEGG)(Fig. 1e). **Oxidative stress** was also inferred from the enrichment of **oxidative phosphorylation** pathways in both males and females (KEGG).

WGCNA results revealed key associations with all of the selected themes (Supplementary Table 8). In Fig. 1h, this is depicted in the top enrichment results for **mitochondria/respiration and protein metabolism** (1, turquoise); **synapse** (2, yellow), **immune response** (3, tan) and **RNA splicing** (4, lightcyan). Further pathways were frequently enriched for relevant WGCNA modules (Supplementary Table 8): terms related to **nucleocytoplasmic transport** were frequent for module *turquoise*, while **lipid metabolism** and related terms appeared frequently for the *Tan* module, as well as for the *Black* and *Blue* modules. WGCNA results were also frequently enriched for **MAPK-related terms** (Supplementary Table 8; Fig. 1h; Fig. 4a). The MAPK cascade emerged as a recurring theme not only for RNA-sequencing based analyses, substantiating its relevance throughout our study, and justified its selection for the clustering analyses. All selected pathways and full clustering results are depicted in Supplementary Fig. 7).

Regarding the second part of the reviewer's question, we have also now added a more detailed explanation about the acquisition of the activity scores to the methods section, as requested. This can be found in **lines 896-919**, as follows:

“Pathway selection for clustering analyses with transcriptomics datasets

The selection of the pathways used for clustering analyses of human samples was based on the enrichment analyses performed with the transcriptomics dataset. We selected the top hits from gseGO and gseKEGG analyses [from the clusterProfiler R package], as well as a co-expression network analysis (weighted gene co-expression analysis - (WGCNA) analysis, analyzing the top 30 most highly enriched terms for each dataset, and then grouped these terms by more general umbrella terms, to select ten pathways that summarized the enriched terms best: mitochondria/respiration; synapse; MAPK cascade; oxidative stress; nucleocytoplasmic transport; protein folding/metabolism; lipid metabolism; RNA splicing;

extracellular matrix and activation of immune response. In more detail, for overlapping KEGG pathway and GO results, a predominant enrichment for extracellular matrix (ECM) and immune response pathways and synapse-related terms was observed for both males and females. The convergence of these themes across sexes suggests common biological themes relevant for the disease and justified their selection for the clustering analyses. In addition to neurodegenerative disease pathways driven by DE genes involved in protein metabolism and oxidative stress mechanisms (Supplementary Fig. 5), female specific results showed enrichment for aerobic respiration (GO) and ribosome (KEGG), while protein metabolism was captured for male results (KEGG). Oxidative stress was also inferred from the enrichment of oxidative phosphorylation pathways in both males and females (KEGG). WGCNA results revealed key associations with all of the selected themes, especially for terms related to mitochondria/respiration and protein metabolism, synapse, nucleocytoplasmic transport, lipid metabolism; immune response and RNA splicing. WGCNA results were also very frequently enriched for MAPK-related terms (Supplementary Table 8), justifying its selection for the clustering analyses.

Identification of transcriptome-based sub clusters in human ALS patients

*We used decoupleR (v1.1.0) 76 to aggregate scores from gene sets derived from GO terms of interest. Gene lists with the components of GO terms of interest were extracted using the AmiGO2 database (<https://amigo.geneontology.org/>). For the unsupervised per sample enrichment analyses, we used decoupleR with the consensus from the *mlm*, *ulm*, *wsum* functions76 and calculated activity scores. These are numerical values associated with genes and represent gene expression levels.”*

Q3 - The supplementary files are not of the same quality as the figures in the paper. The order of the files is inconsistent with the figure legends file (making it hard to review), there are subplots mentioned in the figure legend that are not present (e.g. SI Fig 4), or potentially missing (SI Fig 9, though hard to tell because ordering was off). Below is an additional sampling of what I found (though by no means comprehensive):

A: We apologize for the inconsistency in the supplemental material. We have now completely reworked the supplementary figures to assure that they maintain the same quality as the main figures of the paper. All figures were revised and re-exported with high quality. The order of the figures and all the figure legends were revised. Furthermore, following the suggestion of the reviewers, we have now created a separate file for the supplementary figures. All the legends now appear directly after each SI figure, facilitating the navigation through the content. In addition, we have reduced the number of supplementary figures following another reviewer's request. The new version of the supplementary figures can be found in the supplemental file entitled "SI figures and captions".

a) SI Fig 11 e) ANOVA is likely more appropriate than multiple t-tests here

A: Thank you for your suggestion. We initially used t-tests for pairwise comparisons, but in response to your input, we conducted ANOVA. Subsequently, we applied Tukey's range test due to significant differences among human clusters. Additionally, in accordance with the feedback from reviewer 4, we included the CTRL group in the figure and separated human results for clarity. We also decided to show the mouse results in the barplots in Supplementary Figure 14e, for comparability reasons. The corresponding plots have been incorporated into the current Supplementary Figure 14.

b) SI Fig 2 Euclidean distance should be 0 between sample and itself – scale and legend appear to be incorrect

A: In response to this comment, we have enhanced the figure caption to provide additional details. While the Euclidean distance between a sample and itself is inherently 0, the scale in our figure is specifically tailored for comparing distances between distinct samples. To facilitate a clearer interpretation of differences between various samples, we have masked the self-distance (diagonal) from the color scale. The figure legend in Supplementary Figure 2 has been accordingly modified:

New figure legend, Supplementary Fig. 2:

Heatmap showing the euclidean distance between all samples based on the vst-transformed transcriptomics data. High values (light colors) indicate a big difference between the samples. The condition, the sex, the brain bank, the age at death and the genetic phenotype of each sample is indicated above and on the left side. Samples are clustered according to their distance using hierarchical clustering. The diagonal (self-distance) is zero and masked from the color scale to allow a better distinction of the other sample-distances.

c) SI Figure 26 labeled as male in title when should be female?

A: Thank you for pointing this out, we apologize for this mislabeling. The error was now corrected in the respective figure legend (now Supplementary Figure 12).

Q4 - (4) The computational methods are incomplete. While a “Reproducible data pipeline” is mentioned in the methods, it does not appear to have been made available. Additionally, there are multiple instances where critical parameters to run the code have not been provided and the package versions have not been provided. There is also no code provided, just links to some of the packages/tools utilized. This lack

of detail also extends to the statistics used to analyze the in vivo mouse data, where the post hoc test for the ANOVAs run is not mentioned.

A: We are sorry that we didn't provide all necessary code and analysis details in the first submission. We have included a link to the code in the revised document (https://github.com/imsb-uke/MAXOMOD_Pipeline) and also extended the statistics section in the methods to include version and parameter information that were missing. The corresponding figure legends were also updated accordingly (**lines 1017-1021**):

“Statistical analyses of in vivo experimental data were performed using GraphPad Prism 7.0 and statsmodels Python package (v0.13.0). Student's t-tests or one-way ANOVAs, followed by pairwise TukeyHSD post hoc tests, were used to analyze the differences between the experimental groups for each variable. Statistical significance was set at $p < 0.05$.”

Q5 - (5) The enrichment methods and their description also could be improved. Throughout the paper, GSEA, ORA, and some graph-based methods (to a lesser extent) are used to identify pathways and processes that are of interest. It is not always clear which of these methods is being used in what context. This becomes an issue, because in multiple analyses, there appears to be only one differential molecule supporting the identification of a pathway/term (particularly for the female samples).

A.: We apologize for not having been clear enough with the description of our enrichment methods. Therefore, we now extended the methods part of the enrichment analyses, mentioning the tools that were used for each dataset/context. Similarly, we included the name of the enrichment algorithms in the main text, to avoid confusion. These changes can be found in lines **863-894** of the main methods.

“Enrichment analyses

Gene ontology, pathway enrichment analyses, and protein interaction networks

GSEA was performed using gseGO and gseKEGG from the clusterProfiler R package, with biological processes and molecular functions chosen as background databases for GO enrichment. The p-value cutoff was set at 0.05. Differential expression was presented using volcano plots generated with the Enhanced Volcano plot package in R, and over-representation analysis was performed on genes that showed at least one significant DAS event using the clusterProfiler function enrichGO, with a p-value cutoff of 0.1 and Benjamini–Hochberg correction for multiple hypothesis testing. Protein–protein interaction networks were created using the STRING protein interaction network database version 11.067,68 using standard settings. We summarized the enriched GO terms by clustering pathways based on their descriptions, which is called clustering in semantic space using GO-Figure (PMID: 36303779). In addition, for proteomics results, we performed additional clustering analyses using REVIGO (PMID: 21789182). This tool allows GO term clustering by hierarchy, considering the following such as enrichment p-adjusted values, semantic similarity and term proximity.

Weighted Gene Correlation Network Analysis

To conduct a weighted gene co-expression analysis, we employed the WGCNA package¹⁵. Pairwise Pearson correlations were calculated to establish signed regulatory networks within the WGCNA. By constructing an adjacency matrix, we applied a soft-thresholding technique to approximate a scale-free topological network. Eigengenes or eigenproteins were calculated as the first principal components of each module. This resulted in the development of several modules. We merged similar modules based on hierarchical clustering (SOD1: 0.4, C9orf72: 0.4, FUS: 0.5, TDP-43: 0.4 for RNA-Seq and 0.25 for proteomics). We calculated the relationships between WGCNA modules and traits. Sex was also included in the traits resulting in four traits: male ALS, female ALS, male CTR, and female CTR. First, we filtered modules based on the significance of the module–gene relationship ($p < 0.05$) and then selected modules that were highly correlated with either male or female ALS. The correlation cutoffs differed between the mouse models (SOD1: 0.5, C9orf72: 0.3, FUS: 0.3, and TDP-43: 0.5 for RNA-Seq and SOD1: 0, C9orf72:

0, FUS: 0, and TDP-43: 0 for proteomics). Using WGCNA, we also analyzed co-expression networks in mouse models. The minimal module size was set to 30 with a merge height of 0.4–0.5 and a correlation threshold of 0.3–0.5.”

Reviewer #2 (Remarks to the Author):

The paper by Gomes et al titled “Multiomic ALS signatures highlight sex differences and molecular subclusters and identify the MAPK pathway as a therapeutic target” performs a multi-omic study of transcriptomics, miRNAomic and proteomic (including phosphoproteomics) levels of 51 prefrontal cortex patients (split 70/30 males/females) and 50 controls (split 44/56 males/females) and compared these to four transgenic mouse models (C9orf72, SOD1, TDP-43 and Fus-ALS) to determine if there were changes in the early disease mechanisms and whether sex had an influence. In human patients, transcriptomic level changes were observed in genes clustered to ECM, mitochondrial dynamics and RNA metabolism which was recapitulated in the mouse models and determined that the MAPK pathway was affected in disease and in models with a potential therapeutic target being MAPKK2. The study itself is a very elegant study and provides excellent coverage across multiple omic platforms from a human source which is crucial in ALS research given that much of the neurons that are affected by ALS are already gone by the time post-mortem tissue is obtained revealing only those that are still alive. The mouse models provide detail into the early disease mechanisms that may lead up to neurodegeneration and the fact that MAPK pathway was identified as the main pathway involved early disease. My comments are below:

Q1 - MAPK pathway was identified as the early onset pathway in disease, how disease-specific is this pathway i.e. is it due to ALS or due to a general stress response? Given that MAPK is highly involved in cell stress, apoptosis, inflammation, oncology, other neurodegenerative diseases (PMID: 20079433) it may be premature to claim that MAPKK2 is a therapeutic target and more needs to be teased out to determine if the MAPK pathway is the cause or effect of disease pathogenesis (next point below). Could the authors compare the phosphoproteomic and transcriptomic profiles of the ALS mouse models to a simple stress model (e.g. LPS) as a control to determine if these changes are ALS-specific or stress-specific. This is relevant to the trametinib experiments where Nfs are reduced in concentration and is this the result of treating the stress pathway rather than the underlying disease mechanisms?

A: We thank the reviewer for these insightful considerations on the specificity of MAPK pathway regulation. Concerning the suggestion to compare the phosphoproteomic/transcriptomic profiles of the ALS mouse models to a simple stress model such as LPS aiming to check if MAPK pathway changes are ALS-specific or stress-specific, it has been well documented that LPS activates members of the mitogen-activated protein kinase (MAPK) family, and increases phosphorylated extracellular signal-regulated kinases (ERK)1/2 in multiple cells types, including microglia. Below, we cite from some of these publications:

- 1) LPS stimulation increased MAP kinase expression in intestinal epithelial cells (Talavera et al, 2015 PMID: 25950450) :

[REDACTED]

2) LPS induction of MAPK activation in bone marrow-derived macrophages (An et al, 2018, PMID: 29354064):

3) LPS-induced phosphorylation of ERK and p38 MAPK in BV-2 microglial cells (Wang et al, 2011, PMID: 21831303):

Because LPS-induced regulation of the MAPK pathway has been published in multiple instances, new animal experiments would not yield novel information and according to the 3R principles, we would not receive authorization for such work from the Animal Ethics Committee. We hope to have clarified that performing injection of LPS in mice will not bring new information in comparison to what has already been demonstrated previously. The question of specificity of the MAPK pathway is indeed intriguing - and in our view somewhat philosophical. In a pure sense, it is an activated pathway and if the regulation of this pathway occurs in disease and its attenuation results in beneficial effects, it represents a viable therapeutic target. In our view, this is independent of the fact that the very same pathway could be activated in other circumstances. The same would be true for many other pathways, e.g., oxidative stress,

mitochondrial dysfunction, immune response activation - all these pathways may be considered not very specific, but all of them are explored or used as therapeutic targets in neurodegenerative disorders.

For the above reasons, we do not postulate that the alterations of the MAPK pathway that we identified in our multiomic study are strictly specific to ALS. Our observations capture changes in stress pathways that are abnormally regulated in patients with ALS - as well as in ALS mouse models. As this reviewer mentioned, MAPK is involved in cell stress, apoptosis, inflammation, and neurodegeneration. All these processes are implicated in ALS pathophysiology. Using trametinib, we first show that blocking the MAPK pathway improves the survival and neurite growth in glutamate-stressed culture cells, thus having an impact on two important processes relevant for ALS. Second, we proved that SOD1 female mice treated with trametinib demonstrate decreased activation of MAPK/ERK pathway, accompanied by a significant decrease in neurodegeneration (NfL) and protein aggregation (Triton-insoluble SOD1 and ubiquitinated proteins) likely by activation of autophagy pathway (decreased p62), as demonstrated by Chun et al., 2022. In addition, we demonstrate that disease onset is significantly delayed and survival is significantly improved in trametinib-treated animals compared to controls. These results add substantial evidence in favor of trametinib as a disease-modifying drug in ALS. We thus conclude that this particular (stress) pathway is abnormally activated in ALS, and that restoring this pathway with trametinib might be beneficial for the course of the disease. We believe that the discussion of specificity and suitability as a therapeutic target is important and thank the reviewer for pointing this out. We therefore also appended the discussion on this topic (**lines 465-467**): *“Although the MAPK pathway may represent a less specific answer of the cell in response to stressors, its involvement in ALS makes it a promising target, which could add another step in a combination therapy.”*

Q2 - It was intriguing that pathways involved with RNA metabolism, protein folding, and protein degradation were not observed in human samples as potentially impaired pathways as the literature has suggested that these are affected in ALS (PMID: 28512398). Could the authors comment on potentially why they think these were not observed as major pathway changes (instead of individual proteins) in either the human samples and/or mouse models?

A: We acknowledge the importance of pathways associated with RNA metabolism, protein folding, and protein degradation in ALS. In fact, we do find these pathways enriched in our data, and therefore they were also selected among the main pathways of interest for our clustering analyses (Fig. 1f). For your reference, a new methods section to cover the rationale for the selection of pathways of interest can be found in **lines 896-928**, and it was also answered in detail in Q2 of reviewer #1.

In regard to the pathways mentioned in this question, our clustering analysis (Fig. 1f) revealed that the comparison of C3 vs. C4 shows differences in synaptic function and **protein folding**. Enrichment results for transcriptomics data also revealed the term **protein metabolism** for male samples (KEGG). Furthermore, we observed an upregulation of differentially expressed genes associated with pathways in **neurodegeneration** (Fig. 1e), especially for females. This enrichment was mainly driven by DE genes involved in **protein metabolism** and oxidative stress mechanisms, and are now depicted in a new supplementary figure (Supplementary Fig. 5). WGCNA results were enriched for further important pathways that included **protein metabolism, ribosomal function, nucleocytoplasmic transport, lipid metabolism and RNA splicing** (Fig. 1h; Supplementary Table 8), which also correlates with the pathways mentioned here. Finally, we also approach those topics in the discussion, where we write: *“Integrated analysis of multiomic data sets has identified several dysregulated pathways relevant for ALS, including mitochondrial respiration/oxidative stress, transcriptional regulation/splicing, and protein misfolding/degradation.”*

A detailed description about how the clusters were selected is included now in the methods section (**lines 896-928**):

“Pathway selection for clustering analyses with transcriptomics datasets

The selection of the pathways used for clustering analyses of human samples was based on the enrichment analyses performed with the transcriptomics dataset. We selected the top hits from gseGO

and gseKEGG analyses [from the clusterProfiler R package], as well as a co-expression network analysis (weighted gene co-expression analysis - (WGCNA) analysis, analyzing the top 30 most highly enriched terms for each dataset, and then grouped these terms by more general umbrella terms, to select ten pathways that summarized the enriched terms best: mitochondria/respiration; synapse; MAPK cascade; oxidative stress; nucleocytoplasmic transport; protein folding/metabolism; lipid metabolism; RNA splicing; extracellular matrix and activation of immune response. In more detail, for overlapping KEGG pathway and GO results, a predominant enrichment for extracellular matrix (ECM) and immune response pathways and synapse-related terms was observed for both males and females. The convergence of these themes across sexes suggests common biological themes relevant for the disease and justified their selection for the clustering analyses. In addition to neurodegenerative disease pathways driven by DE genes involved in protein metabolism and oxidative stress mechanisms (Supplementary Fig. 5), female specific results showed enrichment for aerobic respiration (GO) and ribosome (KEGG), while protein metabolism was captured for male results (KEGG). Oxidative stress was also inferred from the enrichment of oxidative phosphorylation pathways in both males and females (KEGG). WGCNA results revealed key associations with all of the selected themes, especially for terms related to mitochondria/respiration and protein metabolism, synapse, nucleocytoplasmic transport, lipid metabolism; immune response and RNA splicing. WGCNA results were also very frequently enriched for MAPK-related terms (Supplementary Table 8), justifying its selection for the clustering analyses.

Identification of transcriptome-based sub clusters in human ALS patients

We used decoupleR (v1.1.0) 76 to aggregate scores from gene sets derived from GO terms of interest. Gene lists with the components of GO terms of interest were extracted using the AmiGO2 database (<https://amigo.geneontology.org/>). For the unsupervised per sample enrichment analyses, we used decoupleR with the consensus from the mlm, ulm, wsum functions76 and calculated activity scores. These are numerical values associated with genes and represent gene expression levels."

Q3 - This is more a structural comment but there is great data presented in this paper but the written results refer to the Figures in an odd order which I think the Figures should either be re-arranged or the written sections moved around. E.g. Fig 1 is referred to in the "Cohort composition" section with Fig 2 referred to as well, and then jumps back to Fig 1 in the next section with Fig 3 mentioned here with no mention of Fig 2. In "Transcriptomes of murine ALS.." refers back to Fig 2 and 3. This made it tricky reviewing the paper and should be fixed prior to publication to improve clarity and readability for a reader.

A: We appreciate the positive comments on our data and we regret that the presentation of the results - as mentioned by several reviewers - was not comprehensible. We have now re-structured the entire manuscript to improve clarity, readability, and to provide better guidance to the reader.

Q4 - The finding that there were changes to miRNA hairpins particularly in males is intriguing, was there any convergence with regards to the TFs that are responsible for these? The regulated genes from the miR screen show components of the MAPK pathway (MAPK, AKT1, BCL2) which suggests that the effect on this pathway is downstream from the mechanism of disease. Was there a common TF that is responsible for the pre-processing of the miRs hairpins?

A: This observation is really intriguing and we attempted to identify a common TF that is responsible for the miR hairpin expression. Our analysis of the TransmiR database identified TP53, MYC and STAT3 as influential regulators of multiple hairpin miRNAs (table below). TP53 is especially interesting in this context, as it is a known stress regulator and as such interacts strongly with the MAPK, ERK and JUN pathways (Wu et al., Cancer Biol Ther. 2004). This could indicate an involvement of MAPK in stress regulation, as the reviewer suggested in prior comments. However, the fraction of miRNAs regulated by these TFs is lower than 10% in all cases and therefore none of these TFs is likely to be the single cause of the observed changes in miRNAs.

We have not included these results in the revised manuscript but would be more than willing if the reviewer or editor deems this relevant.

*Wu GS. The functional interactions between the p53 and MAPK signaling pathways. Cancer Biol Ther. 2004 Feb;3(2):156-61. doi: 10.4161/cbt.3.2.614. Epub 2004 Feb 1. PMID: 14764989.

Transcription factor	Number of deregulated miRNAs	Fraction of deregulated miRNAs [%]
TP53	7	8.54
STAT3	5	6.10
MYC	5	6.10
HIF1A	5	6.10
MYOD1	5	6.10
MYOCD	4	4.88
TWIST1	4	4.88
CTNNB1	4	4.88
NFKB1	4	4.88
EZH2	3	3.66
SRF	3	3.66
YY1	3	3.66
KLF4	3	3.66
CEBPB	3	3.66
MYCN	3	3.66
RREB1	2	2.44
SMAD3	2	2.44
RELA	2	2.44
E2F1	2	2.44
SOX2	2	2.44

Q5 - The authors carried out proteomic screening of the different mouse models and found that the TDP-43 M337V model most closely resembles the human post-mortem proteomic data, what was the genetic status of the human patients obtained? Were they familial or sporadic patients or both? Were the ALS patients pure ALS or did they have comorbidities with FTD or other diseases? This may bias the interpretation as SOD1 and C9ORF mice tend to develop symptoms later on in disease (PMID: 25977373)

A: These deliberations are very relevant as the genetic background of the human cohort might explain some of the observed proximities to mouse models of ALS. To this end, we performed a complete

genotyping for ALS-related genes in all human ALS samples (Supplementary Table 2) and we identified only 2 patients that presented ALS-related genetic abnormalities: one C9orf72 repeat expansion was detected in one patient, while another individual carried a pathogenic variant of NEK1 (c.3107C>G, p.Ser1036Ter)¹⁴ (see e.g. Supplementary Fig. 2, and in Supplementary Table 2). This strongly suggests that our human ALS patient cohort is composed almost entirely of sporadic cases, which we highlighted in the revised manuscript when we described the cohort (**lines 92-101; Supplementary Table 1**). We also identified and labeled these patients in our transcriptomic data (Supplementary Figure 2), and did not observe any particular clustering of these samples, so they were not treated differently. Regarding the question about the co-occurrence of FTD among the patients, we have now extracted all available information from the neuropathological reports of our cohort. Among the 51 ALS patients, 40 demonstrated no FTD-ALS pathology, 6 demonstrated mild FTD-ALS pathology and only one presented clear FTD-ALS pathology. FTD-ALS pathology was defined as ubiquitinated TDP-43 inclusions in the cerebral cortex or limbic system as stated in the available pathology reports (Shi et al., *Acta Neuropathol.* 2005). For 4 patients, this information unfortunately was not provided. Limited clinical data is available for these brain bank samples, but none of the patients was noted to have been diagnosed with FTD. Similar to the genetic cases, the patients did not cluster differently regarding FTD-related pathological findings. We have now included this annotation in the heatmap depicted in Supplementary Figure 2.

*Shi J, Shaw CL, Du Plessis D, Richardson AM, Bailey KL, Julien C, Stopford C, Thompson J, Varma A, Craufurd D, Tian J, Pickering-Brown S, Neary D, Snowden JS, Mann DM. Histopathological changes underlying frontotemporal lobar degeneration with clinicopathological correlation. *Acta Neuropathol.* 2005 Nov;110(5):501-12. doi: 10.1007/s00401-005-1079-4. Epub 2005 Oct 13. PMID: 16222525.

Q6a - This might be beyond the scope of the multi-omic study but was there any behavioural or motor testing of the ALS mouse models treated with tramatenib? Or has another analog of tramatenib such as dabrafenib (or combination therapy) been tested?

A: We now include behavioral and motor testing results of female SOD1 mice treated with trametinib in the revised manuscript, as suggested by the reviewer. Based on the 3R principles, it was not ethical to proceed with more in vivo experiments on SOD1 males since we showed already negative evidence (i.e. NfL, p62, insoluble SOD1 and ubiquitinated proteins) of the trametinib treatment effect in males. Same as in the previous cohort, females (n=8 vehicle and n=10 trametinib) were treated from 9 weeks to survival, 2 times a week, with 3 mg/kg trametinib. We monitored body weight, assessed neuromuscular function with grip strength and extension reflex tests in vehicle- and trametinib-treated mice, which revealed no significant changes between experimental groups. However, we observed a significant delay in the age of paralysis and disease onset, and a significantly improved survival, indicating that trametinib has a beneficial effect in SOD1 female mice. These results are now incorporated in the new main **Figure 6i-k**.

We have not used any other analogs of trametinib and, for the reasons mentioned above, such experiments would fall beyond the scope of this revision, as suggested by the reviewer.

Q6b - Have the authors carried out a phosphoproteomic screen to assess the off-target effects with tramatenib treatment of primary neurons? ...since if MEK is blocked I would assume the other converging MAPK pathways may compensate for this.

A: Thank you for this valuable suggestion. As pointed out by the reviewer, we have initially demonstrated by quantifying Western blots that trametinib has an impact on p-MEK2 and p-ERK1/2 (Fig. 5e-f, Supplemental Fig 19), but we indeed have not studied off-target effects or any putative regulation of other members of the MAPK pathway. Therefore, we carried out a new set of phosphoproteomics experiments. In order to be able to compare with our initial results, instead of using primary neurons, we conducted the experiments with prefrontal cortex tissue from SOD1 mice treated with trametinib (female: n=4; male: n=5) or vehicle (female: n=3; male: n=4). In total, 9 phosphoproteins were differentially regulated in female animals (6 up and 3 down) and 15 in males (10 up and 5 down) ($p_{adj} = 0.1$) following trametinib treatment:

Among the differentially regulated phospho-proteins, we could not identify any proteins related to the MAPK pathway (keeping in mind the limitations in coverage of the phosphoproteomic analysis). Therefore, based on this data, we find no evidence for off target effects on the MAPK pathway.

To explore further pathways that may converge on the MAPK pathway - as also suggested by the reviewer - we conducted enrichment analyses with the candidates identified. These analyses were done using the ShinyGO 0.80 curated database for Reactome (Ge SX, Jung D & Yao R, *Bioinformatics* 36:2628–2629, 2020), and were done separately for males and females (enrichment FDR cutoff = 0.1). The top hits enriched for the DE results are summarized below and included in the new Supplementary Fig. 23:

Interestingly, for females, almost all of the top enrichment results are driven by CAMK2A (mouse accession: KCC2A), which is downregulated in its phosphorylation. CAMK2A is known to be upstream of MAPK, targeting MEK/Erk (PMID: 18817731; PMID: 8855261). Therefore, downregulation in CAMK2A phosphorylation, which is upstream of MEK/Erk goes in line with the effects that were observed in female animals treated with trametinib (Fig. 6), substantiating our initial findings.

Our initial experiments did, however, not support a disease-modifying effect in male animals. Our phospho-proteomic analysis also could not identify any proteins, which interact with the MAPK pathway. We now include this information in the text and added the volcano plot and the enrichment analyses to Supplementary Figure 23.

- Minor – AGC targets spelling mistake should be changed to 3×10^6 ions, currently reading as 3106 for MS1 and 1105 for MS2 in the paper

A: We apologize for this mistake. In our submitted text, we mentioned the adequate numbers with powers superscript: "...switching between MS... automatic gain control [AGC] fixed at $3 \cdot 10^6$ ions...and MS/MS... AGC fixed at $1 \cdot 10^5$ ". We believe that this formatting was lost after the submission of the original text. We have now fixed the spelling as suggested (using ^ to denote the powers) to avoid confusion.

- Minor – Have the authors considered using iPathway or IPA (realizing that these are commercial products) to re-analyse their omics data to make more accurate predictions as an alternative to GSEA and WGCNA which are mostly ways to summarize data.

A: We have indeed used IPA in the past and were quite impressed with the careful curation of the integrated pathways and the nice visualization options. Unfortunately, both iPathway and IPA are quite costly, which is why we have decided to rely by and large on open source algorithms and databases in this work. We appreciate the very constructive nature of the review, however.

I thought this was a great study incorporating different multi-omic technologies and the authors acknowledge the limitations of the current study and other avenues that are out of scope for this paper. They provide a characterisation from the RNA to the proteomic level of 100 tissue samples and 4 mouse models and I feel that the results will contribute to both the ALS and proteomics fields.

A: We truly appreciate the reviewer's considerations and all the constructive criticism of our work and hope to have addressed all the raised points sufficiently.

Reviewer #3 (Remarks to the author):

Amyotrophic Lateral Sclerosis (ALS) is a progressive fatal adult-onset neurodegenerative disorder characterized by the selective loss of lower and upper motor neurons as well as muscle degeneration. Most forms of ALS are sporadic with only 10% of the cases being inherited in a dominant manner (familial ALS). The four most common causative genes are C9orf72, SOD1, FUS and TDP-43. ALS is a devastating and highly heterogenous disease, for which there is currently no cure. It is thus critical to elucidate whether and if so which pathways are commonly dysregulated during pathogenesis to advance progress toward effective therapeutic development.

To discover potential novel, critical targets that are affected during the disease, Gomes, Hanzelmann and colleagues performed a multiomic analysis of a large cohort of postmortem human brain samples (in particular the prefrontal cortex, a region proposed to be less affected at end-stage, of n=51 ALS patients and 50 control non-neurological controls), as well as of four transgenic mouse models for C9orf72 (polypeptide expression)-, SOD1 (G93A)-, TDP-43 (M337V)- and FUS (WT)-ALS. In particular they analyzed the human and mouse samples using bulk transcriptomics, proteomics and miRNAomics and ultimately integrated all the datasets. The authors describe multiple dysregulated pathways, that are proposed to be gender specific. They further focus on the mitogen-activated protein kinase (MAPK) pathway, and more specifically one target, mitogen-activated protein kinase kinase 2 (MAP2K2 or MEK2) that they propose as an early disease-relevant mechanism. Treatment of ALS mice (or cultured neurons) expressing mutant SOD1 with trametinib, a drug inhibitor targeting MEK2 is reported to attenuate mutant-SOD1 associated pathological hallmarks including SOD1 insolubility, in females further leading the authors to propose MEK2 as a potential therapeutic target to treat ALS.

PreQ1 - This study provides an extensive analysis of the molecular changes occurring in ALS in a large cohort of human patients and mouse models. The use and integration of multiple omics approaches of such a large number of samples is certainly impressive and has the potential of being of interest for the scientific community when freely available. Perplexingly, after such a large effort, the authors focus on the MAPK pathway, for which there are already several studies proposing its potential relevance in ALS (Sahana and Zhang 2021, Biomedicines; Kim and Choi 2010, Biochimica et Biophysica Acta; Gibbs et al. 2018, Nature Cell Death and Disease; Pérez-Cabello et al. 2023, PNAS), further limiting the novelty of the findings. Hence, trametinib, a MEK2 inhibitor, is currently being tested in a clinical trial conducted in South Korea.

A: We appreciate the reviewers assessment of the overall potential of our analysis and we are very much aware of the clinical trial going on in South Korea. As we pointed out in our manuscript before, our data

identifies several other molecules that could represent promising therapeutic targets, but we started to explore the MAPK for several reasons: We observed consistent alterations of this pathway in human ALS patients and murine models across a spectrum of data types, integration methodologies, and distinct subclusters. Furthermore, we recognized that the activity of MEK1/MEK2 kinases could be modulated using the highly selective allosteric inhibitor trametinib, an FDA-approved drug. Given our previous experience with translatability of molecular targets in investigator-initiated clinical trials (PMID: 30972018, NCT03792490), the possibility to explore a pathway that is targeted by an approved drug had a strong impact on our choice. It is important to emphasize that, at the time of our project's initiation (2019) the trametinib trial has not started yet and no information about it was available to us.

Indeed, when we learned about the Korean trial, our experiments to validate the MEK2 inhibitor trametinib have already started. At that time, we intensively discussed whether we should pursue our analyses or validate other targets, instead. In conclusion, there was consensus in our consortium that our multiomic data strongly favors MAPK as a target and that there is merit in generating novel and more detailed evidence that reinforces the significance of this pathway in the context of ALS. In fact, we believe that the trametinib trial even further supports the choice of our target and will yield important clinical information on the usefulness of trametinib for ALS therapy.

In line with this reviewer, our team is driven by the motivation to generate novel data. We believe that the role of the MAPK pathway in ALS (substantiated by multiomic data analysis) and the effects of trametinib in models *in vivo* and *in vivo* have not yet been described in this detail before and therefore are truly novel, even if the involvement of the MAPK pathway has been suggested in ALS before. It is important to highlight that we observe sex-specific differences in MAPK regulation, in the multiomic data, in the cell culture validations, and even in the *in vivo* experiments. These findings are absolutely novel and highlight the importance of considering the effect of sex in human trials, especially for diseases like ALS that exhibit a sex-bias in disease frequency and severity. This data can contribute to more personalized treatment approaches and we are not aware of any previously published data in this regard. Lastly, the datasets, which are deposited in repositories, will permit researchers to re-analyze this data and to validate further potential targets, including targets that have been less explored before.

PreQ2 - While the study has the potential to provide valuable insight for disease mechanisms, in the present format there are several critical shortcomings including the execution of the validation of the targets which is weak, the analysis/interpretation of the datasets (the strong bias in numbers of male samples (n=35) in the ALS cohort that were analyzed compared to females (n=16) which should be taken into consideration when interpreting the data and concluding that there are more pronounced changes in males) and the highly confusing assembling/presentation of the data/figures. Overall, the sum of the current effort is confusion as to which specific molecular events are critical for disease onset and progression. Without substantial modifications, the manuscript is not appropriate for publication.

A: The two raised points are valid, the manuscript was not presented in a coherent and optimal way and the imbalance of sex in ALS patients could impact the observed differences in DEGs, while to a lesser degree in threshold-independent analyses like gene set enrichment. We would like to emphasize that sex-differences do not only refer to differences in the number of DEGs, for example, but also to differences in the observed pathways and terms that are enriched. Regarding the structure and content of the manuscript, we now completely remodeled the text, figures, and the order of the manuscript, as suggested by the reviewer and we hope that these amendments resulted in a scientifically and structurally improved manuscript. In regard to the effect of the sex imbalance, we conducted a sub-sampling analysis to provide evidence that the observed differences between male and female DEGs is not due to a mere difference in the size of the cohorts. In general, we are aware that the distribution of the samples is not balanced in terms of sex. Although we attempted to match our samples in the best possible way, we were limited by the availability of postmortem tissues in the brain banks. Although we leveraged the potential of four brain banks, not all parameters could be perfectly matched. Nevertheless, we did not want to reduce our cohort in order to have more balanced numbers. We also considered differences in the incidence of the disease in male vs. female subjects - this likely influenced the numbers of postmortem samples available from

brain banks; a higher incidence of the disease in males would also advocate the use of a male-predominant cohort in sporadic ALS.

To assess a potential bias induced by the uneven sex distribution, we re-ran the DEG analysis for our transcriptomic and miRNA data using 16 random samples (from each male, female, als, ctrl) 20 times. We represented the summary of the 20 bootstraps in the table below and confirmed that significantly more DEGs and DE miRNAs were still observed in males (ALS vs. Ctrl.) compared to females. This proves that the stronger signal observed in males is not due to a low number of female ALS patients in our cohort but to real sex-specific differences. Please be reminded, again, that when we refer to sex-differences in the manuscript this does not only relate to the quantity of DEGs, for instance, but also to differences in the results of GSEA enriched ontologies and pathways. We included these important results in the revised manuscript and would like to thank the reviewer for bringing our attention to this crucial point.

Supplementary Table 4

Sex	Regulation	Number of DEGs (mean of 20 bootstraps ± one standard deviation)	Number of DEGs Full dataset
female	downregulated	3.45 ± 7.21	1
	upregulated	0.30 ± 0.66	1
male	downregulated	37.50 ± 103.06	70
	upregulated	3.65 ± 14.70	3

Sex	Regulation	Number of DE miRNAs (mature) (mean of 20 bootstraps ± one standard deviation)	Number of DE miRNAs (mature) Full dataset
female	downregulated	4.55 ± 1.70	4
	upregulated	4.65 ± 2.23	5
male	downregulated	13.70 ± 4.54	15
	upregulated	2.30 ± 0.86	2

Main Concerns:

Q1 • The flow is very difficult to follow as the figures are currently organized by techniques and not the overall scientific outcomes (findings). This makes the understanding/reading very confusing as the description of the data is not done in the order in which the figures are currently assembled. Effort should be made to reorganize the content of the main figures so that it is easier to follow the experimental flow and findings. One single paragraph should have the corresponding main figure described based on the scientific outcome (and not the technique- for example currently Figure 1 is composed of transcriptomics

data, Figure 2, proteomics..etc..). On this note, the main important missing figure of the paper is the one recapitulating all the findings pointing to MEK2 as the main target to which the authors directed their attention on (corresponding to the paragraph at line 294). This important point is missed in the text, since data is embedded within different main figures with fragmented and weak information throughout the paper.

A: Thank you for this constructive critique. As suggested by the reviewer, we have now re-organized the main text and all figures to increase the legibility and logical flow of the main findings. We attempted to follow the reviewer's suggestion, with individual paragraphs describing the scientific outcomes displayed in main figures, and we have now one dedicated figure (main Figure 4) that highlights all the evidence we gathered for the MAPK pathway within our multiomic study. We have also added a results section to recapitulate the findings that point to this pathway and substantiates MEK2 as an interesting therapeutic target, as suggested by the reviewer.

Furthermore, following the suggestion of the reviewers, we have now created a separate file for the supplementary figures. All the legends also appear directly after each supplementary figure, facilitating the navigation through the content. The new version of the supplementary figures can be found in the supplemental file entitled "SI figures and captions".

We believe that these changes have significantly increased the readability of our manuscript and want to thank the reviewer for raising these topics.

Q2 - a The strong justification mentioned in the main text for the initial separation of the available transcriptomic datasets based on sex is weak. The difference which leads to the transcriptomic separation of samples in males and females is only 15%. Indeed, in the PCA plot in Suppl. Fig. 5 (which was actually saved under Sup. Fig 4), some samples derived from males and females are juxtaposed. It would be valuable to perform unsupervised clustering of transcriptomes (instead of enriched pathways) of all the samples to justify the separation between males and females. This would be more convincing. Moreover, the WGCNA analysis performed in Figure 1e shows consistency between male and female samples when divided by clusters, and not differences, which does not support the conclusion the authors put forward about the sex-difference. Similarly, in Figure 4g, there is no clear clustering of male samples separated from female samples on the UMAP analysis, which again does not support the claim made. On the contrary, male samples seem to be clustered in two sub-clusters. Moreover, in Figure 4h, factors cannot be compared between male and female samples because the analysis was done separately resulting in different genes representing the specific factors and different correlation scales. The data representation in the current format is misleading. Altogether, the evidence presented does not demonstrate that there is a sex-difference as the authors conclude.

A: While we acknowledge and appreciate the reviewer's feedback, we respectfully disagree with the assertion that our data fails to support notable sex differences in ALS. In fact, our findings consistently highlight pronounced sex disparities, both in terms of magnitude and quality and across various omics levels.

The principal component analysis (PCA) for transcriptomics demonstrates a distinct separation by sex, with silhouette scores of 0.29 for males and 0.15 for females, surpassing the modest separation by condition (silhouette scores: 0.11 [ALS], -0.03 [CTR]) (Fig. 1b). The transcriptomic analysis reveals substantial differences, including 70 down-regulated and 3 up-regulated genes in males, compared to 1 up-regulated and 1 down-regulated gene in females (Fig. 1c). Furthermore, bootstrapping experiments confirm the robustness of these sex-specific differences. In the realm of alternative splicing, males exhibit significantly higher numbers of differentially alternative spliced events compared to females across various categories (Fig. 1d). This trend extends to the miRNA analysis, where 93 differentially expressed molecules are identified in males compared to 24 in females.

In addition, the proteomics analysis solidifies our argument, showcasing 379 differentially expressed proteins in males as opposed to 151 in females. Even in the mouse study, with equal numbers of male and female subjects in each group, the PCA across all models distinctly separates males and females,

with sex accounting for significant percentages of sample variance (44.4% in C9ORF72, 27.7.8% in FUS, 29.5% in SOD1, and 28.6% in TDP43).

Strikingly, in the MAPK pathway, we observed sex-specific activity prompting comprehensive investigations into physiological and behavioral assays in mice. Subsequent experiments in cell culture and animals consistently support and strengthen the evidence for the robust sex differences observed in our study. In essence, our results underscore the significance of sex as a factor in the understanding of ALS, with a notable 15% explained variance attributable to sex alone in human data.

These results don't necessarily come as a surprise, given the strong sex-bias in human disease occurrence and severity and some of the recently published studies on sex-specific differences in ALS:

- 1) Murdock et al, 2021, PMID: **33531377**: Higher baseline neutrophil counts are associated with shorter survival in female participants with ALS.
- 2) Santiago et al, 2021, PMID: **34281203**: sex differences identification of genes and pathways in ALS.
- 3) Goutman et al, 2022, PMID: **35088843**: Metabolic differences among ALS males and females
- 4) Murdock et al., 2021, PMID: **33974561**: Correlation between changes in NK cell metrics and changes in ALSFRS-R stratified by sex.
- 5) Pape et al., 2020, PMID: **32147204**: Review - The Effects of Diet and Sex in Amyotrophic Lateral Sclerosis

Our analysis demonstrates that sex-specific differences are present throughout many molecular layers and therefore, we believe it merits special mentioning.

To add a few examples, we include PCAs below showing the 500 most variable genes for the different mouse models. As seen also in the PCA for human samples (Fig. 1b), clear sex differences are observed across the analyzed cohorts:

A) C9ORF72

C) SOD1

B) FUS

D) TDP43

Q2 - b It would be valuable to perform unsupervised clustering of transcriptomes (instead of enriched pathways) of all the samples to justify the separation between males and females. This would be more convincing:

A: We agree that performing unsupervised clustering would be an addition to this work in regard to the differences between males and females. We have now performed the clustering analyses using the top 500 highly variable genes for ALS subjects. From this figure it is clear that our transcriptomic result is mainly driven by sex. The marked sex difference is represented in the PCA (Figure 1b) and, in addition, we provide a clustering scheme that substantiate these findings:

Q3 • More concern is directed to the data in Suppl. Fig. 5 where the clear separation between samples derived from ALS patients and control subjects is lacking. For this reason, in Figure 1b, the differences between ALS and control samples, especially in female underrepresented samples, are very subtle. Interestingly, in one of the studies mentioned by the authors (Aronica et al. 2015, Neurobiology of Disease) where less samples are included, the number of differentially expressed genes was very much higher compared to the one in the present transcriptomic dataset. The authors should discuss this observation.

For the separate analysis of males and females, the authors should generate two separate PCA plots with ALS/control female samples and ALS/control male samples.

A: While we acknowledge the findings of Aronica et al., there are notable differences in our approach. Aronica et al. focused on the motor cortex, a region severely affected in ALS patients at the time of death. In contrast, we targeted the prefrontal cortex, a region affected later in the course of the disease and only presenting intermediate TDP-43 pathology at death (Verde et al., 2017, PMID: 29405032). We here aim to capture early disease alterations, which should be more subtle and variable than the severe changes of cell populations in late ALS motor cortex.

It is important to highlight that Aronica et al. did not refer to any sex adjustment or stratification in their analyses - which may lead to the detection of more differentially expressed genes, particularly given the imbalance in sex distribution between control and ALS groups in their study. In our study, we considered sex differences for all analyses and stratified datasets by sex, which allowed us to capture important variations in ALS-related mechanisms separately for males and females.

Furthermore, our study employed RNA sequencing, while Aronica et al. used microarrays. This distinction in the methodologies and their coverage further complicates direct dataset comparisons. As requested by the reviewer, we now also provide below two separate PCA plots of the 500 most variable genes for each sex. One with ALS/control female samples (left side) and the other with ALS/control male samples (right side).

Q4 • The fact that human sample clusters correlate partially with mouse model ‘omics datasets do not imply that one mouse strain is modeling one specific human cluster in this study. The correlation is very minor and should be discussed as such. Additionally, in Figure 2d, C3 samples resemble more the mouse models respect to the other human clusters, making this analysis rather weak.

A: We completely agree with the reviewer that human sample clusters correlate partially with model omics datasets. Accordingly, we made sure to avoid any overstatement in the revised manuscript, highlighting the observed correlations while avoiding any stronger statements of ‘modeling’ specific human clusters. However, since the mouse models are based on different disease-causing genes, which are involved in different disease-related pathways, we found it interesting and legitimate to assess whether mouse models could partially represent human clusters. To underline our point, we have now added correlation values in the correlation heatmap (Figure 3d) based on changes in cell type fractions (deconvolution) of all transcriptomics data. We also added the information in the text (**lines 274-279**): “Overall, our transcriptome analyses revealed correlations between human clusters and mouse models: C1 and C2 showed the best correlation with the SOD1 model (C1: 0.11 and C2: 0.42), whereas C3 correlated best

with the C9orf72 model (0.31) and, to a lesser extent, TDP-43 (0.23) and FUS (0.14) models. Finally, C4 showed the best correlation with the FUS model (0.14) (Fig. 3d).”

Q5 • Supplementary Figure 12 reports transcriptomics changes of genes related to MAPK pathway which show great variability and overall stable expression across groups (white color). This represents the first evidence in support (according to the authors) to the involvement of MAPK signaling in the disease pathogenesis of mouse models and human patients. This conclusion is weak and appears in the text without a clear rationale (line xxx). Moreover, in main Figure 2e, it is suddenly reported the log2 FC of MAPK pathway genes in ALS cluster 1 and 3. The authors should report also the log2 FC of these genes for cluster 2 and 4. Moreover, the size effect is subtle, the log2 FC scale goes from 1 to -1, which further makes this referee wonder why the authors focused on this target (given that it was not novel).

A: We appreciate the comments of the reviewer on the selection of the MAPK pathway. As already mentioned before, we apologize that the presentation of our data was not clear enough and we therefore re-structured our figures, including a new Figure 4, which summarizes the evidence on the involvement of the MAPK pathway. The text was also re-structured to better describe these experimental findings. We now also show the log2(fold change) of the MAPK-associated genes for cluster 2 and 4 in the Supplementary Figure 20.

Also, we would like to reiterate our answer to a similar question of reviewer 2. Although, we agree (and stated this in our manuscript before) that our data identifies several other molecules that could represent promising therapeutic targets, we started to explore the MAPK for several reasons: We observed consistent alterations in human ALS patients and murine models across a spectrum of data types, integration methodologies, and distinct subclusters. Furthermore, we recognized that the activity of MEK1/MEK2 kinases could be modulated using the highly selective allosteric inhibitor trametinib, an FDA-approved drug. Given our previous experience with translatability of molecular targets to investigator initiated clinical trials (PMID: 30972018, NCT03792490), the possibility to explore a pathway that is targeted by an approved drug had a strong impact on our choice.

Q6 • The rationale for the prioritization of miR-451a by miRNA expression analysis to support MAPK pathway involvement in ALS is very weak. The authors propose in the main text that all the mouse models and human patients showed deregulation of miR-451a, but from the data presented, only human males, SOD1 male/female mice and C9orf72 female mice suggest such an effect. The effect is reported to be significant only in human males and C9orf72 mouse females.

A: We agree with the reviewer that our findings concerning the miR-451a must be better contextualized to include the levels of significance across the analyzed cohorts. The precise numbers are now reported in the text. We also agree that not all of them are significant, but we think it is important to highlight that

this miRNA presents similar expression changes (downregulation) within all of the analyzed cohorts. Following this request, the **lines 338-345** of the manuscript have been rewritten and the legend for the main Figure 4 has been also updated:

“We also carried out a miRNA expression analysis, which identified miRNA-451a consistently deregulated across mouse models and human samples with ALS (Fig. 4c, Supplementary Fig. 19). Target prediction analyses revealed that miRNA-451a interact with multiple members of the MAPK pathway (Fig. 4d), including MAPK1, AKT1, and BCL2, which are known for their roles in the regulation of cell growth, survival, and apoptosis³⁷. In addition, centrality measurements for targets of this miRNA showed MAPK1 as the top hit, followed by AKT1, IKBKB, BCL2 and MYC. This indicates a pivotal role of mir-451a in the context of MAPK signaling (Fig. 4e).”

New legend for Figure 4c:

“Differential expression analyses for mature miRNAs for humans and ALS mouse models reveals consistent deregulation of miR-451a

Mouse models exhibit pronounced differential expression (DE) of miRNAs, with the C9orf72 model showing the most significant changes. Among the top regulated miRNAs, miR-451a is consistently deregulated across all mouse models and human patients with ALS. While C9ORF72, FUS and TDP43 males mouse model demonstrate a trend for increase of miRNA-451a levels, the opposite trend is observed in SOD1 male mouse model ($\log_2FC=-2.34$, $padj=0,1685$) and a significant decrease is visible in human males with ALS ($\log_2FC=-0.91$, $padj=0,0002$). miR-451a also shows significant decreases in the female C9ORF72 mouse model ($\log_2FC=-1.82$, $padj=0,0363$) and trends towards downregulation are observed across all other female mouse models (SOD1: $\log_2FC=-2.34$; TDP43: $\log_2FC=-0.55$) and female humans PFC ($\log_2FC=-0.08$).

Q7 • Finally, to assess the preclinical relevance of the chosen treatment trametinib, behavioral tests and survival curves to assess disease phenotype/course should be performed.

A: We thank the reviewer for asking this pertinent question, which was also asked by Reviewer 2. We performed an additional experiment with new cohorts of vehicle (n=8) and trametinib-treated mice (n=10) treated from 9 weeks up to survival, 2 times a week, with 3 mg/kg trametinib, as done in our initial study. Based on the 3R principles, it was not ethical nor could we get the approval to proceed with more in vivo experiment on SOD1 male, given the negative evidences of trametinib treatment effects in male animals (i.e. NfL, p62, insoluble SOD1 and ubiquitinated proteins). We thus focused only on SOD1 female mice: starting from 9 weeks (start of treatment), we monitored body weight, grip strength and extension reflex twice a week. No effect was observed in body weight loss and behavioral analysis, but there was a significant delay in the age of paralysis ($p=0.0376$ by Student's t test) and onset of the disease ($p=0.0177$ by Log-rank Mantel-Cox test) and a significantly improved survival ($p=0.0405$ by Log-rank Mantel-Cox test), indicating that trametinib has a beneficial effect in SOD1 female mice. All results have been now incorporated into the new Figure 6 (panels i-k).

Additional points:

- C3 and C4 clusters in Figure 1c show their heterogeneity, and RNA splicing seems not to be a strong factor of their segregation.

A: It is important to point out that for former Figure 1c, we actually refer to the WGCNA results (current Fig. 1g-h) when we describe an enrichment for RNA splicing in the human clusters. It is shown for WGCNA modules *tan* and *lightcyan* (Fig. 1h) and shows upregulation for clusters C3 and C4 as seen in the heatmap in Fig. 1g. Even though “RNA splicing” is not a strong factor of separation when considering the clustering by activity scores (current Fig. 1f), it appears in our WGCNA enrichment analysis and it is known to be relevant for ALS pathology. Therefore, we decided to include this pathway in the analysis. It seems that it is equally important/affected in different ALS patient clusters since there is no obvious clustering, as pointed out by the reviewer.

- Supplementary Figure 4 f-g-h missing. RIN analysis should be reported also for human samples in order to exclude defective samples from the analysis.

A: We apologize for not having included human RIN values upon the initial submission and now included them in Supplementary Fig. 4, as requested by the reviewer. We were fully aware that some of the human postmortem samples present low RIN values. This is a limitation intrinsic to the kind of material we are studying, given the long time of storage of some of these samples, as well as the time intervals between the autopsy and the sampling procedures (postmortem interval: PMI). Unfortunately, we could not influence this while applying for these samples. Nevertheless, previous studies showed that low RIN values did not necessarily impact the quality of library preparations for NGS applications. While RIN values are regarded as an accurate calculation for the integrity of ribosomal RNAs, RINs are not ideal to infer about mRNA integrity directly, which is the main input for RNA sequencing (PMID: 26842848; <https://emea.illumina.com/content/dam/illumina-marketing/documents/products/technotes/evaluating-rna-quality-from-ffpe-samples-technical-note-470-2014-001.pdf> [technical note from Illumina, Inc]). Furthermore, we also verified whether the PMIs influenced the expression level of selected neuronal markers for human brain samples (Supplementary Fig. 4a). No significant effects were observed on the abundance of these neuronal markers in our cohort in relation to the PMI values. Therefore, since the availability of such precious postmortem brain samples is very low - making it a challenge to establish such a large cohort of samples - we decided not to exclude any samples based on the RIN values.

- In Supplementary Figure 7, it is not clear which genes are shown with the corresponding color. Colors seem to refer to more than the 50 selected ones. Moreover, from this visualization it is clear that some samples do not belong to the assigned clusters defined by Tam et al. (2019).

A: Thank you for pointing this out. All displayed genes and their groups are found in Supplemental file 4 (table S2A) of Tam et al. (2019). We updated the figure caption to be more precise:

Genes were selected based on Supplemental file 4 (table S2A) from Tam et al. (2019) grouped into ALS-TE (orange), ALS-Ox (light-blue) and ALS-Glia (dark-blue) according to Tam et al. (2019). Gene expression is vst-transformed and mean-variance scaled for display. The sample- subclusters are identified using pathway information as described in Fig 1. Only a representative number of gene names is shown.

Indeed, several samples cannot be directly categorized in the clusters defined by Tam et al. (2019). Therefore, we write in the main manuscript, that we only observe some similarities between the clusterings, but no direct correspondence and thus we do not investigate the Tam et al. clusters in more detail (**lines 147-151**): *“These clusters are reminiscent, but not identical to previously proposed subtypes9, where C1 and C2 align with ALS-Ox (oxidative stress) and showing less resemblance to ALS-TE (elevated transposable element expression), while C3 and C4 correspond to the ALS-Glia (glial dysfunction) subtype (Supplementary Fig. 6).”*

- Xpo1 shows upregulation also in C9orf72 and SOD1 females. It is not reported if the difference in fold change between males and females is significant/relevant. Importantly, the authors highlight the fact that Xpo1 is a target in an ALS clinical trial (line xxx), but in human patients the effect is not present.

A: This is correct: in this paragraph, we highlighted the top regulated proteins in this paragraph and XPO1 remains the only significantly regulated protein in the SOD1 model. In female SOD1 mice, the padj is 0.052 for XPO1, becoming nearly significant (male: FC=0.42, padj 0.028, female: FC=0.34, padj 0.052). There is no significant difference in the fold-change. Also in the C9orf72 model, XPO1 is significantly regulated, but not one of the top regulated proteins (male: FC=2.33, padj 1.436E-4, female: FC=1.74, padj 0.029) (Supplementary Table 11). Because the trial with the XPO1-inhibitor BIIB100 has

ended and the company decided not to go forward with this drug target, we now refrain from including this information in the text (**lines 286-288**): “SOD1 mice showed one upregulated DEP, exportin-1 (XPO1), a major regulator of nuclear RNA export. XPO1 was also among the significantly regulated proteins in the C9orf72 model (Supplementary Table 11).”

- In figure 5, the authors should use the antibody for phospho-MEK2 in primary neurons to assess activation of MEK2 in addition to phospho-Erk1/2 readout. Total MEK2 protein levels should be used as normalization factor.

In supplemental Fig. 21 we show MEK2 expression but we now also used the antibody for p-MEK2 to assess the activation of the MEK2 protein. MEK2 was normalized with the housekeeping protein lamin B and as requested, MEK2 was used as a normalization factor for phospho-MEK2 and the main figure was updated. MEK2 protein level wasn't significantly affected by different concentrations of trametinib in absence or presence of the stressor. We observed a significant increase of pMEK2 activation in response to trametinib treatment. This can be explained by inhibition of the feedback circuit: MEK2 inhibition induces a decrease of pERK1/2 (Fig.5), which also weakens the negative feedback loop, resulting secondly in the activation of the pathway and the accumulation of activated pMEK2 (PMID: 19251651, PMID: 27342992). The new information was inserted **lines 367-371**.

- In phosphorylation analysis through western blot, the normalization signal is always the total protein level of the corresponding protein, not the loading control (in this case lamin b should be reported but not used as normalization factor, instead the authors should use a total Erk1/2 antibody).

A: We updated the main Fig. 5f and now normalized p-ERK1/2 activation on the total amount of ERK1/2 protein. Here total ERK remains stable across all conditions (same as the housekeeping lamin b protein). Thus, as demonstrated in the first version of the manuscript, we observe that trametinib treatment significantly decreases pErk1/2 level.

- The performed western blots should be reported in the main figure with the relative quantifications in Figure 5. The uncropped versions of the images should be reported in the supplementary material.

A: According to the reviewer comment, we now included the new Supplementary Figure 22: Western blot full membranes for 3 different experiments showing the Erk1/2 and MEK2 protein expression and their phosphorylation after treatment with different concentration of trametinib both with and without the presence of glutamate. To optimize resource and time utilization, membranes were precisely cut around the protein size and subsequently exposed to their respective antibodies. Membrane 1 and 2 were systematically run in parallel on the same day.

Style/formatting concerns:

- Table titles/legends are missing. It was extremely difficult to understand which supplementary figure was what given that none have a title, and on top of that some were incorrectly saved (only the pdf files were labeled but there were inversions). For the eps files there were no labels so this referee had to guess based on the summary figure legends. This referee strongly recommends writing on each Supplementary figure, their number.

A: We apologize for the confusion in the supplementary figure files. As we pointed out above, a new document with the SI content was prepared to address all these points.

- The authors should include titles and legends in all the heatmaps and graphs to help the reader understand the architecture of the data. Where possible, resize them to make them more readable (often it is too small to read all the text).

A: We thank the reviewer for the critical comments and have now reworked all artwork to improve readability.

Reviewer #4 (Remarks to the Author):

In this study, Gomes and colleagues generated a variety of omics data on postmortem prefrontal cortex tissue from 51 ALS patients and 50 control donors, as well as on 4 well-known mouse models of ALS genetic mutations. They observed clear sex differences in both mice and humans, and were able to cluster the human patients into 4 subtypes, reminiscent of previous studies on an independent post-mortem cohort. By integrating their data together, they prioritized the MAPK signaling pathway and demonstrated that an inhibitor of MAP2K2 rescued the effects of excitotoxicity on primary cortical neurons, again with a sex-specific effect. Overall, they did a thorough and careful analysis and their findings will be useful for the field. However, I found the paper and methods hard to follow and missing some important technical details.

Q1 - Given that ALS is associated both genetically and clinically with frontotemporal dementia, is it possible that the transcriptomic differences observed in the frontal cortex that separate ALS donors are caused by concomitant FTD in a subset of those donors? Did the authors check whether their donor patients had cognitive decline or FTD symptoms before death?

A: We thank the reviewer for raising this important point, which has been also mentioned by other reviewers. Here, we have taken into account all clinical information from the brain banks that provided the samples for this study. Following the reviewer's suggestions, we have extended the table with the clinical information to report any co-occurrence of FTD among the patients of our cohort. Among the 51 ALS patients, 40 demonstrated no FTD-ALS pathology, 6 demonstrated mild FTD-ALS pathology and only one presented clear FTD-ALS pathology. FTD-ALS pathology was defined as ubiquitinated TDP-43 inclusions in the cerebral cortex or limbic system as stated in the available pathology reports (PMID: 16222525). For 4 patients, this information unfortunately was not provided. Limited clinical data is available for these brain bank samples, but none of the patients was noted to have been diagnosed with FTD. These patients did not seem to cluster within the ALS cohort. To better visualize FTD pathology, we have now included this annotation in the heatmap depicted in Supplementary Figure 2b, in addition to the updated clinical information table that is provided as a supplement.

Q2 - The sequencing methods list two different library preparation kits used for mRNA-seq - one polyA enriched and one for total RNA with ribodepletion. Library preparation usually has a large effect on gene quantification. Was this library batching balanced across disease groups? Was this adjusted for in differential expression analyses?

A: We have now made sure that all information regarding the library preparations are included. The mouse samples were processed using the polyA kit. The human samples were processed using a ribo depletion kit because they were samples from a biobank. Due to differences in the library preparation, we avoided the computation of differentially gene expression between murine and human samples. We updated the methods section of the revised document accordingly:

“mRNA and small RNA sequencing experiments were performed in the Functional Genomics Center in Zurich. For the mRNA sequencing, total RNA libraries were prepared using either the TruSeq Stranded mRNA (Illumina, Inc, California, USA)(Short Read Sequencing), or the SMARTer® Stranded Total RNA-Seq Kit v2 -Pico Input Mammalian (A Takara Bio Company, California, USA)(Short Read Sequencing). Briefly, for the TruSeq protocol, total RNA samples (100-1000 ng) were poly-A enriched and then reverse-transcribed into double-stranded cDNA. The cDNA samples were fragmented, end-repaired and adenylated before ligation of TruSeq adapters containing unique dual indices (UDI) for multiplexing. Fragments containing TruSeq adapters on both ends were selectively enriched with PCR. This produces a smear with an average fragment size of approximately 260 bp. The libraries were normalized to 10nM in Tris-Cl 10 mM, pH8.5 with 0.1% Tween 20. For the SMARTer® Stranded Total RNA-Seq Kit v2 -Pico Input Mammalian protocol, total RNA samples (0.25–10 ng) were reverse-transcribed using random priming into double-stranded cDNA in the presence of a template switch oligo (TSO). This results in a cDNA fragment that contains sequences derived from the random priming oligo and TSO. PCR amplification using primers binding to these sequences adds full-length Illumina adapters, including the index for multiplexing. Ribosomal cDNA is cleaved by ZapR in the presence of the mammalian-specific R-Probes. Remaining fragments are enriched with a second round of PCR amplification using primers designed to match Illumina adapters. The product was a smear with an average fragment size of approximately 360 bp. These libraries were normalized to 5nM in Tris-Cl 10 mM, pH8.5 with 0.1% Tween 20. The quality and quantity of the isolated RNA and the enriched libraries were validated using a Fragment Analyzer (Agilent, Santa Clara, California, USA) (for the TruSeq kit) and the TapeStation (Agilent, Waldbronn, Germany) (for the SMARTer® Kit).”

Q3 - How exactly was differential expression performed? Which package was used? What covariates did they include? Please include this.

A: Following this request, we added a more detailed description to the methods section of the paper:

Differential expression analysis

To compare murine and human samples, transcript counts were normalized using DESeq2 size factor estimation. Subtype-specific differential expression of transcripts was determined using a 2-fold change cut-off and an adjusted p-value <0.05 (unless stated otherwise). Two different models were designed to analyze sex-specific differences: The cohort was divided into male and female cohorts. For each cohort we carried out differential expression between case and control samples using the RNA-seq batch as covariate for the human samples.

Transcriptomics data has been processed using the NextFlow Core RNASeq pipeline, version 3.0 described at (Ewels et al., 2020). The data has been demultiplexed with bcl2fastq, and the fastq files have undergone several quality checks including FastQC (Andrews, S. (2010). FastQC: A Quality Control Tool for High Throughput Sequence Data. Available online at: <http://www.bioinformatics.babraham.ac.uk/projects/fastqc/>) and fastq screen (Wingett & Andrews, 2018). Salmon was used for pseudo alignment and quantitation, with a salmon index built using GRCm39 with

annotations from GENCODE vM26 for the mice data and GRCh38 with annotations from GENCODE v37 for the human data. Count matrices from Salmon were used in downstream analyses.

The Principal Component Analysis and heatmaps used the count matrices from Salmon, after filtering and normalization using a variance stabilizing transformation blind to the experimental design. The count matrices were filtered, keeping genes with at least ten counts in 50% of the samples of any condition and sex.

Following differential expression analysis by DESeq2, we searched for relevant biological processes and molecular functions using gene set enrichment analysis on their Gene Ontology terms, using the clusterProfiler R package (Wu et al., 2021), filtering terms by size between 10 and 500 genes, and adjusting p-values for multiple testing with the Benjamini-Hochberg correction. To assess the heterogeneity of the samples silhouette scores were calculated on the first two principal components and averaged across conditions (Lovmar et al., 2005).

Q4 - Given that the clusters overlap significantly between males and females, did the authors not attempt a ALS vs Control differential expression using all the samples while adjusting for sex? This should boost their discovery power.

A: Thank you for this relevant comment. Our PCA and UMAP analyses (Fig. 1b; 2g) favored independent analyses for each sex, since we observed a better separation by sex rather than by disease condition. For this reason we decided to split the cohorts into males and females, focusing on sex-specific disease alterations. Nevertheless, we did attempt ALS vs Control differential expression using all samples while adjusting for sex. In line with what the reviewer suggests, using the full cohort adjusting for sex might indeed reveal additional targets in transcriptomics by increasing statistical power, as indicated below (A). We found 141 DEGs in all ALS vs all control cohorts (p-adj value = 0.05 and a cutoff of $\log_2(1.5)$), while separating by sex revealed 2 DEGs in females and 73 in males. These results suggest, however, that male signals could predominate the differential results, masking specific female information. Even more important is the fact that some of our enrichment analyses consider the full expression profiles rather than only significant results (for example, GSEA or WGCNA analyses). Having the cohort split by sex allowed us to have a more exploratory view into the expression data, regardless of the hits found by DE analyses. This was especially important for females, given the number of DE hits identified. Similar findings were also seen for miRNomic data (B and C) and proteomic data (D). Adjusting for sex unfortunately seems to cover up female-specific information, while does not always boost our discovery power (B-D).

For all reasons above, we decided to analyze the cohorts for each sex independently, aiming to have a more exploratory view of male- and female-specific disease-mediated pathways. Especially female patient cohorts are, in our opinion, not sufficiently studied in the field.

Venn diagrams of the DE results in transcriptomics (A) - padj value 0.05 and a cutoff of $\log_2(1.5)$, microRNAomics (mature miRNA, B; hairpin, C) - p-value 0.1 and a cutoff of $\log_2(1.5)$ - and proteomics

(D) - p value 0.1 no fold change cut off - comparing results for males only, females only and for all ALS patients adjusted for sex.

Q5 - The authors should clarify in the main text what thresholds were used for selecting genes for pathway enrichments.

A: Thank you for raising this important point. We added the cut offs in the respective results sections and added more details to the methods section **lines 863-894**:

“Enrichment analyses

Gene ontology, pathway enrichment analyses, and protein interaction networks

GSEA was performed using gseGO and gseKEGG from the clusterProfiler R package, with biological processes and molecular functions chosen as background databases for GO enrichment. The p-value cutoff was set at 0.05. Differential expression was presented using volcano plots generated with the Enhanced Volcano plot package in R, and over-representation analysis was performed on genes that showed at least one significant DAS event using the clusterProfiler function enrichGO, with a p-value cutoff of 0.1 and Benjamini–Hochberg correction for multiple hypothesis testing. Protein–protein interaction networks were created using the STRING protein interaction network database version 11.067,68 using standard settings.

Weighted Correlation Network Analysis

To conduct a weighted gene co-expression analysis, we employed the WGCNA package¹⁵. Pairwise Pearson correlations were calculated to establish signed regulatory networks within the WGCNA. By constructing an adjacency matrix, we applied a soft-thresholding technique to approximate a scale-free topological network. Eigengenes or eigenproteins were calculated as the first principal components of each module. This resulted in the development of several modules. We merged similar modules based on hierarchical clustering (SOD1: 0.4, C9orf72: 0.4, FUS: 0.5, TDP-43: 0.4 for RNA-Seq and 0.25 for proteomics). We calculated the relationships between WGCNA modules and traits. Sex was also included in the traits resulting in four traits: male ALS, female ALS, male CTR, and female CTR. First, we filtered modules based on the significance of the module–gene relationship ($p < 0.05$) and then selected modules that were highly correlated with either male or female ALS. The correlation cutoffs differed between the mouse models (SOD1: 0.5, C9orf72: 0.3, FUS: 0.3, and TDP-43: 0.5 for RNA-Seq and SOD1: 0, C9orf72: 0, FUS: 0, and TDP-43: 0 for proteomics). Using WGCNA, we also analyzed co-expression networks in mouse models. The minimal module size was set to 30 with a merge height of 0.4–0.5 and a correlation threshold of 0.3–0.5.”

Q6 - My biggest source of confusion was understanding how exactly clusters C1-C4 were defined. The text describes them as arising from hierarchical clustering of enriched GO/KEGG pathways, but how is that converted into an “activity score” for each ALS patient? Fig 1c mentions “Activity scores calculated by Decoupler”, but there is no explanation of this or citation in the methods.

A: We apologize for any inconsistencies that might have driven these points of confusion. A similar point was raised in the question #2 by the first reviewer, and we tried to address the pathway selection for clustering analyses as detailed as possible. For that, we added a detailed description on how we selected the pathways and calculated the activity scores (**lines 896-928**):

“Pathway selection for clustering analyses with transcriptomics datasets

The selection of the pathways used for clustering analyses of human samples was based on the enrichment analyses performed with the transcriptomics dataset. We selected the top hits from gseGO and gseKEGG analyses [from the clusterProfiler R package], as well as a co-expression network analysis (weighted gene co-expression analysis - (WGCNA) analysis, analyzing the top 30 most highly enriched terms for each dataset, and then grouped these terms by more general umbrella terms, to select ten pathways that summarized the enriched terms best: mitochondria/respiration; synapse; MAPK cascade; oxidative stress; nucleocytoplasmic transport; protein folding/metabolism; lipid metabolism; RNA splicing;

extracellular matrix and activation of immune response. In more detail, for overlapping KEGG pathway and GO results, a predominant enrichment for extracellular matrix (ECM) and immune response pathways and synapse-related terms was observed for both males and females. The convergence of these themes across sexes suggests common biological themes relevant for the disease and justified their selection for the clustering analyses. In addition to neurodegenerative disease pathways driven by DE genes involved in protein metabolism and oxidative stress mechanisms (Supplementary Fig. 5), female specific results showed enrichment for aerobic respiration (GO) and ribosome (KEGG), while protein metabolism was captured for male results (KEGG). Oxidative stress was also inferred from the enrichment of oxidative phosphorylation pathways in both males and females (KEGG). WGCNA results revealed key associations with all of the selected themes, especially for terms related to mitochondria/respiration and protein metabolism, synapse, nucleocytoplasmic transport, lipid metabolism; immune response and RNA splicing. WGCNA results were also very frequently enriched for MAPK-related terms (Supplementary Table 8), justifying its selection for the clustering analyses.

Identification of transcriptome-based sub clusters in human ALS patients

*We used decoupleR (v1.1.0) 76 to aggregate scores from gene sets derived from GO terms of interest. Gene lists with the components of GO terms of interest were extracted using the AmiGO2 database (<https://amigo.geneontology.org/>). For the unsupervised per sample enrichment analyses, we used decoupleR with the consensus from the *mlm*, *ulm*, *wsum* functions76 and calculated activity scores. These are numerical values associated with genes and represent gene expression levels.”*

Q7 - When presenting WGCNA modules, the convention is to just use the colour names - Mturquoise should be turquoise, etc.

A: Thank you for the suggestion, we changed the module names accordingly.

Q8 - Splicing results are out of order in the results. Should be discussed after Figure 2, could be joined with the miRNA work. Fig 3d in results is linked to TPRN, which is not actually present in that figure - instead a set of pathway enrichments which does not appear to be discussed in the results.

A: Thank you for these very valuable suggestions. Indeed, the presentation of the results was confusing and we thoroughly reworked it in the revised manuscript. We have adjusted figure and text order to have a more appropriate flow for the presentation of our results. We hope that the new structure attends the points of critique raised here.

About the second point: the results related to TPRN are related to DAS analyses for humans. Differential results for DAS and derived pathway analyses in humans are currently presented in Supplementary tables 5 and 6. A summary for enrichment results was indeed presented in former Fig. 3d (current Fig. 4b). TPRN cannot be linked to any of the summarizing pathways, since the majority of those come from mouse model results. Therefore, we present these findings directly in text (lines **121-125**) and also directly refer to the supplementary tables where they can be found accordingly.

In the splicing analysis they state that cryptic splicing events “are rarely detectable in bulk RNA-Seq at our sequencing depth”. However, several papers have shown that cryptic splicing in STMN2 and UNC13A can be detected in bulk tissue RNA-seq (Prudencio, JCI, 2020; Brown, Nature, 2022, Ma, Nature, 2022). Why did the authors not attempt to at least quantify STMN2 and UNC13A in their samples?

A: We thank the reviewer for this suggestion. We have now attempted to identify alternative splicing events for STMN2 and UNC13A based on our bulk RNA sequencing data (table below). We could not detect any significant cryptic splicing events:

	gene	dPSI (male)	p-value (male)	dPSI (female)	p-value (female)
Alternative first exon chr8:79611117:79611214-79636802:79611735:79611791-79636802:+	STMN2	-0.038659	1.0	-0.019715	1.0
Alternative first exon chr8:79611152:79611214-79636802:79611735:79611791-79636802:+	STMN2	-0.000285	1.0	-0.000519	1.0
Skipped exon chr8:79655062-79663580:79663661-79664815:+	STMN2	-0.000201	1.0	-0.000356	1.0
Skipped exon chr19:17611855-17616408:17616464-17617702:-	UNC13A	0.007354	1.0	0.000131	1.0
Skipped exon chr19:17648983-17649339:17649344-17649509:-	UNC13A	-0.048488	1.0	-0.028133	1.0

Furthermore, we have quantified gene expression of STMN2 and UNC13A in our transcriptomics data, but we could not identify any pattern which was associated with ALS. Therefore, we did not include this data in the supplements and rather present it in the rebuttal letter.

We modified the paragraph that referred to these events, and also mention now the challenges and limitation for the detection of cryptic exon events in bulk RNA sequencing data, as follows (lines 248-262):

“Further, transcriptome-based analyses were performed to explore DAS in our mouse models (Supplementary Fig. 13, Supplementary Table 6). It is well documented that TARDBP/TDP-43 and FUS regulate alternative splicing and transcript usage in hundreds of genes^{28,29}. TDP-43 also represses cryptic exon splicing events in STMN230 and UNC13A31. We were not able to detect any significant cryptic splicing events for these two genes in our data. However, it is important to note that such cryptic exon inclusion events are specific to neurons with TDP-43 pathology³². They are also not conserved in mice³¹ and are rarely detectable in bulk RNA-Seq at our sequencing depth³³. Despite the absence of notable cryptic splicing events, our analysis revealed differential splicing in important genes such as FLNB, CPLANE1 (involved in ciliogenesis and migration), and ATP1B1 (a membrane-bound Na⁺/K⁺-ATPase) in multiple mouse models (Supplementary Table 6). DAS analyses provided additional insights into sex-specific variations for the mouse cohorts (Supplementary Fig. 13) showing enrichment in the terms mitochondria and myelin sheath (C9orf72), GTPase activity and myelin sheath (SOD1), DNA binding and heat shock protein binding (TDP-43), and protein binding and ribosome (FUS) (Supplementary Table 5).”

Deconvolution results are better presented in the main text as box-plots rather than the heatmap used in Figure 2. But even the boxplots in the Supplemental figure are confusing as they omit the control humans and non-transgenic mice as a comparison. They also don't appear to correspond exactly - endothelial cell proportions are clearly increased in C3 samples, but that is not reflected in the heatmap. Is this why?

A: We agree with the reviewer that the deconvolution results could be presented and analyzed in a more auspicious manner. Following this suggestion, we now present updated boxplots in Supplementary Figure 14. In the boxplots, we have also added the results of the ANOVA with a post-hoc Tukey test following the suggestion of the reviewer 1. We also decided to show the mouse results in the barplots in Supplementary Figure 14e, for comparability reasons.

Furthermore, we agree that understanding the concordance between results of the boxplots and heatmap could be confusing due to different scales of the mouse and human cell type proportions. We have now separated the human and mouse box plots and have added the Pearson correlation coefficients in the updated Figure 3d to avoid confusion. Figure 3c was likewise updated, as follows:

Fig 2e - how many of the MAPK genes are significant after multiple testing?

A: Please find the exact numbers in the table below. (adjusted p-value < 0.05, FC > 1.5). Genes of the MAPK pathway were derived from <https://www.genome.jp/entry/hsa04010>.

Comparison	Number of significant genes involved in the MAPK pathway
Male only (case vs. ctrl)	3 of 73
Female only (case vs. ctrl)	0 of 2
C1 vs ctrl	3 of 88
C2 vs. ctrl	63 of 2220
C3 vs. ctrl	0 of 7
C4 vs. ctrl	11 of 470

Fig 3 - Why was the miRNA analysis not linked back to the C1-4 patient clusters?

A: We thank the reviewer for this question and we now illustrate the behavior of differentially expressed miRNAs within the four clusters presenting heatmaps with varying differential expression cutoffs. This is now shown in Fig. 2c:

Q9 - Supplemental Figures are lacking keys for colour gradients. Additionally, supplementary figures are mismatching the numbers on the reviewer portal. Would the authors please upload all supplemental figures in a single document, with the corresponding legends paired with each figure?

A: Thank you for this comment. We added explanatory gradients and legends to all supplementary figures and improved readability. All supplementary figures are now reorganized and presented in one single file, as requested by several reviewers. Furthermore, following the suggestion of the reviewers, we have now created a separate file for the supplementary figures - all the legends also appear directly after each supplementary figure, facilitating the navigation through the content. The new version of the supplementary figures can be found in the supplemental file entitled "SI figures and captions".

REVIEWER COMMENTS

Reviewer #1 (Remarks to the Author):

Thank you for your thoughtful responses and updates to your paper. While your extensive efforts are appreciated, there are a few concerns that remain:

Major comments:

Thank you for clarifying the means by which you selected the pathways/processes that were foundational to subcluster identification. Respectfully, it is my perception that there are better ways to perform subclustering. In the introduction, you cite Tam et al. and Eshima et al. as examples of recent studies that stratify their samples based on “transcriptomics and gene set enrichment analyses (GSEA).” While it is true that these studies use transcriptomics, GSEA is not a part of the cluster formation. For both of the cited examples, non-negative matrix factorization on ALS samples (not ALS and control) was used for subtype identification. Your approach is interesting in that you first separate out ALS-relevant signals and then try to identify subclusters within those signals. To my knowledge, this is not a conventional approach – as seen in the cited examples, it is more typical to identify the subclusters within the disease samples only. To venture into this realm of identifying relevant disease signatures and then trying to subdivide them, I think it would be better to do this at the gene level rather than the pathway level – e.g. use differentially expressed genes with a generous cutoff – possibly weighting based on significance. This is particularly true because you appear to be manually curating groups of pathways, which can lead to perception biases. It is not that there is an absence of evidence that these pathways are enriched in ALS relative to control, merely a concern that by selecting the most prominent pathways enriched between control and ALS, that the pathways that are most different between the ALS samples (within ALS) are missed. This issue can be observed practically in that most of the pathways identified in Figure 1f show no patterning across C1-C4. This may also explain why no subcluster patterning is observed in the proteomic data.

Consider running the WGCNA in Fig 1g with only ALS samples. Please also clarify the correlation method.

Consider running MOFA analysis with just ALS samples and then correlating with subtypes.

For enrichment analyses, I think it is important to use a significance metric that has been corrected for multiple testing. This has relevance particularly for the results in Figure 3b – where the enriched pathways for the SOD1 and TDP43 models have very low gene ratios and gene counts. While p values are not shown explicitly, I would guess that the FDRs are not significant. When I checked Supplementary Table 15, I did not see the KEGG terms in Figure 3b – was this a separate analysis? Were the terms summarized in some systematic way? For example, I did not find the ERK1/2 cascade in the male SOD1 model.

For the deconvolution analysis, conclusions are drawn on differences in cell fractions that are sometimes

less than 10%. I am wondering if the Scaden method is sufficiently sensitive to draw conclusions based on this level of change. This concern is partially justified by the results from Supplementary Figure 14f, where there is only a significant change observed between control and up to one subtype (C1-C4) for each cell type for human and most changes in SOD1, FUS, TDP43 are not significant (excepting oligodendrocytes in SOD1). Individually, I think there is an interesting story about the changes in the cell types in human subclusters and the C9orf72 mouse (oligo in the SOD1 mouse). For example, further expounding upon how “our data suggests that the neurovascular unit in ALS may be affected differently in different subclusters of patients with ALS” based on 14f would be interesting. I do, however, echo the concerns of Reviewer 3 and 4 regarding the weak relationship between the mouse models and the human data in the cell type proportions data.

Line 265: “The SOD1 model and human C1 and C2 clusters showed a decrease in glial and endothelial cells and a relative increase in excitatory neurons.” SOD1 has an increase or no change in endothelial cells.

SOD1 clusters with TDP43 and FUS, but is grouped in the description of 4c with C2 and C1 (similar to comment by Reviewer 3)

Please check Figure 4c c9orf72 microglia – a difference of 1 suggests to me that the differences in the heatmap may have been divided by the maximum difference? If so, I think the raw differences should be kept instead. Additionally, are these differences in the mean, median, etc?

Line 841: What is the rationale behind substituting the L1 loss validated in the Scaden paper for the mean squared error?

There certainly is evidence that the mouse models reflect ALS pathways and processes to an extent, but my concern is the suggestion that there is a noteworthy connection between the four mouse models and the four human subtypes. This deconvolution data seems to be the principal evidence for the following conclusion: “Furthermore, we suggest that the molecular phenotype in the four analyzed mouse models can be approximated to these subtypes, making these models potential surrogates for the molecular subgroups in humans. Notably, the oldest and most frequently used model, the SOD1.G93A mouse, correlated best with the largest clusters C1 and C2. Although not representative of all human ALS cases, our findings indicate that this model represents an important subgroup of the disease based on the identified dysregulated pathways.” (Lines 433-437) While “dysregulated pathways” are mentioned after the SOD mouse model as additional support for the claim, I only find the activation of the MAPK pathway in C1, C2, and SOD1 as support. It would be more compelling if the transcriptomic, proteomic expression signatures were directly compared between the mouse models and subclusters at the molecular (gene, protein) and pathway levels. See also Lines 485-486 “Although clear correlations were found between mouse models and molecular subtypes of sALS, the four analyzed models represent specific scenarios.”

Please clarify how Figure 3e contributes to the generalized knowledge of the paper. My understanding is that this plot is intended to compare the enriched proteomic pathways across the mouse and human datasets, yet in the analysis (lines 297-306), each model seems to be separated out. If no major cross-

species trends are worth highlighting, consider keeping this figure in the Supplementary and using the Supplementary Figure 17 here instead.

Please explain how to interpret pie charts in Figure 3e for those unfamiliar with Go-Figure plots – e.g. fraction of significant terms related to each theme? Fraction of differentially expressed genes?

Consider grouping results from 3e based on semantic criteria

Line 300: Consider including group 13 for females and excluding 17 – 17 does not nicely cluster semantically with other terms

Line 301: Groups 16, 17 not semantically related

Line 302: Consider male groups 4,6,8,11

Please clarify the provenance of the WGCNA co-expression modules in Figure 4a. Do they come from the individual WGCNAs from mouse, human? If so, human modules were not specifically mentioned in Fig 1g, and the cutoffs in the methods do not necessarily correspond to the modules kept (e.g. Darkturquoise had lower correlation than Salmon in males, Brown in females). Please also include the method by which absolute counts and significance were derived.

In the “Methods” section (lines 879-894), please include cutoff values for humans.

Minor comments:

Line 112-113: There should be only one silhouette score for condition and one for sex. I opine that a silhouette score of 0.29 does not indicate “robust” separation.

Supplementary Fig 6. Consider allowing hierarchical clustering of C1-C4 subclusters as a figure b. “clusters” in y-axis.

Line 154: consider “positively correlated with” instead of “upregulated in”

Supplementary Fig 7. Change label to “activity score” instead of “correlation.” Consider including lines separating C1-C4.

Figure 2c – if point is to show “distinctive miRNA profiles”, why are only miRNAs that are significant in more than one subcluster selected? Would this not instead show the similarities between miRNA profiles?

Supplementary Figure 8: Consider making miR-869-3-ANXA2-ANXA2 line thicker than other lines.

Supplementary Figure 10b, 12b: Add y-axis label to scatter plots.

Supplementary Figure 10a, 12a: “distance” heatmap label should be “expression”?

Line 226: Please check “negative correlation with NFH” – I think it may be positive based on Fig 2h?

Figure 2g: Consider showing disease state as subfigure

Line 1429: Space between “conditionfor”

Supplementary Fig. 14: Please include label for heatmap color bar in a). Is expression normalized in some way since scale is 0-1?

Please label the color bar in Figures 3c, 3d

Generally, the label to Figure 3 repeats the results section rather than providing key details about how to read the figure.

Please check Figure 3c “Endoc” label.

Line 294: “A cluster for the MAPK cascade...” – MAPK cascade is a single term in Supplementary Figure 17. Since this pathway is a key element of the paper, it may be worthwhile to indicate enrichment significance.

Line 309: Could not locate XPO1 labeled in Supplementary Fig 18

Supplementary Fig. 18b labels are cut off

Figure 4: Figure legend replicates much of information in results rather than describing the figure elements and aiding in individual interpretation.

Line 892-893: Consider removing “Using WGCNA, we also analyzed co-expression networks in mouse models” as redundant.

Figure 4b: Consider using the visualization in 3e to improve readability (or using this visualization in 3e).

Add arrow for MAP kinase activity to show that belongs to SOD1- male columns

Remove extra lines in diagram (e.g. gray vertical line? Small pink lines next to legend)

Clarify the meaning of the percentages along the vertical axis.

Line 920: Extra line

Reviewer #1 (Remarks on code availability):

The MAXOMOD pipeline was neatly organized and available on GitHub. However, because the underlying datasets are not available nor are clear descriptions of the files and organization requirements necessary for running the pipeline, reproducibility is difficult to assess. This analysis pipeline is not set up as a tool to be used on additional datasets by the community.

When following the instructions in the minimal README, the following error was returned at the “dvc repro” step:

```
ERROR: failed to reproduce 'proteomics_organize_samples@c9': [Errno 2] No such file or directory:
'/home/user/maxomod/datasets/consortium/C9orf72-
mouse/01_received_data/cohort/sample_annotations.csv'
```

This result is unsurprising since the datasets directory is not available in GitHub.

Reviewer #2 (Remarks to the Author):

The authors have done a good job in addressing my concerns, answered my comments and added these to their revised manuscript. There are still some formatting issues that were mentioned were addressed e.g. AGC targets are not showing the correct superscript however overall the authors have addressed my specific comments raised.

Reviewer #3 (Remarks to the Author):

Gomes and colleagues have made a significant effort to address the numerous comments/suggestions raised by the four reviewers including some key points that were essential to fully establish the relevance of targeting MAPK pathway in a sex-specific manner. In particular new evidence is provided including 1) new pre-clinical testing whereby the authors report that modulation of the MAPK pathway using trametinib delays age of onset and survival or 2) phospho-proteome analysis of potential off-targets upon trametinib treatment.

While the substantial effort made by the authors is acknowledged, which certainly improves the manuscript (with new analysis provided, reorganization of the data/figures), confusion still remains regarding some of the conclusions made. In particular it is still not clear whether the gender-related proposed differences are driven by the biased composition of the cohort, the RIN quality of the samples and/or the clinical manifestations or other.

This reviewer appreciates the explanation/new analysis (bootstrap analysis- which relies on random sampling and re-analysis of an equal number of samples) provided in attempt to convince the reader that the bias effect is not driven by the unbalanced number of females compared to males with ALS. That said, it is not clear if this is sufficient to address this question given that the random sampling was always done for one of the groups (ALS females n=16) with the same samples - if I understood correctly

what was done (so not much randomness in that group if the case). It may have been more informative to use a smaller number of samples from each group and try multiple random comparisons from different subset of samples.

While this reviewer understands that this is the cohort of samples that was available to the authors, there are a few aspects in the omics data obtained that remain perplexing. While previous work has indeed shown that males are more affected than females and that this may account for the more profound changes observed in males as the authors are proposing, what remains still unclear is why on one hand, females show no or less differences compared to males in some analyses but not others (miRNAs for example). It is really difficult to reconcile that there are no DEGs in ALS female postmortem samples compared to the control female cohort. What would be the explanation? It is even more intriguing since, from what this reviewer interprets from Figure 1f, most ALS females in the study had an early age of death which is counterintuitive and not aligned with the results. How do the authors reconcile this?

To completely rule out that some of the findings are not biased by some of the possible cofounding factors, additional clarification is needed:

- in figure Sup 4b, the RIN values are plotted, which is helpful to see that there is a large spread in the quality of the samples (yet highly expected to human material). What would be more informative is to provide a table listing each sample analyzed and its respective RIN value. Could it be that samples that have the lowest RIN were the females?
- clinical manifestations: as pointed above it is very difficult to the map clinical information with transcriptomic data per sample. Figure 1f and Sup2 are an attempt to provide such correlation. However, in this format it is impossible to clearly correlate patient-based symptoms, age of onset/death, gender, RIN etc. Table 1 does list the demographics of the patient cohort, but by pooling all control and disease patients together. To convincingly demonstrate that there is no bias, a table listing each individual analyzed should be included providing for each case demographics (gender, age of diagnosis, disease duration, age of death), clinical manifestations, time of postmortem interval, biobank source, RIN value, etc. This will help to rule out that the findings are not caused by a bias from the cohort.

Ultimately what in my view remains weak is the evidence provided in support of MAPK being a critical target in ALS. From the omics datasets it is certainly not completely evident why this particular pathway was chosen given that the changes overall are subtle (not in the top ranked GO terms either) and not always consistent. The in vivo testing only shows at best a very modest extension in survival of mutant SOD1 females (N=10) and there is no obvious effect on disease onset as mentioned by the authors. That combined with the lack of beneficial effect on the behavioral assays (including grip strength which is typically a good indicator of the muscle innervation status) does not provide solid and convincing evidence supporting the value of targeting MAPK in ALS patients.

Given the modest effect observed in females, it would be key to show that there is no effect in survival rate of ALS males to ascertain that this is not simply just due to variability in the model used. This would further strengthen that the modest effect observed in females is real to support the conclusion put forward by the authors that targeting MAPK is likely to be beneficial in females only.

Altogether, while the authors have commendably tried to address the criticisms and suggestions raised by reviewers and conducted an extensive amount of work, the cumulative findings are still perplexing

and fall short of fully supporting MAPK as an appealing therapeutic target for treating female ALS patients.

Additional points:

1. There is a word duplication in the paragraph “Human proteomic proteomic analysis confirms stronger deregulation in males”. This paragraph title should be removed and merged together with the miRNA results described in 2 a-c, following the structure of the figure.

2. Same comment as above for the paragraph “Integration of multiomic data reveals sex-specific molecular networks of ALS”.

3. The authors should clarify in the text the chosen timepoint for their analyses in the different mouse models.

4. Fig. 3c: what does the number displayed in the square represent?

5. Fig. 3d: what do the colors of the squares represent? Some squares have the same number but different colors.

6. “Overall, correlation analyses show remarkable similarities between human clusters and mouse models (Fig. 3d).” This reviewer recommends to remove “remarkable”, as the similarities are limited.

7. Fig. 3f: letter missing in the figure.

8. The effort in constructing a new figure with the main results pointing out to the importance of MAPK pathway selection is appreciated, but Figure 4a is not clear. Can the authors explain what was done in Fig. 4a? Same question regarding Fig. 4b. What does the % represent?

9. How was the target centrality calculated in Fig. 4e?

10. Referral to Fig. 5e-f missing in the main text.

Reviewer #4 (Remarks to the Author):

I am satisfied that all my comments have been addressed.

SECOND REVISION ROUND

REVIEWER COMMENTS

Reviewer #1 (Remarks to the Author):

Thank you for your thoughtful responses and updates to your paper. While your extensive efforts are appreciated, there are a few concerns that remain:

Major comments:

Thank you for clarifying the means by which you selected the pathways/processes that were foundational to subcluster identification. Respectfully, it is my perception that there are better ways to perform subclustering. In the introduction, you cite Tam et al. and Eshima et al. as examples of recent studies that stratify their samples based on “transcriptomics and gene set enrichment analyses (GSEA).” While it is true that these studies use transcriptomics, GSEA is not a part of the cluster formation. For both of the cited examples, non-negative matrix factorization on ALS samples (not ALS and control) was used for subtype identification. Your approach is interesting in that you first separate out ALS-relevant signals and then try to identify subclusters within those signals. To my knowledge, this is not a conventional approach – as seen in the cited examples, it is more typical to identify the subclusters within the disease samples only.

Q1 - To venture into this realm of identifying relevant disease signatures and then trying to subdivide them, I think it would be better to do this at the gene level rather than the pathway level – e.g. use differentially expressed genes with a generous cut off – possibly weighting based on significance. This is particularly true because you appear to be manually curating groups of pathways, which can lead to perception biases. It is not that there is an absence of evidence that these pathways are enriched in ALS relative to control, merely a concern that by selecting the most prominent pathways enriched between control and ALS, that the pathways that are most different between the ALS samples (within ALS) are missed. This issue can be observed practically in that most of the pathways identified in Figure 1f show no patterning across C1-C4. This may also explain why no subcluster patterning is observed in the proteomic data.

A: The constructive comments of the reviewer about the clustering approaches we used for our cohort are much appreciated. We would like to address the reviewer’s concern in four parts.

Selection bias in clustering approaches: Gene-based clustering is surely the most straightforward approach, yet seldom used without the introduction of a selection bias. In most cases, highly variable genes between conditions of interest are chosen, which is the absolute standard for clustering in the transcriptomics field. In our case, we wanted to observe differences between control and ALS patients and then look for sub-groups in genes and pathways that are deregulated in ALS. The observed pathways, which stem from gene expression data, were then augmented with a few pathways that were previously associated with ALS. We believe this analysis is a very viable strategy to obtain novel and (in-) validate known differences, especially if our base assumption is that these differences should only contain genes and pathways that are different between control and ALS patients. While we do understand and value the reviewer’s interest in differences observed in ALS, this is not the only question we tried to address in our analysis.

Pathway-based clustering: In fact, clustering techniques based on pathway level have been widely used in recent years, in the context of neurological disorders, but also for several other diseases (doi.org/10.1038/s41593-022-01205-3; doi.org/10.1016/j.ygeno.2018.02.012; doi.org/10.1038/s41698-021-00186-z; <https://doi.org/10.1016/j.ymeth.2019.06.017>). It’s worth noting that genes do not act in isolation, but rather in

concerted and interdependent manners. In our view, pathways are more stable, and working at the pathway level might aid in boosting the signal-to-noise ratio, increasing biological meaning during cluster identification. It also allows capturing subtle changes related to the disease mechanisms, as opposed to striking, individual genes that are not taken within any functional context. We agree that clustering based on the gene level is a valid and widely used approach, but given the different nature of the signals used as the basis for the clustering, using gene or protein expression data here would likely yield somehow different clustering, as also pointed out by the reviewer.

While we acknowledge the validity of the reviewer's suggestions to use different significance cutoffs to select gene lists used for clustering, we think this might also lead to a considerable bias towards the changes observed in male samples, since these are much more prominent than the change in females. These dimensionality differences in gene expression results faded out when accounting for the activity scores at the pathway level, and this was one of the strongest reasons why we decided to digest the raw gene expression data into functional enrichment results for our clustering analyses.

Gene-based clustering: Following the reviewer's suggestions, we tested if we could achieve a similar subdivision using gene expression only, attempting to cluster the cohort based on the 500 most variable genes. We observed that hierarchical clustering on the gene level yielded different results than our previous hierarchical pathway-based clustering (Fig a, p-value<0.001, Chi-square test of difference between clusterings). However, by performing a more advanced clustering approach, k-means clustering, on the 500 most variable genes, we observed a better congruence with our previous results (Fig b, p-value = 0.59). Interestingly, especially Clusters C1 and C2, and C3 and C4 showed mixed cluster assignments (Fig b). These clusters were, according to our initial pathway-based clustering, the most similar clusters (Fig. 1f). Therefore, we checked if a broader clustering would be possible on the gene level, matching these bigger clusters C1-C2 and C3-C4. Hierarchical clustering was in that case also not able to reproduce the pathway-based clustering approach (Fig c, p-value<0.001), however, we obtained similar clusters by using k-means clustering (Fig d, p-value=0.78). Thus, we think that we are still able to capture a similar sub-clustering also on the gene level, but we were able to detect a more fine-grained clustering using our pathway-based approach.

Figure legend: Comparison of pathway-based clustering results and top 500 variable gene-based clustering using hierarchical clustering (A) on genes (x-axis) and our pathway-based clustering on the y-axis and using minibatch k-means (B). Comparison of larger clusters C1-C2 and C3-C4 using our pathway-based clustering (C) and using hierarchical clustering (D).

pathway-based clustering approach and hierarchical clustering (C) and minibatch k-means clustering (D) on the top 500 variable genes.

Disease vs. control comparison: Finally, the selection for pathways is indeed based on the differences between ALS and control samples. This choice is by design, as outlined briefly above, since we were only interested in gene and pathway differences that are distinct between control and ALS groups and might contribute to patient subgroups. While we completely agree with the reviewer that there are other ways to address this question, we do think we found a technically correct and viable solution. With our approach, we were able to obtain quite similar results to already described clusters, almost exclusively driven by altered pathways from our transcriptome data. Furthermore, thanks to the reviewer's suggestion, we were able to show that similar clusters can also be obtained in a purely unsupervised way using the top 500 variable genes, indicating that our clusters are rather consistent with the choice of the underlying pathways (or highly variable genes) used.

Q2 - Consider running the WGCNA in Fig 1g with only ALS samples. Please also clarify the correlation method.

A: We thank the reviewer for this suggestion. About the second part of the question, the correlation method used for our WGCNA analyses was the default Pearson's correlation from this tool.

Regarding the suggestion of running these analyses only for ALS samples, we would like to point out that WGCNA finds gene modules associated with sample traits, and excluding control samples would not lead to a different interpretation of clusters themselves (correlation between a trait and co-expression of a module is independent of the other traits). Furthermore, a downside of such an approach would be the loss of information about gene module co-expression in the control samples. We believe that using the whole human cohort - including controls - provides more valuable insights, including differences between disease and non-disease states. We have outlined our view on this specific matter also in our response to Q1 of the reviewer.

To evaluate it empirically, we performed WGCNA on only ALS samples, as suggested here, treating pairwise combinations of cluster identity and biological sex as traits. We could identify gene sets that functionally resemble the original WGCNA clusters (Figure shown below). The original 4 modules of interest identified in our initial analyses could be traced back to the 4 modules in this current analysis, showing identical broad functional gene sets. Since the interpretation of the results is basically the same as the one we presented before, we have not included these new analyses in the manuscript.

Figure legend: **a.** Pearson's correlation between the cluster-sex combinations and gene modules identified using only ALS samples. **b.** The top 5 GO enrichments of the four modules that resemble the originally identified gene modules of interest. Blue, Tan, Purple, and Salmon modules correspond to Turquoise, Yellow, Tan, and Lightcyan modules respectively in the original WGCNA.

Q3 - Consider running MOFA analysis with just ALS samples and then correlating with subtypes.

A: We observed only a modest clustering by subtype in our MOFA analysis. In line with the response above, in our opinion, this favors the inclusion of the control samples so that we focus not only on differences from ALS subtypes but also on differences from the disease group in relation to the control group. This was the rationale behind the selection of the whole cohort for the MOFA approaches.

Figure legend: UMAP of MOFA (run on all samples) showing only ALS samples colored by cluster.

Q4 - For enrichment analyses, I think it is important to use a significance metric that has been corrected for multiple testing. This has relevance particularly for the results in Figure 3b – where the enriched pathways for the SOD1 and TDP43 models have very low gene ratios and gene counts. While p values are not shown explicitly, I would guess that the FDRs are not significant. When I checked Supplementary Table 15, I did not see the KEGG terms in Figure 3b – was this a separate analysis? Were the terms summarized in some systematic way? For example, I did not find the ERK1/2 cascade in the male SOD1 model.

A: We thank the reviewer for pointing this out. Indeed, for these results, we used enrichment analysis based on both overrepresentation (for GO term enrichment) and aggregated gene-set enrichment (for KEGG pathway enrichment), as done for the other datasets in this manuscript. What we report in Figure 3b are actually the results for GO terms instead of the previously stated KEGG pathways. We accidentally added the table for aggregated gene set enrichment including only KEGG results in the supplements, which led to the inconsistencies mentioned, for which we apologize.

To further clarify the reviewer's questions: KEGG and GO analyses were done separately, and the exact methods have been updated in the manuscript. The p-values were originally not shown explicitly in Fig. 3b due to space limitations and readability of the results, but these can be found in full in the updated Supplementary Table 15. The enrichment for KEGG pathways is corrected for multiple testing, as indicated in p-adjusted values reported in the supplementary material, while the GO term enrichment considers non-adjusted p-values, as recommended and embedded in default settings for the script used for this analysis (topGO). The reasoning for this is related to the way the algorithm calculates the p-values, and it is extensively described in the original documentation of the package (<https://bioconductor.org/packages/release/bioc/vignettes/topGO/inst/doc/topGO.pdf>, section 6.2). We deemed it important to report all results but we decided to include only the representative GO term enrichment results (top 5 per category) in the main figure. No summary or curation was performed with the enriched terms - we display the top hits based on significance in the figure. We have now added the corresponding results to the supplement as new tabs in the existing Supplementary Table 15 for humans and mouse models, including all the findings listed in Figure 3b that could not be located by the reviewer previously.

Q5 - For the deconvolution analysis, conclusions are drawn on differences in cell fractions that are sometimes less than 10%. I am wondering if the Scaden method is sufficiently sensitive to draw conclusions based on this level of change. This concern is partially justified by the results from Supplementary Figure 14f, where there is only a significant change observed between control and up to one subtype (C1-C4) for each cell type for human and most changes in SOD1, FUS, TDP43 are not significant (excepting oligodendrocytes in SOD1). Individually, I think there is an interesting story about

the changes in the cell types in human subclusters and the C9orf72 mouse (oligo in the SOD1 mouse). For example, further expounding upon how “our data suggests that the neurovascular unit in ALS may be affected differently in different subclusters of patients with ALS” based on 14f would be interesting. I do, however, echo the concerns of Reviewer 3 and 4 regarding the weak relationship between the mouse models and the human data in the cell type proportions data.

A: The reviewer's comment raises a valid concern about the sensitivity of Scaden in detecting subtle changes in cell fractions, particularly when they are less than 10%. This is shown in detail in Fig 4c in the original manuscript by Menden et al., 2020 (<https://doi.org/10.1126/sciadv.aba2619>). This concern is supported by the limited significant changes observed in Supplementary Figure 14f, where only a few subtypes show significant differences between control and experimental conditions. While the changes in individual cell types depicted here may not be statistically significant, our clustering approaches reveal trends that, in our opinion, are very informative and therefore were reported in detail. Nevertheless, we have decided to tone down the discussion about the relationship between the cell composition results for the mouse models and the human data, following the concerns of this and other reviewers.

We share the interest in investigating the neurovascular unit based on our findings in ALS subclusters. We have actually performed this early on but decided to exclude the data from the manuscript due to the mentioned potential limitations of Scaden to reliably detect and quantify scarce cell populations. Given these deliberations of the reviewer and ourselves, we have decided to not include these findings in the manuscript.

Figure legend: Barplots showing the fold changes in cell type enrichment as estimated by EWCE. PVF: perivascular fibroblast, EC: endothelial cell, VSMC: vascular smooth muscle, PC: pericyte, MG: microglia, AC: astrocyte, OD: oligodendrocyte, OPC: oligodendrocyte precursor, IN: interneuron, EN: excitatory neuron. Stars indicate the significance of the enrichment and refer to the p-value output of EWCE ($p < 0.05$).

Q6 - Line 265: “The SOD1 model and human C1 and C2 clusters showed a decrease in glial and endothelial cells and a relative increase in excitatory neurons.” SOD1 has an increase or no change in endothelial cells.

SOD1 clusters with TDP43 and FUS, but is grouped in the description of 4c with C2 and C1 (similar to comment by Reviewer 3)

A: We apologize for this inconsistency. We have reworked the text to properly describe the results and made sure that the description for the SOD1 model does not group with the clusters C1-2 in terms of the hierarchical clustering for the cell type compositions.

Q7 - Please check Figure 4c c9orf72 microglia – a difference of 1 suggests to me that the differences in the heatmap may have been divided by the maximum difference? If so, I think the raw differences should be kept instead. Additionally, are these differences in the mean, median, etc?

A: We believe that the reviewer is referring to Figure 3c here. The clustering for this figure was performed on the relative differences in cell type abundances between control and ALS samples using the following formula to account for differing cell type abundances (e.g., a change of 1 percentage point in microglia abundance is larger than a change of 1 percentage point in the abundance of neurons):

$$\text{Relative difference} = (\text{median composition in ALS} - \text{median composition in controls}) / (\text{median in composition in control samples}).$$

So, a difference of 1 (rounded to the first decimal point) in C9orf72 in microglia indicates that microglia abundance has been doubled in C9orf72 samples. Following the reviewer's comment, we have added this information to the methods section (lines 853-857), and also include an additional heatmap to this rebuttal letter corresponding to raw differences from what is displayed in the original Figure 3c, for the reviewer's reference.

Figure legend: Heatmap showing differences (ALS vs. ctrl) in the estimated cell-type fractions (microglia, inhibitory neurons, excitatory neurons, oligodendrocytes, endothelial cells, oligodendrocyte precursor cell, astrocytes) for human subclusters and mouse ALS models. Raw values are depicted.

Q8 - Line 841: What is the rationale behind substituting the L1 loss validated in the Scaden paper for the mean squared error?

A: In the original Scaden manuscript (<https://doi.org/10.1126/sciadv.aba2619>), we have tested both the L1 and L2 loss. We demonstrated that the choice of which made very little difference in the performance. While the L1 loss was described in the manuscript, the L2 is actually in the co-published code. To prove this point to the reviewer and reader, we have computed the correlation of cell fraction estimates for the human cohort using L1 or L2 loss in Scaden. We have clarified this in the re-revised manuscript under Cell-type deconvolution analyses in the Methods section. We would like to apologize for the regularization inconsistency between the original manuscript and published code.

Figure legend: Correlation between Scaden with L1 and L2 errors for the human cohort.

Q9 - There certainly is evidence that the mouse models reflect ALS pathways and processes to an extent, but my concern is the suggestion that there is a noteworthy connection between the four mouse models and the four human subtypes. This deconvolution data seems to be the principal evidence for the following conclusion: “Furthermore, we suggest that the molecular phenotype in the four analyzed mouse models can be approximated to these subtypes, making these models potential surrogates for the molecular subgroups in humans. Notably, the oldest and most frequently used model, the SOD1.G93A mouse, correlated best with the largest clusters C1 and C2. Although not representative of all human ALS cases, our findings indicate that this model represents and import subgroup of the disease based on the identified dysregulated pathways.” (Lines 433-437)

While “dysregulated pathways” are mentioned after the SOD mouse model as additional support for the claim, I only find the activation of the MAPK pathway in C1, C2, and SOD1 as support. It would be more compelling if the transcriptomic, proteomic expression signatures were directly compared between the mouse models and subclusters at the molecular (gene, protein) and pathway levels. See also Lines 485-486 “Although clear correlations were found between mouse models and molecular subtypes of sALS, the four analyzed models represent specific scenarios.”

A: We agree with the reviewer that the claims about correlations between the mouse models and the ALS clusters have to be drawn with caution. Following the suggestions, we have decided to rework these passages in the discussion to tone down these claims and focus on the specific findings that are captured in the data. As also suggested in one of the questions above, we do not highlight an obvious correlation of the SOD1 mouse model with human clusters C1-2, and rather report specific features that are similar between all analyzed cohorts.

Lastly, we would like to point out that the cell type deconvolution data is entirely based on RNA sequencing data, and common findings between mouse models and human data in other omics layers are still approached in a lot of detail, so that all omics were extensively explored regarding these similarities across the analyzed cohorts. This can be found, for instance, in lines 291-299 where we describe REVIGO results for mouse models in terms of what has been found in humans, as well as for several other sections (e.g., for proteomics with GOFigure analyses, for miRNAs with the correlation heatmaps, for integrative approaches with MOFA and triplets/quadruplets plots, to name a few).

Q10 - Please clarify how Figure 3e contributes to the generalized knowledge of the paper. My understanding is that this plot is intended to compare the enriched proteomic pathways across the mouse and human datasets, yet in the analysis (lines 297-306), each model seems to be separated out. If no major cross-species trends are worth highlighting, consider keeping this figure in the Supplementary and using the Supplementary Figure 17 here instead.

A: We thank the reviewer for this suggestion. Following the request, we have moved the content from the former Supplementary Figure 17 with the REVIGO analysis for the mouse models into the main Figure 3. For space limitations in the main figure, we keep the diagrams with 5 or more enriched terms, and indicate this in the legend. The GO-Figure panels that were formerly in Fig. 3e are now moved into the new Supplementary Figure 17.

Q11 - Please explain how to interpret pie charts in Figure 3e for those unfamiliar with Go-Figure plots – e.g. fraction of significant terms related to each theme? Fraction of differentially expressed genes?

A: As suggested by the reviewer, we swapped the figures and now display the REVIGO results in the main Figure 3. Furthermore, we added a more detailed description to the legend of Supplementary Figure 17 on how the GO-Figure pie-charts should be interpreted:

“GO-Figure comparative analysis of enrichment results for proteomics results. The GO-Figure analysis groups semantically similar pathways into bigger groups (groups 1-20), separately for males (a) and females (b). The size of the circle of each group corresponds to the total number of pathways included in the group and it is colored by the fraction of pathways from each model. The analysis showed clustering for differentiation and development in human females [...].”

Q12 - Consider grouping results from 3e based on semantic criteria

A: Thank you for the suggestion. The pathways are already grouped into bigger semantically similar groups in an unsupervised way. We think that the current grouping is the ideal way to display these results, not to confound automatically and manually grouped results.

Q13 - Line 300: Consider including group 13 for females and excluding 17 – 17 does not nicely cluster semantically with other terms

Line 301: Groups 16, 17 not semantically related

Line 302: Consider male groups 4,6,8,11

A: We have adjusted the highlighted groups as indicated by the reviewer. For the last point, we decided to keep the clusters of terms as 4, 6 and 11, since group 8 (nervous system development) is quite different from the rest (immune/defense response).

Q14 - Please clarify the provenance of the WGCNA co-expression modules in Figure 4a. Do they come from the individual WGCNAs from mouse, human? If so, human modules were not specifically mentioned in Fig 1g, and the cutoffs in the methods do not necessarily correspond to the modules kept (e.g. Darkturquoise had lower correlation than Salmon in males, Brown in females). Please also include the method by which absolute counts and significance were derived.

A: The co-expression modules depicted in Figure 4a originate from individual Weighted Gene Correlation Network Analysis (WGCNA) conducted separately for mouse and human datasets. These modules were selected by filtering for terms related to the MAPK/MAP kinase pathway to highlight the dysregulation of the pathway. For determining absolute counts and significance, we used the clusterProfiler function enrichGO. This information has been included in the Method section under the subsection "Weighted Gene Correlation Network Analysis (WGCNA)".

In contrast, Figure 1g focuses on highlighting four gene modules where the top hits correlate most strongly with pathways of interest identified through clustering analysis. These modules serve a different purpose from those presented in Figure 4a, as they are chosen based on their correlation with specific pathways rather than their distribution of MAPK activity throughout different models and omics datasets.

Q15 - In the “Methods” section (lines 879-894), please include cutoff values for humans.

A: The cut-off value for humans is 0 for module merging and correlations and is added to the methods section now (line 907-908).

Minor comments:

- Line 112-113: There should be only one silhouette score for condition and one for sex. I opine that a silhouette score of 0.29 does not indicate “robust” separation.

A: The silhouette scores were computed per sample using either the condition or the sex annotation. Afterwards, the scores were averaged by case and ctrl, or male/female respectively, to measure the cluster density per group. These silhouette scores were reported in the manuscript and are smaller for ALS and CTR than the ones for sex indicating a stronger separation driven by sex (score: 0.11 [ALS]; - 0.03 [CTR]; sex (silhouette score: 0.29 [male], 0.15 [female]).

- Supplementary Fig 6. Consider allowing hierarchical clustering of C1-C4 subclusters as a figure b. “clusters” in y-axis.

A: This panel is now added to the referred SI Figure. The typo in the axis was also fixed.

- Line 154: consider “positively correlated with” instead of “upregulated in”

A: Thank you for the suggestion, we changed the text accordingly.

- Supplementary Fig 7. Change label to “activity score” instead of “correlation.” Consider including lines separating C1-C4.

A: The figure was adjusted accordingly.

- Figure 2c – if point is to show “distinctive miRNA profiles”, why are only miRNAs that are significant in more than one subcluster selected? Would this not instead show the similarities between miRNA profiles?

A: We apologize for the inconsistencies and changed the manuscript text and figure legends accordingly. The cutoff criteria were chosen to select miRNAs expressed in at least one of the subclusters.

- Supplementary Figure 8: Consider making miR-869-3-ANXA2-ANXA2 line thicker than other lines.

A: We have made the line thicker and also highlighted ANXA2 for better visualization of the triplet.

- Supplementary Figure 10b, 12b: Add y-axis label to scatter plots.

A: The figure was adjusted accordingly.

- Supplementary Figure 10a, 12a: “distance” heatmap label should be “expression”?

A: The figure was adjusted accordingly.

- Line 226: Please check “negative correlation with NFH” – I think it may be positive based on Fig 2h?

A: Thank you, we corrected the text accordingly.

- Figure 2g: Consider showing disease state as subfigure

A: We have now added a new subpanel (Fig. 2h) with a UMAP plot colored by disease state, as requested.

- Line 1429: Space between “condition for”

A: This was corrected in the text.

- Supplementary Fig. 14: Please include label for heatmap color bar in a). Is expression normalized in some way since scale is 0-1?

A: Thank you for the question. These are expression values and min-max normalized. This was added to the legend.

- Please label the color bar in Figures 3c, 3d

A: This was corrected in the figure.

- Generally, the label to Figure 3 repeats the results section rather than providing key details about how to read the figure.

A: We have modified the legend for this figure to keep only the most important details about the panels, rather than describing the results.

- Please check Figure 3c “Endoc” label.

A: We corrected the label and amended the caption.

- Line 294: “A cluster for the MAPK cascade...” – MAPK cascade is a single term in Supplementary Figure 17. Since this pathway is a key element of the paper, it may be worthwhile to indicate enrichment significance.

A: We have now indicated the significance of the MAPK pathway, as requested (adjusted p-value = 0.04). This can be found now in line 298.

- Line 309: Could not locate XPO1 labeled in Supplementary Fig 18

A: Thank you for mentioning this. XPO1 is not labeled in Supplementary Figure 18, but mentioned in Supplementary Table 13. Therefore, we clarified the references in the text as follows:

“In addition, multiple mouse omics datasets, including phospho-proteomics, were also integrated to visualize valid interacting partners based on their expression levels in so-called quadruplets. This analysis identified a coherent regulation of GFAP, SQSTM1, ATXN10 (Supplementary Fig. 18, Supplementary Table 13) and XPO1 (Supplementary Table 13).”

- Supplementary Fig. 18b labels are cut off

A: This was corrected in the figure.

- Figure 4: Figure legend replicates much of information in results rather than describing the figure elements and aiding in individual interpretation.

A: Similar to what we did for Figure 3, we have modified the legend for this figure to keep only important details about the panels.

- Line 892-893: Consider removing “Using WGCNA, we also analyzed co-expression networks in mouse models” as redundant.

A: Thank you, we removed the sentence.

- Figure 4b: Consider using the visualization in 3e to improve readability (or using this visualization in 3e).

A: For the assessment of enrichment concerning differentially spliced transcripts, our methodology encompassed overrepresentation analysis, as demonstrated in the circular bar chart. The enrichment analysis presented is similar to the representation depicted in Figure 1e and the mouse enrichment figure (Fig. 3b). In contrast, in Suppl. Fig. 17 (formerly 3e), we employed clustering within a semantic framework to condense the abundance of enriched terms. This decision was driven by the considerable volume of such terms and the aim to enhance comprehension by consolidating semantically similar terms into broader groups. These two methodologies represent distinct approaches with different objectives.

- Add arrow for MAP kinase activity to show that belongs to SOD1- male columns

A: We added an arrow as suggested.

- Remove extra lines in diagram (e.g. gray vertical line? Small pink lines next to legend)

A: The gray lines featured in the diagram denote percentages (0, 5, 10, 20, etc.), thereby alleviating the necessity of annotating each bar individually. We are afraid that we cannot find any pink lines in this figure.

- Clarify the meaning of the percentages along the vertical axis.

A: We added a more detailed figure caption for Fig 4b, describing the vertical axis, the enrichment method and the used threshold:

“GO-Enrichment results for DAS genes. Bar heights represent the fraction of Gene Ratio of differentially alternatively spliced genes in the pathways. All pathways have an adjusted p-value < 0.1.”

- Line 920: Extra line

A: Thank you. We removed the extra line.

Reviewer #1 (Remarks on code availability):

Q16 - The MAXOMOD pipeline was neatly organized and available on GitHub. However, because the underlying datasets are not available nor are clear descriptions of the files and organization requirements necessary for running the pipeline, reproducibility is difficult to assess. This analysis pipeline is not set up as a tool to be used on additional datasets by the community.

When following the instructions in the minimal README, the following error was returned at the “dvc repro” step:

```
ERROR: failed to reproduce 'proteomics_organize_samples@c9': [Errno 2] No such file or directory:
'/home/user/maxomod/datasets/consortium/C9orf72-mouse/01_received_data/cohort/sample_annotations.csv'
```

This result is unsurprising since the datasets directory is not available in GitHub.

A: We highly appreciate that the reviewer is testing our pipeline. Uploading raw human data to public repositories like GitHub is generally not recommended due to privacy concerns and ethical issues. Instead, it's advisable to store sensitive data in secure and compliant environments, adhering to relevant regulations. Moreover, there are special storage servers for high throughput sequencing data, dealing with large data sizes. Thus, we recommend that the reviewer downloads the data from the repositories that are linked in the manuscript and run the pipeline then. The information is listed under data availability in the manuscript:

Data availability

Raw RNA-Seq and processed gene expression data were deposited to the National Center for Biotechnology Information Gene Expression Omnibus database (GSE234246). Encrypted raw RNA-Seq data for the human cohort were deposited to the European Genome Phenome Archive (registered study: EGAS00001007318). Proteomics and phosphoproteomics datasets have been deposited in the ProteomeXchange Consortium database with the identifiers PXD043300 and PXD043297, respectively.

Reviewer #2 (Remarks to the Author):

The authors have done a good job in addressing my concerns, answered my comments and added these to their revised manuscript.

Q1 - There are still some formatting issues that were mentioned were addressed e.g. AGC targets are not showing the correct superscript however overall the authors have addressed my specific comments raised.

A: We apologize for this remaining inconsistency in the font for the aforementioned sentence. We have taken this comment carefully into account in the last round, and set the text to superscript, as written in our former response. While we are again making sure that this is fixed in the text, we think that the problem might lie in the file conversion process during the submission. Therefore, if the same error is repeated upon this submission, we will contact the editorial office, so it can be fixed during typesetting.

Reviewer #3 (Remarks to the Author):

Gomes and colleagues have made a significant effort to address the numerous comments/suggestions raised by the four reviewers including some key points that were essential to fully establish the relevance of targeting MAPK pathway in a sex-specific manner. In particular new evidence is provided including 1) new pre-clinical testing whereby the authors report that modulation of the MAPK pathway using trametinib delays age of onset and survival or 2) phospho-proteome analysis of potential off-targets upon trametinib treatment. While the substantial effort made by the authors is acknowledged, which certainly improves the manuscript (with new analysis provided, reorganization of the data/figures), confusion still remains regarding some of the conclusions made.

Q1 - In particular it is still not clear whether the gender-related proposed differences are driven by the biased composition of the cohort, the RIN quality of the samples and/or the clinical manifestations or other. This reviewer appreciates the explanation/new analysis (bootstrap analysis- which relies on random sampling and re-analysis of an equal number of samples) provided in attempt to convince the reader that the bias effect is not driven by the unbalanced number of females compared to males with ALS. That said, it is not clear if this is sufficient to address this question given that the random sampling was always done for one of the groups (ALS females n=16) with the same samples - if I understood correctly what was done (so not much randomness in that group if the case). It may have been more informative to use a smaller number of samples from each group and try multiple random comparisons from different subset of samples.

A: We recognize the importance of clarifying whether observed differences are influenced by sample composition bias. Our decision to use bootstrapping analysis, with all available female samples and randomly sampled equal numbers of male samples, was made to directly assess the impact of sample size variation. Our approach provides a robust examination of gender-related gene expression differences. Furthermore, our analysis consistently demonstrates more downregulated genes in males, even under balanced sampling conditions. We believe this underscores the robustness of our findings and shows that sample size bias is not the sole driver of observed gender-related differences.

Using smaller sample sizes for highly variable human samples will lead to a decrease in the power of the analysis and, consequently, to fewer significant DEGs. Thus Ching, Huang & Garmire (2014) [<https://www.ncbi.nlm.nih.gov/pmc/articles/PMC4201821/>] promote the use of larger sample sizes for differential gene expression analysis whenever possible.

To further address the reviewer's concerns, we repeated our bootstrapping approach (20 bootstraps) using eight random samples per condition and sex as requested. As expected, we observed that in eight bootstraps for females and seven bootstraps for males no DEGs were significantly deregulated. In summary, while we appreciate the constructive critique by the reviewer, we believe that we have performed the best possible bootstrapping experiment to validate male-predominant expression changes.

Q2 - While this reviewer understands that this is the cohort of samples that was available to the authors, there are a few aspects in the omics data obtained that remain perplexing. While previous work has indeed shown that males are more affected than females and that this may account for the more profound changes observed in males as the authors are proposing, what remains still unclear is why on one hand, females show no or less differences compared to males in some analyses but not others (miRNAs for example). It is really difficult to reconcile that there are no DEGs in ALS female postmortem samples compared to the control female cohort. What would be the explanation? It is even more intriguing since, from what this reviewer interprets from Figure 1f, most ALS females in the study had an early age of death which is counterintuitive and not aligned with the results. How do the authors reconcile this?

A: The reviewer raises a couple of interesting points in regard to the sex-specific differences that we observed in our analysis. To summarize our findings, we here present again the differential expression of all reported entities in a table:

Human omics	sex	Total	Down	Up	cut-off
RNA	Female	2	1	1	padj value 0.05, log2(1.5)
RNA	Male	73	70	3	padj value 0.05, log2(1.5)
miRNA	Female	9	4	5	p-value 0.1, log2(1.5)
miRNA	Male	17	15	2	p-value 0.1, log2(1.5)
hairpinRNA	Female	24	12	12	p-value 0.1, log2(1.5)
hairpinRNA	Male	99	86	13	p-value 0.1, log2(1.5)
Proteomic	Female	251	119	132	p value 0.1, log2(0)
Proteomic	Male	379	168	211	p value 0.1, log2(0)

Clearly, there are sex-specific differences in all analyzed entities and females show consistently less deregulation than males. These DE values are based on statistical cut-offs that are given in the last column and which, by nature of such cut-offs, are in some way arbitrary. We obtain the values given below with the cut-offs presented, but we admit that the DE numbers would be higher with less stringent cut-offs. As such, we have to disagree with this reviewer that “there are no DEGs in ALS female postmortem samples compared to the control female cohort”, but with the present cut-offs we consistently obtain lower numbers in females. As this reviewer also points out, there has been previous evidence to suggest sex-specific differences in ALS and we also mention this in our discussion. To follow up with this point in a data-driven manner, we have also conducted DEG analyses controlled for age and found that it did not impact the number nor the quality of DEGs for the female ALS vs. control comparison. Only three DE genes were found when controlling for sex compared to the two when we did not control for sex. Moreover, the two genes found in the age-independent analysis were included in the list of three genes found when controlling for age.

The reviewer’s question about the correlation with the age of death is indeed intriguing and as such suggests that there is a linear relationship between survival and the number of postmortem DEG identified in the prefrontal cortex. As we point out in our introduction, postmortem changes necessarily only capture disease end stage. We cannot say whether these alterations have been more pronounced in an earlier stage of the disease and thus may have contributed to faster disease progression. Such analyses in living human patients are hardly to be ever performed. In addition, we present data on one specific brain region. Survival of a patient is a very global read-out and many different parameters contribute to it. If survival is, for example, limited by respiratory failure (as is mostly the case in patients with ALS), alterations in cervical and thoracic motoneurons responsible for the respiratory function and which are not part of our analysis may be much more important than alterations in the prefrontal cortex. We therefore do not see an inconsistency in the data presented and hope that this reviewer can follow our discussion.

Q3 - To completely rule out that some of the findings are not biased by some of the possible confounding factors, additional clarification is needed:

Q3a - in figure Sup 4b, the RIN values are plotted, which is helpful to see that there is a large spread in the quality of the samples (yet highly expected to human material). What would be more informative is to provide a table listing each sample analyzed and its respective RIN value. Could it be that samples that have the lowest RIN were the females?

A: As suggested, we have included the individual RIN values for the human postmortem brain samples in Supplementary Table 1. We plotted the RIN values for the whole cohort, splitting the data by sex. Regarding the concern of the reviewer about the distribution of the RIN values for the different sexes, our data show no differences related to sex. Therefore, no specific effects related to sample quality can be attributed to the differential results between male and female samples in RNA sequencing analyses.

Figure legend: Bar plots showing RIN values for the human cohort of prefrontal cortex samples

Q3b - clinical manifestations: as pointed above it is very difficult to the map clinical information with transcriptomic data per sample. Figure 1f and Sup2 are an attempt to provide such correlation. However, in this format it is impossible to clearly correlate patient-based symptoms, age of onset/death, gender, RIN etc. Table 1 does list the demographics of the patient cohort, but by pooling all control and disease patients together. To convincingly demonstrate that there is no bias, a table listing each individual analyzed should be included providing for each case demographics (gender, age of diagnosis, disease duration, age of death), clinical manifestations, time of postmortem interval, biobank source, RIN value, etc. This will help to rule out that the findings are not caused by a bias from the cohort.

A: As reported in the answer above, all RIN values were added in Supplementary Table 1. This table lists patients individually and includes all available clinical data as provided by the brain banks. The main Table 1 referred to here by the reviewer includes only a summary of the full information provided in the supplementary material, since we wouldn't be able to report individual details in the main content.

Q4 - Ultimately what in my view remains weak is the evidence provided in support of MAPK being a critical target in ALS. From the omics datasets it is certainly not completely evident why this particular pathway was chosen given that the changes overall are subtle (not in the top ranked GO terms either) and not always consistent. The in vivo testing only shows at best a very modest extension in survival of mutant SOD1 females (N=10) and there is no obvious effect on disease onset as mentioned by the authors. That combined with the lack of beneficial effect on the behavioral assays (including grip strength which is typically a good indicator of the muscle innervation status) does not provide solid and convincing evidence supporting the value of targeting MAPK in ALS patients.

Given the modest effect observed in females, it would be key to show that there is no effect in survival rate of ALS males to ascertain that this is not simply just due to variability in the model used. This would further strengthen that the modest effect observed in females is real to support the conclusion put forward

by the authors that targeting MAPK is likely to be beneficial in females only. Altogether, while the authors have commendably tried to address the criticisms and suggestions raised by reviewers and conducted an extensive amount of work, the cumulative findings are still perplexing and fall short of fully supporting MAPK as an appealing therapeutic target for treating female ALS patients.

A: We thank the reviewer again for the constructive criticism. In our previous rebuttals, we have provided as much evidence as possible supporting the proposed role of MAPK in ALS. We investigated the comprehensive data we have gathered, emphasizing the wealth of evidence coming from omics datasets, biochemical analyses, imaging studies, and behavioral assays. All these results - alongside further evidence available from the literature in the field - in our view underscore the significance of MAPK signaling in ALS pathogenesis. We report that this deregulation is consistently found across all omics levels, not only in humans but also in different mouse models. It is also worth noting that these profiling experiments were completely independent. We respectfully disagree with the reviewer about our evidence for MAPK being weak. In our opinion, given the frequent occurrence of this pathway in multiomic results, highlighting the MAPK pathway is valid. To make our findings more concise, we have recently summarized all evidence in a dedicated figure, as specifically requested in the previous review round. Furthermore, we have also conducted large sets of *in vitro* and *in vivo* experiments, underscoring the role of the MAPK pathway in ALS-related mechanisms. We would like to reiterate some of the most important findings related to the validation experiments, to justify our focus on this pathway:

We have initially demonstrated that modulating the MAPK pathway with trametinib improved the survival and neurite growth in glutamate-stressed cultured neurons, thus having an impact on two important processes relevant to ALS (Fig. 5). Next, we have shown that SOD1 female mice treated with trametinib showed decreased activation of MAPK/ERK pathway, accompanied by a significant decrease in neurodegeneration (neurofilament light chain; NfL) and protein aggregation (Triton-insoluble SOD1 and ubiquitinated proteins) likely by activation of autophagy pathway (decreased p62) (Fig. 6), as demonstrated by Chun et al., 2022¹. In fact, our *in vivo* experiments with mutant SOD1 female mice treated with trametinib showed a significant delay in the age of paralysis ($p=0.0376$ by Student's t-test) and the onset of the disease ($p=0.0177$ by Log-rank Mantel-Cox test), as well as a significantly improved survival ($p=0.0405$ by Log-rank Mantel-Cox test). We have added the statistical values to the legend and the panel in Figure 6j to improve clarity. In our opinion, these effects are clear and the experiments have sufficient power to support our claims. Altogether, these results add substantial evidence in favor of trametinib as a disease-modifying drug in ALS. We thus conclude that the MAPK pathway is abnormally activated in ALS, and that restoring this pathway with trametinib might be beneficial for the course of the disease.

We also would like to highlight the importance of translatability for ALS research. This is particularly important in light of our focus on Trametinib, its license status, and its potential therapeutic implications. We also draw attention to the current approval of the SOD1-ASO Tofersen. The use of this drug for SOD1-ALS patients has been recently approved by the FDA solely based on the drug's ability to decrease blood levels of NfL. These points were also very important for the selection of pathways to be tested in this project.

Aiming to provide even further evidence about the effects of trametinib and the relevance of the MAPK pathway in the context of ALS, we have now conducted a new set of experiments that investigate the extent of muscle denervation in SOD1 mutant mice. In brief, we isolated total RNA from gastrocnemius muscles from male and female SOD1 mice after trametinib (or vehicle) treatment, and evaluated the relative expression in the Acetylcholine Receptor γ subunit (AChR γ) by qRT-PCR. The AChR in adult skeletal muscle endplate forms a heteropentamer consisting of $\alpha 1\beta 1\delta$ subunits with one ϵ -subunit that is replaced by a fetal γ -subunit following muscle denervation². A progressive increase in γ -subunit AChR transcript levels has been observed in SOD1^{G93A} mice³ and a significant downregulation was achieved with a neuroprotective treatment⁴. Our results showed that trametinib affects denervation differently in male and female mice. Male mice treated with trametinib showed a significant increase in the expression

for AChR in comparison to vehicle-treated mice, while the opposite is observed for female mice treated with trametinib indicating reduced denervation. This finding goes in line with the previous results, indicating a beneficial effect of trametinib treatment on female mice - also in terms of preventing the denervation process expected in this model. These new results are presented in the new Supplementary Figure 24, and are also incorporated in the text (lines 392-395).

Figure legend: q-RT-PCR results for AChR- γ mRNA transcripts in gastrocnemius muscle of trametinib-treated and untreated (vehicle) SOD1^{G93A} male (a) and female (b) mice, at 16 weeks of age. Data (2- $\Delta\Delta$ Ct) values; mean \pm SEM; n = 3/4 in each experimental group) are normalized to β -actin and expressed as relative mRNA. *p < 0.05, Student's t-test.

Due to a number of reasons - including the scope and timeline of this revision, ethical considerations, and the principles of the 3Rs (Replacement, Reduction, Refinement) - we think that conducting additional experiments in male mice is not justifiable. No evidence hints at the value of pursuing further experiments.

In summary, given all the evidence above, we are confident to suggest the MAPK pathway and Trametinib as valid therapeutic targets in ALS. We would like to reiterate that a multiomic analysis of this width, including four animal models of ALS, clearly contains evidence for more than one promising therapeutic pathway and that the selection of targets for validation is always based on multiple layers of evidence - its ranking within GO terms being only one of them. We are excited to validate further targets resulting from this work in subsequent projects and we also hope to have provided a resource to other researchers in the field for such endeavors.

References

1. Chun, Yoon Sun, et al. "MEK1/2 inhibition rescues neurodegeneration by TFEB-mediated activation of autophagic lysosomal function in a model of Alzheimer's Disease." *Molecular psychiatry* 27.11 (2022): 4770-4780.
2. Goldman D, Brenner HR, Heinemann S. Acetylcholine receptor alpha-, beta-, gamma-, and delta-subunit mRNA levels are regulated by muscle activity. *Neuron*. 1988 Jun;1(4):329-33. doi: 10.1016/0896-6273(88)90081-5. PMID: 3272739.
3. Margotta C, Fabrizio P, Ceccanti M, Cambieri C, Ruffolo G, D'Agostino J, Trolese MC, Cifelli P, Alfano V, Laurini C, Scaricamazza S, Ferri A, Sorarù G, Palma E, Inghilleri M, Bendotti C, Nardo G. Immune-mediated myogenesis and acetylcholine receptor clustering promote a slow disease progression in ALS mouse models. *Inflamm Regen*. 2023 Mar 9;43(1):19. doi: 10.1186/s41232-023-00270-w. Erratum in: *Inflamm Regen*. 2023 Apr 19;43(1):25. PMID: 36895050; PMCID: PMC9996869.
4. Trolese MC, Scarpa C, Melfi V, Fabrizio P, Sironi F, Rossi M, Bendotti C, Nardo G. Boosting the peripheral immune response in the skeletal muscles improved motor function in ALS transgenic mice. *Mol Ther*. 2022 Aug 3;30(8):2760-2784. doi: 10.1016/j.jymthe.2022.04.018. Epub 2022 Apr 27. PMID: 35477657; PMCID: PMC9372324.

Additional points:

1. There is a word duplication in the paragraph “Human proteomic proteomic analysis confirms stronger deregulation in males”. This paragraph title should be removed and merged together with the miRNA results described in 2 a-c, following the structure of the figure.

A: We thank the reviewer for this suggestion and adopted all changes to the manuscript. The two mentioned sections were merged, and now encompass all the content in Figure 2 in one single section.

2. Same comment as above for the paragraph “Integration of multiomic data reveals sex-specific molecular networks of ALS”.

A: Thank you for this suggestion. We have also merged that section as suggested by the reviewer, so the whole figure is covered in the same section.

3. The authors should clarify in the text the chosen timepoint for their analyses in the different mouse models.

A: We have mentioned previously in the manuscript that we have chosen to work with presymptomatic/early disease stages for the mouse models to allow better comparability to the human results (since we also work with an area that was not heavily affected in human ALS brains). This was previously mentioned in the first section of the results (current lines 101-103) and has now been extended, following the reviewer's suggestions. The original publications for each of the mouse models were considered when selecting the presymptomatic time points, and this is also referenced in the manuscript accordingly.

4. Fig. 3c: what does the number displayed in the square represent?

A: Fig. 3c shows the relative difference in the median cell type abundances computed using the following formulation. The information is added in the Method section under cell type deconvolution (lines 853-857).

$$\text{Relative difference} = (\text{median composition in ALS} - \text{median composition in controls}) / (\text{median in composition in control samples}).$$

5. Fig. 3d: what do the colors of the squares represent? Some squares have the same number but different colors.

A: For this analysis, the values represent Pearson correlations. We updated the figure caption and the legend of the color bar accordingly.

6. “Overall, correlation analyses show remarkable similarities between human clusters and mouse models (Fig. 3d).” This reviewer recommends to remove “remarkable”, as the similarities are limited.

A: We agree with the reviewer that the claims about correlations between the mouse models and the ALS clusters have to be drawn with caution. We have reworked the whole section, as suggested. This sentence was omitted from the text since it has been already approached in the previous paragraph.

7. Fig. 3f: letter missing in the figure.

A: Thank you for pointing it out. Following the suggestion from another reviewer, we have moved this figure to the supplements (current Supplementary Figure 17). We added the correct labels as suggested here.

8. The effort in constructing a new figure with the main results pointing out to the importance of MAPK pathway selection is appreciated, but Figure 4a is not clear. Can the authors explain what was done in Fig. 4a? Same question regarding Fig. 4b. What does the % represent?

A: We apologize for not providing enough description for these figures. Figure 4a is aimed at illustrating the dysregulation of MAPK pathway in the human and mouse models, and the frequency with which it

appears in these analyses, and as such, we selected gene modules from individual WGCNA analyses for mice and humans by filtering for terms with MAPK/MAP kinase.

This was added to the figure legend in full detail: *“The occurrence and importance of MAPK pathways and other related kinase pathways are shown across all mouse models and the human samples, highlighting distinct activities within the co-expression modules identified by WGCNA. For that, we selected gene modules from individual WGCNA analysis for mice and humans by filtering for terms with MAPK/MAP kinase. The upper panel shows the significance (-log₁₀ p-value) and the lower panel shows absolute counts. The legend below the bars depicts the origin of the hits.”*

We have now also added a more detailed figure caption for Fig 4b, describing the enrichment method and the used threshold: *“GO-Enrichment results for DAS genes. Bar heights represent the fraction of Gene Ratio of differentially alternatively spliced genes in the pathways. All pathways have an adjusted p-value < 0.1.”*

9. How was the target centrality calculated in Fig. 4e?

A: Thank you for pointing that out. We added the description to the section “Small RNA sequencing data-processing and miRNA target prediction” in the manuscript (lines 838-840):

“Target network centrality was calculated using networkx (v2.8.8) using the eigenvector_centrality function on the STRING interaction network the miRNA targets.”

10. Referral to Fig. 5e-f missing in the main text.

A: We have fixed the text and made sure to cite every individual panel in the text accordingly (lines 369-373).

Reviewer #4 (Remarks to the Author):

I am satisfied that all my comments have been addressed.

REVIEWERS' COMMENTS

Reviewer #1 (Remarks to the Author):

The authors have thoughtfully considered or addressed all my comments.

Remarks on code availability:

I do not think that having an externally reproducible MAXOMOD pipeline is necessary for the publication of this paper since the pertinent code chunks for a given analysis can be identified in the neatly organized code. I also acknowledge the necessity of depositing human omics data in appropriate databases. However, if the authors wish the pipeline to be externally reproducible or used as a community resource, additional information about the pipeline inputs (data formats, file types, file locations) should be provided in the README. As an additional step, having a functioning sample dataset would be useful.

Reviewer #3 (Remarks to the Author):

I thank the authors for their substantial efforts in revising the manuscript and attempts to address my concerns. The evidence supporting MAPK pathway as a possible therapeutic avenue to treat ALS (in females) remains in my view not very strong. While I appreciate that NFL levels have been very informative as the authors highlight (ie clinical trial of tofersen), the effect observed in females with trametinib is modest compared to non-treated animals. Additionally, it is not clear how the levels after treatment compare with those measured in control non-transgenic animals (this control was missing so it is not possible to directly compare).

The authors reiterate that the treatment with trametinib leads to statistically significant beneficial effects— which based on the statistical test provided it is indeed the case, but the effects measured are simply miniscule. In a cohort of <10 animals, it is difficult to ascertain that the effects are relevant when there is only an extension of 5 days (with already a spread of +/- 3.5-5 days) and 6.7 days (with a variability of +/- 6.5-8 days) (out of 165 days in total) in onset and survival, respectively. This is consistent with the absence of beneficial effect observed in behavioral testing (for which the data is not provided). Altogether, this makes me question the therapeutic value of the approach proposed.

Again this reviewer appreciates the effort in providing additional data such as the measure of AchR subunit mRNA as a possible proxy for neuromuscular junction (NMJ) denervation, but this is not standard. It is not clear why the authors do not show, instead, the effect of trametinib on NMJ innervation/denervation using fluorescence imaging for example, which is more relevant.

With this, I do not have further requests to be addressed.

REMAINING REVIEWER COMMENTS

Reviewer #1 (Remarks to the Author):

The authors have thoughtfully considered or addressed all my comments.

Remarks on code availability:

I do not think that having an externally reproducible MAXOMOD pipeline is necessary for the publication of this paper since the pertinent code chunks for a given analysis can be identified in the neatly organized code. I also acknowledge the necessity of depositing human omics data in appropriate databases. However, if the authors wish the pipeline to be externally reproducible or used as a community resource, additional information about the pipeline inputs (data formats, file types, file locations) should be provided in the README. As an additional step, having a functioning sample dataset would be useful.

A: We thank the reviewer for these considerations. The human omics data have in fact been deposited in the EGA database (for RNA seq experiments) and in PRIDE (for proteomics experiments). Regarding the additional information about the code in a README file, we have now added further instructions about the download locations for the datasets and additional information for the pipeline inputs including data formats and file locations. Furthermore, we provide scripts to download and organize the mouse RNAseq & miRNA data for easier reproducibility.

Reviewer #3 (Remarks to the Author):

I thank the authors for their substantial efforts in revising the manuscript and attempts to address my concerns. The evidence supporting MAPK pathway as a possible therapeutic avenue to treat ALS (in females) remains in my view not very strong. While I appreciate that NfL levels have been very informative as the authors highlight (ie clinical trial of tofersen), the effect observed in females with trametinib is modest compared to non-treated animals. Additionally, it is not clear how the levels after treatment compare with those measured in control non-transgenic animals (this control was missing so it is not possible to directly compare).

A: We thank the reviewer for the critical input which helped us to improve the manuscript substantially. Regarding the point of the NfL levels in non-transgenic (NTG) animals, we observed that these animals have very low NfL plasmatic concentration (60 pg/ml) while at the same age (16 weeks) SOD1G93A mice have a concentration of about 2000 pg/ml, in female and male. This is expected since NTG animals do not present neuro-axonal degeneration and the Simoa assay is both very sensitive and exceptionally specific. In the reported experiment, when we compare untreated to treated SOD1G93A mice, plasmatic NfL at 16 weeks drops greatly from 2200 to 1060 pg/ml and at 19 weeks from 2900 to 1900 pg/ml.

The authors reiterate that the treatment with trametinib leads to statistically significant beneficial effects – which based on the statistical test provided, is indeed the case – but the effects measured are simply minuscule. In a cohort of <10 animals, it is difficult to ascertain that the effects are relevant when there is only an extension of 5 days (with already a spread of +/- 3.5-5 days) and 6.7 days (with a variability of +/- 6.5-8 days) (out of 165 days in total) in onset and survival, respectively. This is consistent with the absence of beneficial effect observed in behavioral testing (for which the data is not provided). Altogether, this makes me question the therapeutic value of the approach proposed.

Again this reviewer appreciates the effort in providing additional data such as the measure of AchR subunit mRNA as a possible proxy for neuromuscular junction (NMJ) denervation, but this is not standard. It is not clear why the authors do not show, instead, the effect of trametinib on NMJ innervation/denervation using fluorescence imaging for example, which is more relevant.

With this, I do not have further requests to be addressed.

A: The concerns regarding the small cohort size and the measured effects of trametinib are acknowledged. However, the statistical methods and cohort sizes employed currently are considered robust enough to substantiate the claims presented, although we agree that larger sample sizes could enhance the statistical power of the findings. Unfortunately, treating new animals to increase the cohort sizes would fall beyond the scope of this revision work. In response to the reviewer's request for additional data, full results from behavioral experiments have been now provided as supplements. As also requested by this reviewer, we included now fluorescence imaging of neuromuscular junction (NMJ) denervation in the updated Supplementary Figure 24.

Figure legend - Updated Supplementary Fig. 24.

Immunostaining of the neuromuscular junction and expression levels of AChR- γ as a measure of muscle denervation in trametinib-treated SOD1G93A mice.

Representative figure of NMJs in the tibialis anterior muscle of SOD1G93A male (**a**) and female (**b**) trametinib-treated and vehicle mice at 16 weeks of age. Neurofilament (2H3, red) and synaptic vesicle glycoprotein (SV2, red) were used to identify the presynaptic terminals. α -Bungarotoxin (α -BTX, green) was used to identify the postsynaptic domain. At this stage, SOD1G93A male and female mice present diffuse denervated endplates (**a-b**: i-v panels). In (**b**), panels viii, ix and x show innervated endplates in female trametinib-treated mice. Panels iv and v in figure A are magnified images of the dashed area in panels iii. Panels ix and x in (**a**) are magnified images of the dashed area in panels viii. Panels iv and v in (**b**) are magnified images of the dashed area in panels iii. Panels ix and x in

(b) are magnified images of the dashed area in panels viii. Panels: i-iii and vi-viii scale bar: 50 μm . Panels: iv-v and ix-x scale bar: 25 μm . q-RT-PCR results (relative expression, $2^{-\Delta\Delta\text{Ct}}$ values) for mouse nicotinic acetylcholinergic receptor, gamma subunit (AChR- γ) mRNA transcripts in gastrocnemius muscle of trametinib-treated and untreated (vehicle) SOD1^{G93A} male **(c)** and female **(d)** mice, at 16 weeks of age. Data (mean \pm SEM; n = 3/4 in each experimental group) are normalized to β -actin and expressed as relative mRNA. * indicates $p < 0.05$, two-tailed Student's t-test.

Furthermore, the use of AChR gamma subunit mRNA levels as a proxy for muscle denervation is justified based on its established correlation with muscle atrophy and the percentage of innervated NMJs (the overlap between neurofilament (SV2/2H3) staining and α -bungarotoxin-labelled endplates). as reported in previous studies (Margotta et al., 2023; Trolese et al., 2022). Using q-RT-PCR for the quantification of muscle denervation is argued to offer a more comprehensive assessment compared to histological analysis, which is typically limited to a small number of tissue slices and might not reflect the overall tissue status. Therefore, both mRNA data and representative images are presented to support the findings, aiming to provide a thorough evaluation of the therapeutic value of trametinib.